# Structured Local Minima in Sparse Blind Deconvolution

**Yuqian Zhang, Han-Wen Kuo, John Wright**
Department of Electrical Engineer and Data Science Institute
Columbia University, New York, NY 10027
{yz2409, hk2673, jw2966}@columbia.edu

## Abstract

Blind deconvolution is a ubiquitous problem of recovering two unknown signals from their convolution. Unfortunately, this is an ill-posed problem in general. This paper focuses on the *short and sparse* blind deconvolution problem, where the one unknown signal is short and the other one is sparsely and randomly supported. This variant captures the structure of the unknown signals in several important applications. We assume the short signal to have unit $\ell^2$ norm and cast the blind deconvolution problem as a nonconvex optimization problem over the sphere. We demonstrate that (i) in a certain region of the sphere, every local optimum is close to some shift truncation of the ground truth, and (ii) for a generic short signal of length $k$, when the sparsity of activation signal $\theta \lesssim k^{-2/3}$ and number of measurements $m \gtrsim \mathrm{poly}\,(k)$, a simple initialization method together with a descent algorithm which escapes strict saddle points recovers a near shift truncation of the ground truth kernel.

## 1 Introduction

Blind deconvolution is the problem of recovering two unknown signals $\boldsymbol{a}_0$ and $\boldsymbol{x}_0$ from their convolution $\boldsymbol{y} = \boldsymbol{a}_0 * \boldsymbol{x}_0$. This fundamental problem recurs across several fields, including astronomy, microscopy data processing [1], neural spike sorting [2], computer vision [3], etc. However, this problem is ill-posed without further priors on the unknown signals, as there are infinitely many pairs of signals $(\boldsymbol{a}, \boldsymbol{x})$ whose convolution equals a given observation $\boldsymbol{y}$. Fortunately, in practice, the target signals $(\boldsymbol{a}, \boldsymbol{x})$ are often structured. In particular, a number of practical applications exhibit a common *short-and-sparse* structure:

In *Neural spike sorting*: Neurons in the brain fire brief voltage spikes when stimulated. The signatures of the spikes encode critical features of the neuron and the occurrence of such spikes are usually sparse and random in time [2, 4].

In *Microscopy data analysis*: Some nanoscale materials are contaminated by randomly and sparsely distributed "defects", which change the electronic structure of the material [1].

In *Image deblurring*: Blurred images due to camera shake can be modeled as a convolution of the latent sharp image and a kernel capturing the motion of the camera. Although natural images are not sparse, they typically have (approximately) sparse gradients [5, 6].

In the above applications, the observation signal $\boldsymbol{y} \in \mathbb{R}^m$ is generated via the convolution of a *short* kernel $\boldsymbol{a}_0 \in \mathbb{R}^k (k \ll m)$ and a *sparse* activation coefficient $\boldsymbol{x}_0 \in \mathbb{R}^m$ ($\|\boldsymbol{x}_0\|_0 \ll m$). Without loss of generality, we let $\boldsymbol{y}$ denote the circular convolution of $\boldsymbol{a}_0$ and $\boldsymbol{x}_0$

$$\boldsymbol{y} = \boldsymbol{a}_0 \circledast \boldsymbol{x}_0 = \widetilde{\boldsymbol{a}_0} \circledast \boldsymbol{x}_0, \tag{1}$$

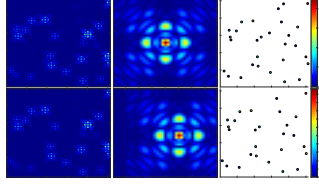

**Figure 1: Local Minimum.** Top: observation $\boldsymbol{y} = \boldsymbol{a}_0 \circledast \boldsymbol{x}_0$, and ground truth $\boldsymbol{a}_0$, and $\boldsymbol{x}_0$; Bottom: recovered $\boldsymbol{a} \circledast \boldsymbol{x}$, $\boldsymbol{a}$, and $\boldsymbol{x}$ at one local minimum of a natural formulation in [16].

with $\widetilde{\boldsymbol{a}_0} \in \mathbb{R}^m$ denoting the zero padded $m$-length version of $\boldsymbol{a}_0$, which can be expressed as $\widetilde{\boldsymbol{a}_0} = \iota_k \boldsymbol{a}_0$. Here, $\iota_k : \mathbb{R}^k \to \mathbb{R}^m$ is a zero padding operator. Its adjoint $\iota_k^* : \mathbb{R}^m \to \mathbb{R}^k$ acts as a projection onto the lower dimensional space by keeping the first $k$ components.

The short-and-sparse blind deconvolution problem exhibits a *scaled-shift ambiguity*, which derives from the basic properties of a convolution operator. Namely, for any observation signal $\boldsymbol{y}$, and any nonzero scalar $\alpha$ and integer shift $\tau$, the following equality always holds

$$\boldsymbol{y} = (\pm\alpha s_\tau[\widetilde{\boldsymbol{a}_0}]) \circledast \left(\pm\alpha^{-1} s_{-\tau}[\boldsymbol{x}_0]\right). \tag{2}$$

Here, $s_{-\tau}[\boldsymbol{v}]$ denotes the cyclic shift of the vector $\boldsymbol{v}$ by $\tau$ entries:

$$s_\tau[\boldsymbol{v}](i) = \boldsymbol{v}\left([i - \tau - 1]_m + 1\right), \quad \forall\, i \in \{1, \cdots, m\}. \tag{3}$$

Clearly, both scaling and cyclic shifts preserve the short-and-sparse structure of $(\boldsymbol{a}_0, \boldsymbol{x}_0)$. This *scaled-shift symmetry* raises nontrivial challenges for computation, making straightforward convexification approaches ineffective.[1]

Nonconvex algorithms for sparse blind deconvolution have been well developed and practiced, especially in computer vision [12, 13, 14, 15]. Despite its empirical success, little was known about its working mechanism. Recently, [16] studies the optimization landscape of the natural nonconvex formulation for sparse blind deconvolution, assuming the kernel $\boldsymbol{a} \in \mathbb{R}^k$ to have unit Frobenius norm (denote as $\boldsymbol{a} \in \mathbb{S}^{k-1}$). [16] argues that under conditions, this problem has well-structured local optima, in the sense that *every local optimum is close to some shift truncation of the ground truth* (Figure 1).

The presence of these local optima can be viewed as a result of the shift symmetry associated to the convolution operator: the shifted and truncated kernel $\iota_k^* s_\tau[\widetilde{\boldsymbol{a}_0}]$ can be convolved with the sparse signal $s_{-\tau}[\boldsymbol{x}_0]$ (shifted in the other direction) to produce a near approximation to the observation $\iota_k^* s_\tau[\widetilde{\boldsymbol{a}_0}] \circledast s_{-\tau}[\boldsymbol{x}_0] \approx \boldsymbol{y}$.

In [16], this geometric insight about local optima is corroborated with a lot of experiments, but rigorous proof is only available in the "dilute limit" in which the sparse coefficient signal $\boldsymbol{x}_0$ is a single spike. In this paper, we adopt the unit Frobenius norm constraint for the short convolution kernel $\boldsymbol{a}$ as in [16], but consider a different objective function. We formulate the sparse blind deconvolution problem as the following optimization problem over the sphere:

$$\min\ -\|\check{\boldsymbol{y}} \circledast \boldsymbol{r}_{\boldsymbol{y}}(\boldsymbol{q})\|_4^4 \quad \text{s.t.} \quad \|\boldsymbol{q}\|_F = 1. \tag{4}$$

Here, $\check{\boldsymbol{y}}$ denotes the reversal of $\boldsymbol{y}$[2] and $\boldsymbol{r}_{\boldsymbol{y}}(\boldsymbol{q})$ is a preconditioner which we will discuss in detail later. Convolution $\boldsymbol{y} \circledast \boldsymbol{r}_{\boldsymbol{y}}(\boldsymbol{q})$ approximates the reversed underlying activation signal $\boldsymbol{x}_0$, and $-\|\cdot\|_4^4$ serves as the sparsity penalty.

We demonstrate that even when $\boldsymbol{x}_0$ is relatively dense, any local minimum in certain region of the sphere is close to a shift truncation $\iota_k^* s_\tau[\widetilde{\boldsymbol{a}_0}]$ of the ground truth. This benign region contains the sub-level set of small objective value. Algorithmically, if initialized at a point with small enough objective value, then a descent algorithm always decreases the objective value and hence stays in this region. Specifically, for a generic kernel[3] $\boldsymbol{a}_0 \in \mathbb{S}^{k-1}$, if the

sparsity rate[4] $\theta \lesssim k^{-2/3}$ and the number of measurement $m \gtrsim \text{poly}(k)$, initializing at some preconditioned $k$ consecutive entries of $\boldsymbol{y}$, and applying any descent method that converges to a local minimizer under a strict saddle hypothesis [17, 18], produces a near shift-truncation of the ground truth.[5]

**Assumptions and Notations**  We assume that $\boldsymbol{x}_0 \in \mathbb{R}^m$ follows Bernoulli-Gaussian (BG) model with sparsity level $\theta$: $\boldsymbol{x}_0(i) = \omega_i g_i$ with $\omega_i \sim \text{Ber}(\theta)$ and $g_i \sim \mathcal{N}(0,1)$, where all the different random variables are jointly independent. For simplicity, we write $\boldsymbol{x}_0 \sim_{\text{i.i.d.}} \text{BG}(\theta)$. Throughout this paper, a vector $\boldsymbol{v} \in \mathbb{R}^k$ is indexed as $\boldsymbol{v} = [v_1, v_2, \cdots, v_k]$, and $[\cdot]_m$ denotes the modulo operator of $m$. We use $\|\cdot\|_{\text{op}}, \|\cdot\|_F$, and $\|\cdot\|_p$ to denote operator norm, Frobenius norm, and entry wise $\ell^p$ norm respectively. $\mathcal{P}_{\mathbb{S}}[\cdot] \doteq \frac{\cdot}{\|\cdot\|_F}$ denotes projection onto the Frobenius sphere. $(\cdot)^{\circ p}$ is the entry wise $p$-th order exponent operator. We use $C, c$ to denote positive constants, and their value change across the paper.

## 2  Problem Formulation

In the short-and-sparse blind deconvolution problem, any $k$ consecutive entries in $\boldsymbol{y}$ only depend on $2k-1$ consecutive entries in $\boldsymbol{x}_0$:

$$\boldsymbol{y}_i = \left[y_i, \cdots, y_{1+[i+k-1]_m}\right]^T = \sum_{\tau=-(k-1)}^{k-1} x_{1+[i+\tau-1]_m} \cdot \boldsymbol{\iota}_k^* s_\tau[\widetilde{\boldsymbol{a}_0}] \qquad (5)$$

$$= \underbrace{\begin{bmatrix} a_k & a_{k-1} & \cdots & a_1 & \cdots & 0 & 0 \\ 0 & a_k & \cdots & a_2 & \cdots & 0 & 0 \\ \vdots & \vdots & \ddots & \vdots & \ddots & \vdots & \vdots \\ 0 & 0 & \cdots & a_{k-1} & \cdots & a_1 & 0 \\ 0 & 0 & \cdots & a_k & \cdots & a_2 & a_1 \end{bmatrix}}_{\boldsymbol{A}_0 \in \mathbb{R}^{k \times (2k-1)}} \underbrace{\begin{bmatrix} x_{1+[i-k]_m} \\ \vdots \\ x_i \\ \vdots \\ x_{1+[i+k-2]_m} \end{bmatrix}}_{\boldsymbol{x}_i \in \mathbb{R}^{(2k-1) \times 1}}. \qquad (6)$$

Write $\boldsymbol{Y} = [\boldsymbol{y}_1, \boldsymbol{y}_2, \ldots, \boldsymbol{y}_m] \in \mathbb{R}^{k \times m}$ and $\boldsymbol{X}_0 = [\boldsymbol{x}_1, \ldots, \boldsymbol{x}_m] \in \mathbb{R}^{2k-1 \times m}$. Using the above expression, we have that

$$\boldsymbol{Y} = \boldsymbol{A}_0 \boldsymbol{X}_0. \qquad (7)$$

Each column $\boldsymbol{x}_i$ of $\boldsymbol{X}_0$ only contains some $2k-1$ entries of $\boldsymbol{x}_0$. The *rows* of $\boldsymbol{X}_0$ are cyclic shifts of the reversal of $\boldsymbol{x}_0$:

$$\boldsymbol{X}_0 = \begin{bmatrix} s_0[\check{\boldsymbol{x}}_0] \\ \vdots \\ s_{2k-2}[\check{\boldsymbol{x}}_0] \end{bmatrix}. \qquad (8)$$

The shifts of $\check{\boldsymbol{x}}_0$ are *sparse vectors* in the linear subspace $\text{row}(\boldsymbol{X}_0)$. Note that if we could recover some shift $s_\tau[\boldsymbol{x}_0]$, we could subsequently determine $s_{-\tau}[\boldsymbol{a}_0]$ by solving a linear system of equations, and hence solve the deconvolution problem, up to the shift ambiguity.[6]

### 2.1  Finding a Shifted Sparse Signal

In light of the above observations, a natural computational approach to sparse blind deconvolution is to attempt to find $\boldsymbol{x}_0$ by searching for a sparse vector in the linear subspace $\text{row}(\boldsymbol{X}_0)$, e.g., by solving an optimization problem

$$\min \quad \|\boldsymbol{v}\|_\star \quad \text{s.t.} \quad \boldsymbol{v} \in \text{row}(\boldsymbol{X}_0), \ \|\boldsymbol{v}\|_2 = 1, \qquad (9)$$

where $\|\cdot\|_\star$ is chosen to encourage sparsity of the target signal [20, 21, 22, 23].

In sparse blind deconvolution, we do not have access to the row space of $X_0$. Instead, we only observe the subspace $\mathrm{row}(Y) \subset \mathrm{row}(X_0)$. The subspace $\mathrm{row}(Y)$ does not necessarily contain the desired sparse vector $e_i^T X_0$, but it *does* contain some approximately sparse vectors. In particular, consider following vector in $\mathrm{row}(Y)$,

$$v = Y^T a_0 = \underbrace{\check{x}_0}_{\text{sparse}} + \underbrace{\sum_{i\neq 0} \langle a_0, s_i[a_0]\rangle s_i[\check{x}_0]}_{\text{"noise" } z}. \tag{10}$$

The vector $v$ is a superposition of a sparse signal $\check{x}_0$ and its scaled shifts $\langle a_0, s_i[a_0]\rangle s_i[\check{x}_0]$. If the shift-coherence $|\langle a_0, s_\tau[a_0]\rangle|$ is small[7] and $x_0$ is sparse enough, $z$ can be viewed as small noise.[8] The vector $v$ is not sparse, but it is *spiky*: a few of its entries are much larger than the rest. We deploy a milder sparsity penalty $-\|\cdot\|_4^4$ to recover such a spiky vector, as $\|\cdot\|_4^4$ is very flat around 0 and insensitive to small noise in the signal.[9] This gives

$$\min \quad -\tfrac{1}{4}\|v\|_4^4 \quad \text{s.t.} \quad v \in \mathrm{row}(Y), \; \|v\|_2 = 1. \tag{11}$$

We can express a generic unit vector $v \in \mathrm{row}(Y)$ as $v = Y^T (YY^T)^{-1/2} q$, with $\|v\|_2 = \|q\|_2$. This leads to the following equivalent optimization problem over the sphere

$$\min \; \psi(q) \doteq -\frac{1}{4m}\left\|Y^T (YY^T)^{-1/2} q\right\|_4^4 \quad \text{s.t.} \quad \|q\|_2 = 1. \tag{12}$$

**Interpretation: preconditioned shifts.** This objective $\psi(q)$ can be rewritten as

$$\psi(q) = -\frac{1}{4m}\left\|\check{y} \circledast (YY^T)^{-1/2} q\right\|_4^4 = -\frac{1}{4m}\left\|\check{x}_0 \circledast A_0^T (YY^T)^{-1/2} q\right\|_4^4 \sim \|\check{x}_0 \circledast \zeta\|_4^4, \tag{13}$$

where $\zeta = A_0^T (A_0 A_0^T)^{-1/2} q$. This approximation becomes accurate as $m$ grows.[10] This objective encourages the convolution of $\check{x}_0$ and $\zeta$ to be as spiky as possible. Reasoning analogous to (10) suggests that $\check{x}_0 \circledast \zeta$ will be spiky if

$$\zeta = A_0^T (A_0 A_0^T)^{-1/2} q \approx e_l, \quad l \in \{1, \cdots, 2k-1\}. \tag{14}$$

For simplicity, we define the preconditioned convolution matrix

$$A \doteq (A_0 A_0^T)^{-1/2} A_0 = [a_1 \quad a_2 \quad \cdots \quad a_{2k-1}], \tag{15}$$

with column coherence (preconditioned shift coherence) $\mu \doteq \max_{i\neq j}|\langle a_i, a_j\rangle|$. Then $\zeta$ can also be interpreted as measuring the inner products of $q$ with columns of $A$. Making this intuition rigorous, we will show that minimizing this objective over a certain region of the sphere yields a preconditioned shift truncate $a_l$, from which we can recover a shift truncate of the original signal $a_0$.

## 2.2 Structured Local Minima

We will show that in a certain region $\mathcal{R}_{C_\star} \subset \mathbb{S}^{k-1}$, the preconditioned shift truncations $a_l$ are the *only* local minimizers. Moreover, the other critical points in $\mathcal{R}_{C_\star}$ can be interpreted as resulting from competition between several of these local minima (Figure 2). At any saddle point, there exists strict negative curvature in the direction of a nearby local minimizer which breaks the balance in favor of some particular $a_l$. The region $\mathcal{R}_{C_\star}$ is defined as follows:

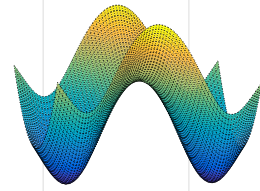

**Figure 2: Saddles points are approximately superpositions of local minima.**

**Definition 2.1.** *For fixed $C_\star > 0$, letting $\kappa$ denote the condition number of $A_0$, and $\mu \doteq \max_{i\neq j}|\langle a_i, a_j\rangle|$ the column coherence of $A$, we define two regions $\mathcal{R}_{C_\star}, \hat{\mathcal{R}}_{C_\star} \subset \mathbb{S}^{k-1}$, as*

$$\mathcal{R}_{C_\star} \doteq \left\{ q \in \mathbb{S}^{k-1} \,|\, \|A^T q\|_4^6 \geq C_\star \mu \kappa^2 \|A^T q\|_3^3 \right\}. \tag{16}$$

$$\hat{\mathcal{R}}_{C_\star} \doteq \left\{ q \in \mathbb{S}^{k-1} \,|\, \|A^T q\|_4^6 \geq C_\star \mu \kappa^2 \right\} \subseteq \mathcal{R}_{C_\star}. \tag{17}$$

A simpler and smaller region $\hat{\mathcal{R}}_{C_\star}$ is also introduced in Definition (2.1). This region $\hat{\mathcal{R}}_{C_\star}$ can be viewed as a sub-level set for $-\left\|\boldsymbol{A}^T\boldsymbol{q}\right\|_4^4$, which is proportional to the objective value $\psi(\boldsymbol{q})$ assuming $m$ is sufficiently large[11]. Therefore, once initialized within $\hat{\mathcal{R}}_{C_\star}$, the iterates produced by a descent algorithm will stay in $\hat{\mathcal{R}}_{C_\star}$.

In particular, at any stationary point $\boldsymbol{q} \in \mathcal{R}_{10}$, the local optimization landscape can be characterized in terms of the number of spikes (entries with nontrivial magnitude[12]) in $\boldsymbol{\zeta}$. If there is only one spike in $\boldsymbol{\zeta}$, then such stationary point $\boldsymbol{q}$ is a local minimum that is close to one local minimizer; if there are more than two spikes in $\boldsymbol{\zeta}$, then such stationary point $\boldsymbol{q}$ is saddle point. Based on the above characterizations of stationary points in $\mathcal{R}_{C_\star}$ with $C_\star \geq 10$, we can deduce that any local minimum is close to $\boldsymbol{a}_l$ for some integer $l$, a preconditioned shift truncation of the ground truth $\boldsymbol{a}_0$.

**Theorem 2.2** (Main Result). *Assuming observation $\boldsymbol{y} \in \mathbb{R}^m$ is the circulant convolution of $\boldsymbol{a}_0 \in \mathbb{R}^k$ and $\boldsymbol{x}_0 \sim_{\text{i.i.d.}} \text{BG}(\theta) \in \mathbb{R}^m$, where the convolutional matrix $\boldsymbol{A}_0$ has minimum singular value $\sigma_{\min} > 0$ and condition number $\kappa \geq 1$, and $\boldsymbol{A}$ has column incoherence $0 \leq \mu < 1$. There exists a positive constant $C$ such that whenever the number of measurements*

$$m \geq C \frac{\min\left\{\mu^{-4/3}, \kappa^2 k^2\right\}}{(1-\theta)^2 \sigma_{\min}^2} \kappa^8 k^4 \log^3\left(\frac{\kappa k}{(1-\theta)\sigma_{\min}}\right) \tag{18}$$

*and $\theta \geq \log k / k$, then with high probability, any local optima $\bar{\boldsymbol{q}} \in \hat{\mathcal{R}}_{2C_\star}$ satisfies*

$$|\langle\bar{\boldsymbol{q}}, \mathcal{P}_{\mathbb{S}}[\boldsymbol{a}_l]\rangle| \geq 1 - c_\star \kappa^{-2} \tag{19}$$

*for some integer $1 \leq l \leq 2k-1$. Here, $C_\star \geq 10$ and $c_\star = 1/C_\star$.*

This theorem says that any local minimum in $\hat{\mathcal{R}}_{2C_\star}$ is close to some normalized column of $\boldsymbol{A}$ given polynomially many observation. The parameters $\sigma_{\min}$, $\kappa$ and $\mu$ effectively measure the spectrum flatness of the ground truth kernel $\boldsymbol{a}_0$ and characterize how broad the results hold. A random like kernel usually has big $\sigma_{\min}$, small $\kappa$ and $\mu$, which equivalently implies the result holds in a large sub-level set $\hat{\mathcal{R}}_{2C_\star}$ even with fewer observations.

Hence, once assuring the algorithm finds a local minimum in $\hat{\mathcal{R}}_{2C_\star}$, then some shifted truncation of the ground truth kernel $\boldsymbol{a}_0$ can be recovered. In other words, if we can find an initialization point with small objective value then a descent algorithm minimizing the objective function guarantees that $\boldsymbol{q}$ always stays in $\hat{\mathcal{R}}_{2C_\star}$ in proceeding iterations. Therefore, any descent algorithm that escapes a strict saddle point can be applied to find some $\boldsymbol{a}_l$, or some shift truncation of $\boldsymbol{a}_0$.

### 2.3 Initialization with a Random Sample

Recall that $\boldsymbol{y}_i = \boldsymbol{A}_0\boldsymbol{x}_i$, which is a sparse superposition of about $2\theta k$ columns of $\boldsymbol{A}_0$. Intuitively speaking, such $\boldsymbol{q}_{\text{init}}$ already encodes certain preferences towards a few preconditioned shift truncations of the ground truth. Therefore, we randomly choose an index $i$ and set the initialization point as

$$\boldsymbol{q}_{\text{init}} = \mathcal{P}_{\mathbb{S}}\left[\left(\boldsymbol{Y}\boldsymbol{Y}^T\right)^{-1/2}\boldsymbol{y}_i\right], \quad \boldsymbol{\zeta}_{\text{init}} = \boldsymbol{A}^T\boldsymbol{q}_{\text{init}} \approx \mathcal{P}_{\mathbb{S}}\left[\boldsymbol{A}^T\boldsymbol{A}\boldsymbol{x}_i\right].^{13} \tag{20}$$

For a generic kernel $\boldsymbol{a}_0 \in \mathbb{S}^{k-1}$, $\boldsymbol{A}^T\boldsymbol{A}$ is close to a diagonal matrix, as the magnitudes of off-diagonal entries are bounded by column incoherence $\mu$. Hence, the sparse property of $\boldsymbol{x}_i$ can be approximately preserved, that $\mathcal{P}_{\mathbb{S}}\left[\boldsymbol{A}^T\boldsymbol{A}\boldsymbol{x}_i\right]$ is spiky vector with small $-\left\|\cdot\right\|_4^4$. By leveraging the sparsity level $\theta$, one can make sure such initialization point $\boldsymbol{q}_{\text{init}}$ falls in $\hat{\mathcal{R}}_{2C_\star}$. Therefore, we propose Algorithm 1 for solving sparse blind deconvolution with its working conditions stated in Corollary 2.3. For the choice of descent algorithms which escape strict saddle points, there are several such algorithms specially tailored for sphere constrained optimization problems [24, 25].

**Algorithm 1** Short and Sparse Blind Deconvolution

---

**Input:** Observations $\boldsymbol{y} \in \mathbb{R}^m$ and kernel size $k$.
**Output:** Recovered Kernel $\bar{\boldsymbol{a}}$.

1: Generate random index $i \in [1, m]$ and set $\boldsymbol{q}_{init} = \mathcal{P}_{\mathbb{S}} \left[ \left( \boldsymbol{Y} \boldsymbol{Y}^T \right)^{-1/2} \boldsymbol{y}_i \right]$.
2: Solve following nonconvex optimization problem with a descent algorithm that escapes saddle point and find a local minimizer $\bar{\boldsymbol{q}} = \arg\min_{\boldsymbol{q} \in \mathbb{S}^{k-1}} \varphi(\boldsymbol{q})$
3: Set $\bar{\boldsymbol{a}} = \mathcal{P}_{\mathbb{S}} \left[ \left( \boldsymbol{Y} \boldsymbol{Y}^T \right)^{1/2} \bar{\boldsymbol{q}} \right]$.

---

**Corollary 2.3.** *Suppose the ground truth $\boldsymbol{a}_0$ kernel has preconditioned shift coherence $0 \leq \mu \leq \frac{1}{8 \times 48} \log^{-3/2}(k)$ and sparse coefficient $\boldsymbol{x}_0 \sim_{\text{i.i.d.}} \mathrm{BG}(\theta) \in \mathbb{R}^m$. There exist positive constants $C \geq 2560^4$ and $C'$ such that whenever the sparsity level*

$$64k^{-1} \log k \leq \theta \leq \min \left\{ \tfrac{1}{48^2} \mu^{-2} k^{-1} \log^{-2} k, \left( \tfrac{1}{4} - \tfrac{640}{C^{1/4}} \right) \left( 3 C_\star \mu \kappa^2 \right)^{-2/3} k^{-1} \left( 1 + 36 \mu^2 k \log k \right)^{-2} \right\}$$

*and signal length*

$$m \geq C' \max \left\{ \frac{\theta^2 \kappa^6}{\sigma_{\min}^2} k^3 \left( 1 + 36 \mu^2 k \log k \right)^4 \log \left( \frac{\kappa k}{\sigma_{\min}} \right), \frac{\min \left\{ \mu^{-4/3}, \kappa^2 k^2 \right\}}{(1-\theta)^2 \sigma_{\min}^2} \kappa^8 k^4 \log^3 \left( \frac{\kappa k}{(1-\theta) \sigma_{\min}} \right) \right\},$$

*then with high probability, Algorithm 1 recovers $\bar{\boldsymbol{a}}$ such that*

$$\left\| \bar{\boldsymbol{a}} \pm \mathcal{P}_{\mathbb{S}} \left[ \iota_k s_\tau [\widetilde{\boldsymbol{a}_0}] \right] \right\|_2 \leq 2\sqrt{2} c_\star \tag{21}$$

*for some integer shift $-(k-1) \leq \tau \leq k-1$.*

For a generic $\boldsymbol{a}_0 \in \mathbb{S}^{k-1}$, plugging in the numerical estimation of the parameters $\sigma_{\min}$, $\kappa$ and $\mu$ (Figure 3), accurate recovery can be obtained with $m \gtrsim \theta^2 k^6 \operatorname{poly} \log(k)$ measurements and sparsity level $\theta \lesssim k^{-2/3} \operatorname{poly} \log(k)$. For bandpass kernels $\boldsymbol{a}_0$, $\sigma_{\min}$ is smaller and $\kappa$, $\mu$ are larger, and so our results require $\boldsymbol{x}_0$ to be longer and sparser.

## 3 Optimization Function Landscape

We next briefly present the key elements in deriving the main results of this paper. We first investigate the stationary points of the "population" objective $\mathbb{E}_{\boldsymbol{x}_0}[\psi(\boldsymbol{q})]$. We demonstrate that any local minimizer in $\mathcal{R}_{C_\star}$ is close to a signed column of $\boldsymbol{A}$, a preconditioned shift truncation of $\boldsymbol{a}_0$. We then demonstrate that when $m$ is sufficiently large, the "finite sample" objective $\psi(\boldsymbol{q})$ has similar properties.

Using $\mathbb{E}[\boldsymbol{Y} \boldsymbol{Y}^T] = \theta m \boldsymbol{A}_0 \boldsymbol{A}_0^T$ again, the expectation of the objective function $\psi(\boldsymbol{q})$ can be approximated as follows:

$$\mathbb{E}[\psi(\boldsymbol{q})] \approx \mathbb{E} \left[ -\frac{1}{m} \left\| \boldsymbol{Y}^T \left( \theta m \boldsymbol{A}_0 \boldsymbol{A}_0^T \right)^{-1/2} \boldsymbol{q} \right\|_4^4 \right] = -\frac{3(1-\theta)}{\theta m^2} \left\| \boldsymbol{A}^T \boldsymbol{q} \right\|_4^4 - \frac{3}{m^2}. \tag{22}$$

This approximation can be made rigorous (see Lemma 2.1 of the supplementary material), allowing us to study the critical points of $\mathbb{E}[\psi]$ by studying the simpler problem

$$\min_{\boldsymbol{q} \in \mathbb{R}^{k-1}} \varphi(\boldsymbol{q}) \doteq -\frac{1}{4} \left\| \boldsymbol{A}^T \boldsymbol{q} \right\|_4^4 = -\frac{1}{4} \left\| \boldsymbol{\zeta} \right\|_4^4. \tag{23}$$

The Euclidean gradient and Riemannian gradient [26] of $\varphi$ are

$$\nabla \varphi(\boldsymbol{q}) = -\boldsymbol{A} \boldsymbol{\zeta}^{\circ 3}, \quad \operatorname{grad}[\varphi](\boldsymbol{q}) = -\boldsymbol{A} \boldsymbol{\zeta}^{\circ 3} + \boldsymbol{q} \left\| \boldsymbol{\zeta} \right\|_4^4. \tag{24}$$

### 3.1 Critical Points of the Population Objective

We wish to argue that every local minimizer of $\varphi$ is close to a preconditioned shift-truncation $\boldsymbol{a}_i$. We do this by showing that at any *other* critical point, there is a direction of strict negative curvature. We will show that at any critical point $\boldsymbol{q} \in \mathcal{R}_4$, the correlation $\boldsymbol{\zeta}$ exhibits a very special structure:

(P) The entries $\boldsymbol{\zeta}_i = \langle \boldsymbol{a}_i, \boldsymbol{q} \rangle$ are either close to zero, or have magnitude $|\boldsymbol{\zeta}_i|$ close to $\|\boldsymbol{\zeta}\|_4^4 / \|\boldsymbol{a}_i\|_2$.

We can demonstrate this property directly from the stationarity condition $\mathrm{grad}\,[\varphi]\,(\boldsymbol{q}) = \boldsymbol{0}$.

$$\boldsymbol{A}\boldsymbol{\zeta}^{\circ 3} - \boldsymbol{q}\,\|\boldsymbol{\zeta}\|_4^4 = \boldsymbol{0} \quad \Rightarrow \quad \boldsymbol{A}^T\boldsymbol{A}\boldsymbol{\zeta}^{\circ 3} - \boldsymbol{A}^T\boldsymbol{q}\,\|\boldsymbol{\zeta}\|_4^4 = \boldsymbol{0}. \tag{25}$$

The $i$-th entry $\zeta_i$ of the correlation $\boldsymbol{\zeta}$ therefore satisfies the following cubic equation

$$\|\boldsymbol{a}_i\|_2^2\,\zeta_i^3 + \sum_{j\neq i} \langle \boldsymbol{a}_i, \boldsymbol{a}_j \rangle\,\zeta_j^3 - \zeta_i\,\|\boldsymbol{\zeta}\|_4^4 = 0 \quad \Rightarrow \quad \zeta_i^3 - \zeta_i\,\underbrace{\frac{\|\boldsymbol{\zeta}\|_4^4}{\|\boldsymbol{a}_i\|_2^2}}_{\alpha_i} + \underbrace{\frac{\sum_{j\neq i}\langle \boldsymbol{a}_i, \boldsymbol{a}_j\rangle\,\zeta_j^3}{\|\boldsymbol{a}_i\|_2^2}}_{\beta_i} = 0. \tag{26}$$

If $\alpha_i \gg \beta_i$, the roots of (26) are either very close to $0$, or very close to $\pm\sqrt{\alpha_i}$. The condition $\alpha_i \gg \beta_i$ obtains whenever $\|\boldsymbol{A}^T\boldsymbol{q}\|_4^6 \geq 4\mu\,\|\boldsymbol{A}^T\boldsymbol{q}\|_3^3$, and hence on $\mathcal{R}_4$, every critical point satisfies property (P).

## 3.2 Asymptotic Function Landscape on $\mathcal{R}_{C_\star}$

The local optimization landscape around any stationary point $\boldsymbol{q}$ is characterized by the Riemannian Hessian. In particular, at a stationary point $\boldsymbol{q}$, if $\mathrm{Hess}\,[\varphi]\,(\boldsymbol{q})$ is positive semidefinite, then the function is convex and $\boldsymbol{q}$ is a local minimum; if $\mathrm{Hess}\,[\varphi]\,(\boldsymbol{q})$ has a negative eigenvalue, then there exists a direction along which the objective value decreases and $\boldsymbol{q}$ is a saddle point. Technically, on $\mathcal{R}_{C_\star}$ with $C_\star \geq 10$, the minimum eigenvalue of the Riemannian Hessian can be controlled based on the spikiness of $\boldsymbol{\zeta}$.

First, we demonstrate that once constrained in $\mathcal{R}_{C_\star}$ with $C_\star \geq 10$, then any stationary point must have cross correlation $\boldsymbol{\zeta}$ with entries of nontrivial magnitude, or entries of $\boldsymbol{\zeta}$ cannot be simultaneously close to $0$. Geometrically, this implies that any stationary point $\boldsymbol{q} \in \mathcal{R}_{C_\star}$ should be "close" to certain preconditioned shift truncations.

**Lemma 3.1.** *For any stationary point $\boldsymbol{q} \in \mathcal{R}_{C_\star}$ with $C_\star \geq 10$, magnitude of vector $\boldsymbol{\zeta} = \boldsymbol{A}^T\boldsymbol{q}$ cannot be uniformly bounded by $2\mu\,\|\boldsymbol{\zeta}\|_3^3 / \|\boldsymbol{\zeta}\|_4^4$.*

**Local Minima** If $\boldsymbol{q}$ is a stationary point in $\mathcal{R}_{C_\star}$ with $C_\star \geq 10$, and $\boldsymbol{\zeta}$ only has one single entry $\zeta_l$ with magnitude larger than $2\mu\,\|\boldsymbol{\zeta}\|_3^3 / \|\boldsymbol{\zeta}\|_4^4$, then the Riemannian Hessian $\mathrm{Hess}[\varphi]\,(\boldsymbol{q})$ is always positive definite, and the function is locally convex. In addition, $|\langle \boldsymbol{q}, \boldsymbol{a}_l \rangle| > \left(1 - 2C_\star^{-1}\kappa^{-2}\right)\|\boldsymbol{a}_l\|_2$, hence such $\boldsymbol{q}$ is one local minimum near $\boldsymbol{a}_l$.

**Lemma 3.2.** *Suppose $\boldsymbol{q}$ is a stationary point in $\mathcal{R}_{C_\star}$ with $C_\star \geq 10$, and $\boldsymbol{\zeta} = \boldsymbol{A}^T\boldsymbol{q}$ has only one entry $\zeta_l$ of magnitude no smaller than $2\mu\,\|\zeta\|_3^3 / \|\zeta\|_4^4$, then $\boldsymbol{q}$ is a local minimum near $\boldsymbol{a}_l$ such that $|\langle \boldsymbol{q}, \mathcal{P}_\mathbb{S}\,[\boldsymbol{a}_l]\rangle| > 1 - 2c_\star\kappa^{-2}$ with $c_\star = 1/C_\star$.*

**Saddle Points** If $\boldsymbol{q}$ is a stationary point in $\mathcal{R}_{C_\star}$ with $C_\star \geq 10$, and $\boldsymbol{\zeta}$ has more than one nontrivial entry, then the Riemannian Hessian $\mathrm{Hess}\,\varphi\,(\boldsymbol{q})$ has negative eigenvalue(s) and hence $\boldsymbol{q}$ is a saddle point. Especially, denoting any two nontrivial entries of $\boldsymbol{\zeta}$ with $\zeta_l$ and $\zeta_{l'}$, then there exists a negative curvature in the span of $\boldsymbol{a}_l$ and $\boldsymbol{a}_{l'}$.

**Lemma 3.3.** *Suppose $\boldsymbol{q}$ is a stationary point in $\mathcal{R}_{C_\star}$ with $C_\star \geq 10$, and $\boldsymbol{\zeta} = \boldsymbol{A}^T\boldsymbol{q}$ has at least two entries $\zeta_l$ and $\zeta_{l'}$ with magnitude larger than $2\mu\,\|\boldsymbol{\zeta}\|_3^3 / \|\boldsymbol{\zeta}\|_4^4$, then the Riemannian Hessian at $\boldsymbol{q}$ has negative eigenvalue(s) and $\boldsymbol{q}$ is a saddle point.*

## 3.3 Finite Sample Concentration

We argue that the critical points of the finite sample objective function $\psi(\boldsymbol{q})$ are similar to those of the asymptotic objective function $\varphi(\boldsymbol{q})$:

**Critical points are close.** The Riemannian gradient concentrates, such that there is a bijection between critical points $\boldsymbol{q}_{\mathrm{pop}}$ of $\varphi$ and critical points $\boldsymbol{q}_{\mathrm{fs}}$ of $\psi$, with $\|\boldsymbol{q}_{\mathrm{pop}} - \boldsymbol{q}_{\mathrm{fs}}\|_2$ small.

**Curvature is preserved.** The Riemannian Hessian concentrates, such that $\mathrm{Hess}[\psi](\boldsymbol{q}_{\mathrm{fs}})$ has a negative eigenvalue if and only if $\mathrm{Hess}[\varphi](\boldsymbol{q}_{\mathrm{pop}})$ has a negative eigenvalue, and $\mathrm{Hess}[\psi](\boldsymbol{q}_{\mathrm{fs}})$ is positive definite if and only if $\mathrm{Hess}[\varphi](\boldsymbol{q}_{\mathrm{pop}})$ is positive definite.

This implies that every local minimizer of the finite sample objective function is close to a preconditioned shift-truncation. While conceptually straightforward, the proofs of these

properties are somewhat involved, due to the presence of the preconditioner $(\boldsymbol{Y}\boldsymbol{Y}^T)^{-1/2}$. We give rigorous versions of all of the above statements, and a complete proof, in the supplementary appendix.

## 4   Experiments

**Properties of a Random Kernel.**   In our main result, the sparsity rate $\theta$ depends on the condition number $\kappa$ and induced column coherence $\mu$. Figure 3 plots the average values (over 100 independent simulations) of $\kappa$ and $\mu$ for generic unit kernels of varying dimension $k = 10, 20, \cdots, 1000$.

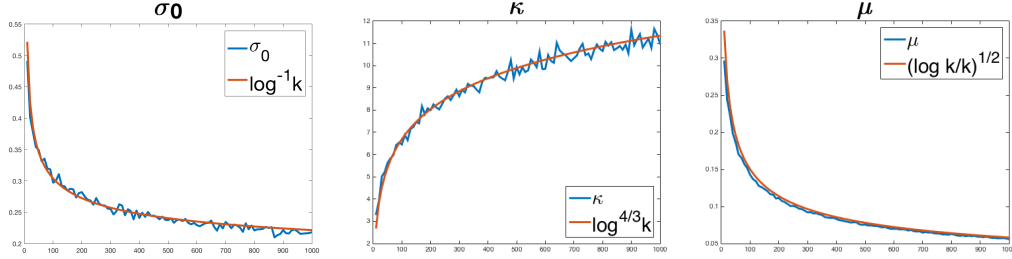

**Figure 3: Coherence of random kernels.** Average of $\sigma_{\min}$ (left), $\kappa$ (middle), and $\mu$ (right) over 100 independent trials, for varying kernel length $k$.

These simulations suggest the following estimates:

$$\sigma_{\min} \sim \log^{-1}(k), \quad \kappa \sim \log^{4/3} k, \quad \mu \sim \sqrt{\log(k)/k}. \tag{27}$$

Hence, reliable recovery of the shift truncation of a generic kernel can be guaranteed even when the sparse signal is relatively dense ($\theta \sim k^{-2/3}$). On the other hand, if the convolution kernel $\boldsymbol{a}_0$ is lowpass, then $\sigma_{\min}$ decreases, and $\kappa$, $\mu$ increase, then more observations $m$ and smaller sparsity level $\theta$ is required for the proposed algorithm to perform as desired.

**Recovery Error of the Proposed Algorithm**   We present the performance of Algorithm 1 under varying settings. We define the recover error as $\mathrm{err} = 1 - \max_\tau |\langle \bar{\boldsymbol{a}}, \mathcal{P}_\mathbb{S}[\boldsymbol{\iota}_k^* s_\tau [\widetilde{\boldsymbol{a}_0}]]\rangle|$, and calculate the average error from 50 independent experiments. The left figure plots the average error when we fix the kernel size $k = 50$, and vary the dimension $m$ and the sparsity $\theta$ of $\boldsymbol{x}_0$.[14] The right figure plots the average error when we vary the dimensions $k, m$ of both convolution signals, and set the sparsity as $\theta = k^{-2/3}$.

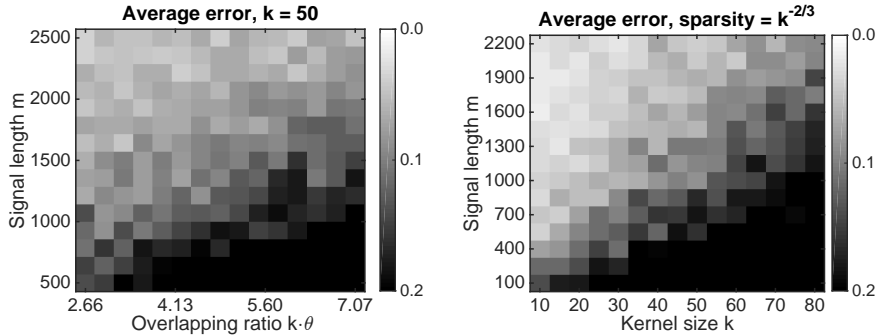

**Figure 4: Recovery Error** of the Shift Truncated Kernel by Algorithm 1.

**Acknowledgement**   The authors gratefully acknowledge support from NSF 1343282, NSF CCF 1527809, NSF CCF 1740833, and NSF IIS 1546411.

## Footnotes

[1]A number of works [7, 8, 9, 10, 11] have developed provable methods for blind deconvolution under the assumption that $\boldsymbol{a}_0$ and $\boldsymbol{x}_0$ belong to random subspaces, or are sparse in random dictionaries. These random models exhibit simpler geometry than the short-and-sparse model. Because our target signal is sparse in the standard basis, the aforementioned results are not applicable in our setting.

[2]Denote $\boldsymbol{y} = [y_1, y_2, \cdots, y_{m-1}, y_m]^T$, then its reversal $\check{\boldsymbol{y}} = [y_1, y_m, y_{m-1}, \cdots, y_2]^T$ with $y_1$ not moved.

[3]In this paper, we refer a kernel sampled following a uniform distribution over the sphere as a generic kernel on the sphere.

[4]This equivalently says there could be as many as $\mathcal{O}(k^{1/3})$ shifts of the kernel in a $k$-length window of the observation.

[5][16] proposes to solve the short-and-sparse blind deconvolution problem with a two phase algorithm which first recovers a shift truncation, and then recovers the ground truth kernel with an annealing algorithm. We present additional experimental results on the recovery of the ground truth in the supplementary material.

[6][19] considers the *multi-channel* blind deconvolution problem, where many independent observations $\boldsymbol{y}_p = \boldsymbol{a}_0 * \boldsymbol{x}_p$ are available. [19] shows how to formulate this problem as searching for a sparse vector in a linear subspace. Our approach is also inspired by the idea of looking for a sparse/spiky vector in a subspace. However, it pertains to a different problem, in which only a single observation is available. The short and sparse problem exhibits a more complicated optimization landscape, due to the signed shift ambiguity.

[7]For a generic kernel $a_0$, the shift-coherence is bounded by $|\langle a_0, s_\tau[a_0]\rangle| \approx 1/\sqrt{k}$ for any shift $\tau$.

[8]In particular, under a Bernoulli-Gaussian model, for each $j$, $\mathbb{E}[z_j^2] = \theta \sum_{i\neq 0} \langle a_0, s_i[a_0]\rangle^2$.

[9]In comparison, the classical choice $\|\cdot\|_\star = \|\cdot\|_1$ is a strict sparsity penalty that essentially encourages all small entries to be 0.

[10]As $\mathbb{E}_{x_0 \sim \text{i.i.d. BG}(\theta)}[YY^T] = \mathbb{E}_{x_0 \sim \text{i.i.d. BG}(\theta)}[A_0 X_0 X_0^T A_0^T] = \theta m A_0 A_0^T$.

[11]Please refer to Section 3 for more arguments.

[12]We call any $\zeta_l$ with magnitude no smaller than $2\mu \|\boldsymbol{\zeta}\|_3^3 / \|\boldsymbol{\zeta}\|_4^4$ to be nontrivial and defer technical reasonings to later sections.

[13]As $\mathbb{E}_{\boldsymbol{x}_0 \sim_{\text{i.i.d.}} \text{BG}(\theta)}[\boldsymbol{Y}\boldsymbol{Y}^T] = \theta m \boldsymbol{A}_0 \boldsymbol{A}_0^T$.

[14]Note that the $x$-axis is indexed with overlapping ratio $k \cdot \theta$, which indicates how many copies of $\boldsymbol{a}_0$ present in a $k$-length window of $\boldsymbol{y}$ on average.

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
