[Supplementary Material · bd_nips_supp.pdf]

# Supplementary Material for NIPS Submission 1169 Structured Local Optima in Sparse Blind Deconvolution

Yuqian Zhang, Han-Wen Kuo, John Wright

January 8, 2019

Section 1 contains additional experiment results of recovery accuracy of the ground truth kernel (rather than a shift truncation). Section 2 contains some basic lemmas for quantities used repeatedly; Section 6 presents the proofs of the main theorem and corollary of this paper. Section 4 and Section 5 proves for lemmas around the initialization point $q_{\text{init}}$ and the preconditioning term $Y^T Y$ (or $A_0^T A_0$) respectively. Finite sample concentration for the Riemannian gradient and Hessian are presented in Section 7 and Section 8 respectively.

## 1 Experiments

**Recovery Accuracy of the Ground Truth Kernel** In this section, we provide experiment results for the recovery of the ground truth kernel obtained by the annealing algorithm proposed in [ZLK+17]. The annealing algorithm recovers the ground truth kernel by minimizing the Lasso cost, initialized at the zero-padded shift truncated kernel rendered from Algorithm 1. The recovery accuracy presented in Figure 2 is measured as $err = \min_\tau \left\| \bar{a}^{(+)} \pm s_\tau[\widetilde{a_0}] \right\|_2$. Here, $\bar{a}^{(+)}$ denote the local minimum in the lifted optimization space.

**Figure 1: Recovery Error of the Ground Truth Kernel** with Algorithm 1 finding a shift truncated kernel and the annealing Lasso problem recovering the ground truth kernel.

For comparison, we also present experiment results of the algorithm proposed by [ZLK+17], which is composed of solving two Lasso minimization problems over the original kernel sphere and lifted kernel sphere respectively.

In terms of the recovery accuracy of the ground truth kernel, Algorithm 1 proposed in this paper achieves better recovery for sparser and longer observations, while the [ZLK+17] manifests slight advantages when the observations is limited. As the optimization landscape studied in [ZLK+17] varies with different choice

**Figure 2: Recovery Error of the Ground Truth Kernel** by minimizing the Lasso objective function recovering both the shift truncated kernel as well as the ground truth kernel.

of sparsity parameter $\lambda$, it is possible that experiment results for [ZLK$^+$17] could be improved. On the other hand, only empirical knowledge about the choice of $\lambda$ is available while there is little disciplined understanding. In contrast, Algorithm 1 does not depend on any parameter tuning and guarantees recovery once the working conditions are met.

# 2 Basics

**Lemma 2.1 (Expectation of the Approximate Objective Function)** *Assuming $\boldsymbol{x}_0 \sim_{\text{i.i.d.}} \text{BG}(\theta) \in \mathbb{R}^m$, then*

$$\mathbb{E}_{\boldsymbol{x}_0} \left[ \frac{1}{m} \left\| \boldsymbol{Y}^T \left( \boldsymbol{A}_0 \boldsymbol{A}_0^T \right)^{-1/2} \boldsymbol{q} \right\|_4^4 \right]$$
$$= 3\theta (1-\theta) \left\| \boldsymbol{A}^T \boldsymbol{q} \right\|_4^4 + 3\theta^2 \left\| \boldsymbol{A}^T \boldsymbol{q} \right\|_2^4 . \tag{2.1}$$

**Proof** Let $\boldsymbol{g} \in \mathbb{R}^{2k-1}$ be a standard random Gaussian vector and $\boldsymbol{P}_I$ be the projection operator onto Bernoulli vector $I \sim \text{Ber}(\theta)$ Then any column $\boldsymbol{x}_i \in \mathbb{R}^{2k-1}$ of $\boldsymbol{X}_0$ is equal in distribution to $\boldsymbol{x}_i = \boldsymbol{P}_I \boldsymbol{g}$ with $\boldsymbol{g} \sim_{\text{i.i.d.}} \mathcal{N}(0,1)$.

$$\mathbb{E}_{\boldsymbol{x}_0} \left[ \frac{1}{m} \left\| \boldsymbol{Y}^T \left( \boldsymbol{A}_0 \boldsymbol{A}_0^T \right)^{-1/2} \boldsymbol{q} \right\|_4^4 \right]$$

$$= \frac{1}{m} \mathbb{E}_I \mathbb{E}_{\boldsymbol{g}} \left\| \boldsymbol{q}^T \boldsymbol{A} \boldsymbol{X}_0 \right\|_4^4 \tag{2.2}$$

$$= \mathbb{E}_I \mathbb{E}_{\boldsymbol{g}} \left\| \boldsymbol{q}^T \boldsymbol{A} \boldsymbol{x}_i \right\|_4^4 \tag{2.3}$$

$$= \mathbb{E}_I \mathbb{E}_{\boldsymbol{g}} \left( \boldsymbol{q}^T \boldsymbol{A} \boldsymbol{P}_I \boldsymbol{g} \right)^4 \tag{2.4}$$

$$= 3\mathbb{E}_I \left( \boldsymbol{q}^T \boldsymbol{A} \boldsymbol{P}_I \boldsymbol{A}^T \boldsymbol{q} \right)^2 \tag{2.5}$$

$$= 3\mathbb{E}_I \left( \sum_{i \in I} \langle \boldsymbol{a}_i, \boldsymbol{q} \rangle^4 + \sum_{\{i \neq j\} \in I} \langle \boldsymbol{a}_i, \boldsymbol{q} \rangle^2 \langle \boldsymbol{a}_j, \boldsymbol{q} \rangle^2 \right) \tag{2.6}$$

$$= 3\theta (1-\theta) \left\| \boldsymbol{A}^T \boldsymbol{q} \right\|_4^4 + 3\theta^2 \left\| \boldsymbol{A}^T \boldsymbol{q} \right\|_2^4 \tag{2.7}$$

∎

**Lemma 2.2 (Root Estimation for Cubic Gradient Function)** *Consider an equation of the form*

$$f(x) = x(\alpha - x^2) - \beta = 0, \tag{2.8}$$

*with $\alpha > 0$. Suppose that $\beta < \frac{1}{4}\alpha^{3/2}$. Then $f(x) = 0$ has three solutions, $x_1, x_2, x_3$ satisfying*

$$\max\left\{ |x_1 - \sqrt{\alpha}|, |x_2 + \sqrt{\alpha}|, |x_3| \right\} \leq \frac{2\beta}{\alpha}. \tag{2.9}$$

**Proof** Suppose first that $\beta > 0$. Then $f(0) < 0$. Moreover,

$$f\left(\frac{2\beta}{\alpha}\right) = 2\beta - 8\beta^3/\alpha^3 - \beta \tag{2.10}$$

$$= \beta\left(1 - 8\beta^2/\alpha^3\right) \tag{2.11}$$

$$> 0. \tag{2.12}$$

Hence, $f$ has at least one root in the interval $\left[0, \frac{2\beta}{\alpha}\right]$. Similarly, notice that $f(\sqrt{\alpha}) < 0$ and that

$$f\left(\sqrt{\alpha} - \frac{2\beta}{\alpha}\right)$$

$$= \alpha^{3/2} - 2\beta - \left(\sqrt{\alpha} - 2\beta/\alpha\right)^3 - \beta \tag{2.13}$$

$$= \alpha^{3/2} - 3\beta - \alpha^{3/2} + 6\beta - 12\beta^2/\alpha^{3/2} + 8\beta^3/\alpha^3 \tag{2.14}$$

$$= \beta\left(3 - \frac{12\beta}{\alpha^{3/2}} + \frac{8\beta^2}{\alpha^3}\right) \tag{2.15}$$

$$> 0. \tag{2.16}$$

Thus, there is at least one root in the interval $\left[\sqrt{\alpha} - \frac{2\beta}{\alpha}, \sqrt{\alpha}\right]$. Finally, note that $f(-\sqrt{\alpha}) < 0$, $\frac{df}{dx}(-\sqrt{\alpha}) = -2\alpha$, and $\frac{d^2 f}{dx^2}(x') = -3x'$ is positive for $x' \leq -\sqrt{\alpha}$. Hence, convexity gives that

$$f\left(-\sqrt{\alpha} - \frac{2\beta}{\alpha}\right)$$

$$\geq f(-\sqrt{\alpha}) + \frac{df}{dx}(-\sqrt{\alpha}) \times (-2\beta/\alpha) \tag{2.17}$$

$$= -\beta + (-2\alpha) \times (-2\beta/\alpha) \tag{2.18}$$

$$= 3\beta \quad > \quad 0. \tag{2.19}$$

Under this condition, there is at least one root in the interval, $[-\sqrt{\alpha} - 2\beta/\alpha, -\sqrt{\alpha}]$. These three intervals do not overlap, as long as $\frac{4\beta}{\alpha} < \sqrt{\alpha}$, or $\beta < \frac{1}{4}\alpha^{3/2}$.

In the case that $\beta \leq 0$, a symmetric argument applies. Thus there are exactly three solutions to equation (2.8) in the specified intervals. ∎

**Lemma 2.3** *Let $\boldsymbol{a}_l$ and $\boldsymbol{a}_{l'}$ be two nonzero vectors with inner product $\mu_{l,l'} \doteq \langle \boldsymbol{a}_l, \boldsymbol{a}_{l'} \rangle$. Then for any unit vector $\boldsymbol{v} \in \mathrm{span}\,(\boldsymbol{a}_l, \boldsymbol{a}_{l'})$,*

$$\left| \left\langle \frac{\boldsymbol{a}_l}{\|\boldsymbol{a}_l\|_2}, \boldsymbol{v} \right\rangle \right|^2 + \left| \left\langle \frac{\boldsymbol{a}_{l'}}{\|\boldsymbol{a}_{l'}\|_2}, \boldsymbol{v} \right\rangle \right|^2 \geq 1 - \frac{|\mu_{l,l'}|}{\|\boldsymbol{a}_l\|_2 \|\boldsymbol{a}_{l'}\|_2}. \tag{2.20}$$

**Proof** Let $\boldsymbol{u}$ and $\boldsymbol{u}^\perp$ be two orthogonal unit vectors, such that

$$\boldsymbol{a}_l = \|\boldsymbol{a}_l\|_2 \, \boldsymbol{u}, \tag{2.21}$$

$$\boldsymbol{a}_{l'} = \frac{\mu_{l,l'}}{\|\boldsymbol{a}_l\|_2} \boldsymbol{u} + \sqrt{\|\boldsymbol{a}_{l'}\|_2^2 - \frac{\mu_{l,l'}^2}{\|\boldsymbol{a}_l\|_2^2}} \, \boldsymbol{u}^\perp. \tag{2.22}$$

Suppose $\boldsymbol{v} = a\boldsymbol{u} + b\boldsymbol{u}^\perp$ with $a^2 + b^2 = 1$. Let $\mu_{\mathrm{rel}} = \frac{\mu_{l,l'}}{\|\boldsymbol{a}_l\|_2 \|\boldsymbol{a}_{l'}\|_2}$, then we can expand the quantity of interests as

$$
\left| \left\langle \frac{\boldsymbol{a}_l}{\|\boldsymbol{a}_l\|_2}, \boldsymbol{v} \right\rangle \right|^2 + \left| \left\langle \frac{\boldsymbol{a}_{l'}}{\|\boldsymbol{a}_{l'}\|_2}, \boldsymbol{v} \right\rangle \right|^2
$$

$$
= \left| \langle \boldsymbol{u}, a\boldsymbol{u} + b\boldsymbol{u}^\perp \rangle \right|^2
$$
$$
+ \left| \left\langle \mu_{\mathrm{rel}}\boldsymbol{u} + \sqrt{1 - \mu_{\mathrm{rel}}^2}\boldsymbol{u}^\perp, a\boldsymbol{u} + b\boldsymbol{u}^\perp \right\rangle \right|^2 \tag{2.23}
$$

$$
= a^2 + \left( a\mu_{\mathrm{rel}} + b\sqrt{1 - \mu_{\mathrm{rel}}^2} \right)^2 \tag{2.24}
$$

$$
= a^2 + b^2 + \left( a^2 - b^2 \right) \mu_{\mathrm{rel}}^2 + 2ab\mu_{\mathrm{rel}}\sqrt{1 - \mu_{\mathrm{rel}}^2} \tag{2.25}
$$

$$
= 1 + \left[ a^2 - b^2, 2ab \right] \left[ \mu_{\mathrm{rel}}^2, \mu_{\mathrm{rel}}\sqrt{1 - \mu_{\mathrm{rel}}^2} \right]^T \tag{2.26}
$$

Since $\left[ a^2 - b^2, 2ab \right]$ is a unit vector, then above equation is lower bounded by

$$
1 - \left\| \left[ \mu_{rel}^2, \mu_{rel}\sqrt{1 - \mu_{rel}^2} \right] \right\|_2
$$

$$
= 1 - |\mu_{rel}| \tag{2.27}
$$

$$
= 1 - \frac{|\mu_{l,l'}|}{\|\boldsymbol{a}_l\|_2 \|\boldsymbol{a}_{l'}\|_2} \tag{2.28}
$$

as claimed. ∎

**Lemma 2.4 (Nonzeros in a Bernoulli Vector)** *Let $\boldsymbol{v} \sim_{\mathrm{i.i.d.}} \mathrm{Ber}\left(\theta\right) \in \mathbb{R}^n$, then*

$$
\mathbb{P}\left[ \|\boldsymbol{v}\|_0 \geq (1 + t)\theta n \right] \leq 2\exp\left( -\frac{3t^2}{2t + 6}\theta n \right). \tag{2.29}
$$

**Proof** As $\|\boldsymbol{v}\|_0 = v_0 + \cdots + v_{n-1}$, and

$$
|v_i - \theta| \leq 1, \quad \mathbb{E}\left[ (v_i - \theta)^2 \right] = \theta(1 - \theta) \leq \theta \tag{2.30}
$$

with Bernstein's inequality, we obtain that

$$
\mathbb{P}\left[ \|\boldsymbol{v}\|_0 \geq (1 + t)\theta n \right]
$$
$$
\leq 2\exp\left( -\frac{t^2\theta^2 n^2}{2(\theta - \theta^2)n + \frac{2}{3}t\theta n} \right) \tag{2.31}
$$

$$
\leq 2\exp\left( -\frac{3t^2}{2t + 6}\theta n \right), \tag{2.32}
$$

as claimed. ∎

**Lemma 2.5 (Entry-wise Truncation of a Bernoulli Gaussian Vector)** *Suppose $\boldsymbol{x}_0 \sim_{\mathrm{i.i.d.}} \mathrm{BG}\left(\theta\right) \in \mathbb{R}^m$, then*

$$
\mathbb{P}\left[ \|\boldsymbol{x}_0\|_\infty > t \right] \leq 2\theta m e^{-t^2/2}. \tag{2.33}
$$

**Proof** A Bernoulli-Gaussian variable $x = \omega \cdot g$ satisfies

$$\mathbb{P}\left[|x| \geq t\right] = \theta \cdot \mathbb{P}\left[|g| \geq t\right] \leq 2\theta e^{-t^2/2}, \tag{2.34}$$

Taking a union bound over the $m$ entries of $\boldsymbol{x}_0$, we obtain

$$\mathbb{P}\left[\|\boldsymbol{x}_0\|_\infty > t\right] \leq m\mathbb{P}\left[|x| > t\right] \tag{2.35}$$

$$\leq 2\theta m e^{-t^2/2}, \tag{2.36}$$

as claimed. ∎

**Lemma 2.6 (Operator Norm of a Bernoulli Gaussian Circulant Matrix)** *Let $\boldsymbol{C}_{\boldsymbol{x}_0} \in \mathbb{R}^{m \times m}$ be the circulant matrix generated from $\boldsymbol{x}_0 \sim_{\text{i.i.d.}} \text{BG}(\theta) \in \mathbb{R}^m$, then*

$$\mathbb{P}\left[\|\boldsymbol{C}_{\boldsymbol{x}_0}\|_2 \geq t\right] \leq 2m \exp\left(-\frac{t^2}{2\theta m + 2t}\right). \tag{2.37}$$

**Proof** The operator norm of a circulant matrix is

$$\|\boldsymbol{C}_{\boldsymbol{x}_0}\|_2 = \max_l |\langle \boldsymbol{x}_0, \boldsymbol{w}_l \rangle|, \tag{2.38}$$

where $\boldsymbol{w}_l$ is the $l$-th (discrete) Fourier basis vector

$$\boldsymbol{w}_l = \left[1, \ e^{l\frac{2\pi j}{m}}, \cdots, \ e^{l(m-1)\frac{2\pi j}{m}}\right]^T, \quad l = 0, \cdots, m-1, \tag{2.39}$$

and $j$ is the imaginary unit. With moment control Bernstein inequality, we obtain

$$\mathbb{P}\left[|\langle \boldsymbol{x}_0, \boldsymbol{w}_l \rangle| \geq t\right] \leq 2\exp\left(-\frac{t^2}{2\theta \|\boldsymbol{w}_l\|_2^2 + 2\|\boldsymbol{w}_l\|_\infty t}\right)$$

$$\leq 2\exp\left(-\frac{t^2}{2\theta m + 2t}\right) \tag{2.40}$$

together with the union bound,

$$\mathbb{P}\left[\|\boldsymbol{C}_{\boldsymbol{x}_0}\|_2 \geq t\right] \leq m\mathbb{P}\left[|\langle \boldsymbol{x}_0, \boldsymbol{w}_l \rangle| \geq t\right] \tag{2.41}$$

$$\leq 2m \exp\left(-\frac{t^2}{2\theta m + 2t}\right), \tag{2.42}$$

as claimed. ∎

**Lemma 2.7 (Norms of $\eta$ and $\bar{\eta}$)** *Suppose $\delta = \left\|\frac{1}{\theta m} \boldsymbol{X}_0 \boldsymbol{X}_0^T - \boldsymbol{I}\right\|_2 \leq 1/\left(2\kappa^2\right)$, then vectors $\boldsymbol{\eta} = \boldsymbol{Y}^T \left(\boldsymbol{Y}\boldsymbol{Y}^T\right)^{-1/2} \boldsymbol{q}$ and $\bar{\boldsymbol{\eta}} = \boldsymbol{Y}^T \left(\theta m \boldsymbol{A}_0 \boldsymbol{A}_0^T\right)^{-1/2} \boldsymbol{q}$ satisfy*

$$\|\boldsymbol{\eta}\|_\infty \leq \left(1 + \frac{4\kappa^3\delta}{\sigma_{\min}}\right) \left(\frac{2k}{\theta m}\right)^{1/2} \|\boldsymbol{x}_0\|_\infty, \tag{2.43}$$

$$\|\bar{\boldsymbol{\eta}}\|_\infty \leq \left(\frac{2k}{\theta m}\right)^{1/2} \|\boldsymbol{x}_0\|_\infty, \tag{2.44}$$

$$\|\boldsymbol{\eta}\|_6^6 \leq \left(1 + \frac{4\kappa^3\delta}{\sigma_{\min}}\right)^4 \frac{4k^2}{\theta^2 m^2} \|\boldsymbol{x}_0\|_\infty^4, \tag{2.45}$$

$$\|\bar{\boldsymbol{\eta}}\|_2 \leq 1 + \delta/2, \tag{2.46}$$

$$\|\boldsymbol{\eta} - \bar{\boldsymbol{\eta}}\|_\infty \leq \frac{4\kappa^3\delta}{\sigma_{\min}} \left(\frac{2k}{\theta m}\right)^{1/2} \|\boldsymbol{x}_0\|_\infty, \tag{2.47}$$

$$\|\boldsymbol{\eta} - \bar{\boldsymbol{\eta}}\|_2 \leq (1 + \delta/2) \frac{4\kappa^3\delta}{\sigma_{\min}}. \tag{2.48}$$

**Proof** Since $\delta = \left\| \frac{1}{\theta m} X_0 X_0^T - I \right\|_2$, then

$$\|X_0\|_2 \le (\theta m)^{1/2} \sqrt{1 + \delta} \tag{2.49}$$

$$\le (\theta m)^{1/2} (1 + \delta/2). \tag{2.50}$$

As $\boldsymbol{\eta} = Y^T \left( YY^T \right)^{-1/2} q = X_0^T A_0^T \left( YY^T \right)^{-1/2} q$, together with Lemma 5.3:

$$\left\| A_0^T \left( YY^T \right)^{-1/2} q \right\|_\infty$$

$$\le \left\| A_0^T \left( YY^T \right)^{-1/2} q \right\|_2 \tag{2.51}$$

$$\le \left\| A_0^T \left( \left( YY^T \right)^{-1/2} - \left( \theta m A_0 A_0^T \right)^{-1/2} \right) q \right\|_2$$

$$+ \left\| A_0^T \left( \theta m A_0 A_0^T \right)^{-1/2} q \right\|_2 \tag{2.52}$$

$$\le (\theta m)^{-1/2} \frac{4\kappa^3 \delta}{\sigma_{\min}} \|q\|_2 + (\theta m)^{-1/2} \left\| A^T q \right\|_2 \tag{2.53}$$

$$\le (\theta m)^{-1/2} \left( 1 + \frac{4\kappa^3 \delta}{\sigma_{\min}} \right) \tag{2.54}$$

**Norms of $\boldsymbol{\eta}$.** Since $\|X_0 e_l\|_2 \le \sqrt{2k-1} \|X_0 e_l\|_\infty$, we have

$$\|\boldsymbol{\eta}\|_\infty = \max_{l \in [1, \cdots, m]} \left\langle X_0 e_l, A_0^T \left( YY^T \right)^{-1/2} q \right\rangle \tag{2.55}$$

$$\le \max_l \|X_0 e_l\|_2 \left\| A_0^T \left( YY^T \right)^{-1/2} q \right\|_2 \tag{2.56}$$

$$\le \sqrt{2k} \|x_0\|_\infty \cdot (\theta m)^{-1/2} \left( 1 + \frac{4\kappa^3 \delta}{\sigma_{\min}} \right). \tag{2.57}$$

At the same time, plugging in $\|\boldsymbol{\eta}\|_2 = 1$, we have

$$\|\boldsymbol{\eta}\|_6^6 \le \|\boldsymbol{\eta}\|_2^2 \|\boldsymbol{\eta}\|_\infty^4 \le \left( 1 + \frac{4\kappa^3 \delta}{\sigma_{\min}} \right)^4 \frac{4k^2}{\theta^2 m^2} \|x_0\|_\infty^4. \tag{2.58}$$

**Norms of $\bar{\boldsymbol{\eta}}$.** Here, $\bar{\boldsymbol{\eta}} = Y^T \left( \theta m A_0 A_0^T \right)^{-1/2} q = X_0^T A_0^T \left( \theta m A_0 A_0^T \right)^{-1/2} q$ with

$$\left\| A_0^T \left( \theta m A_0 A_0^T \right)^{-1/2} q \right\|_\infty$$

$$\le \left\| A_0^T \left( \theta m A_0 A_0^T \right)^{-1/2} q \right\|_2 \tag{2.59}$$

$$= (\theta m)^{-1/2}, \tag{2.60}$$

therefore

$$\|\bar{\boldsymbol{\eta}}\|_\infty \le \max_l \|X_0 e_l\|_2 \left\| A_0 \left( \theta m A_0 A_0^T \right)^{-1/2} q \right\|_2$$

$$\le \left( \frac{2k}{\theta m} \right)^{1/2} \|x_0\|_\infty, \tag{2.61}$$

$$\|\bar{\boldsymbol{\eta}}\|_2 \le \left\| X_0^T \right\|_2 \left\| A_0 \left( \theta m A_0 A_0^T \right)^{-1/2} q \right\|_2$$

$$\le 1 + \delta/2. \tag{2.62}$$

**Norms of $\boldsymbol{\eta} - \bar{\boldsymbol{\eta}}$.** With similar reasoning, we can obtain

$$
\begin{aligned}
&\|\boldsymbol{\eta} - \bar{\boldsymbol{\eta}}\|_\infty \\
&= \left\| \boldsymbol{Y}^T \left( \boldsymbol{Y}\boldsymbol{Y}^T \right)^{-1/2} \boldsymbol{q} - \boldsymbol{Y}^T \left( \theta m \boldsymbol{A}_0 \boldsymbol{A}_0^T \right)^{-1/2} \boldsymbol{q} \right\|_\infty \\
&\leq \max_{l \in [1, \cdots, m]} \|\boldsymbol{X}_0 \boldsymbol{e}_l\|_2 \, (\theta m)^{-1/2} \times \\
&\quad \left\| \boldsymbol{A}_0^T \left( \frac{1}{\theta m} \boldsymbol{Y}\boldsymbol{Y}^T \right)^{-1/2} - \boldsymbol{A}_0^T \left( \boldsymbol{A}_0 \boldsymbol{A}_0^T \right)^{-1/2} \right\|_2 \quad\quad (2.63) \\
&\leq \frac{4\kappa^3 \delta}{\sigma_{\min}} \left( \frac{2k}{\theta m} \right)^{1/2} \|\boldsymbol{x}_0\|_\infty, \quad\quad (2.64)
\end{aligned}
$$

and

$$
\begin{aligned}
&\|\boldsymbol{\eta} - \bar{\boldsymbol{\eta}}\|_2 \\
&\leq \|\boldsymbol{X}_0\|_2 \, (\theta m)^{-1/2} \|\boldsymbol{q}\|_2 \times \\
&\quad \left\| \boldsymbol{A}_0^T \left( \frac{1}{\theta m} \boldsymbol{Y}\boldsymbol{Y}^T \right)^{-1/2} - \boldsymbol{A}_0^T \left( \boldsymbol{A}_0 \boldsymbol{A}_0^T \right)^{-1/2} \right\|_2 \quad\quad (2.65) \\
&\leq (\theta m)^{-1/2} \frac{4\kappa^3 \delta}{\sigma_{\min}} \|\boldsymbol{X}_0\|_2 \quad\quad (2.66) \\
&\leq (1 + \delta/2) \frac{4\kappa^3 \delta}{\sigma_{\min}}, \quad\quad (2.67)
\end{aligned}
$$

completing the proof. ∎

# 3 Proof for Geometry around the Stationary Points

## 3.1 Local Minima

**Lemma 3.1** *Suppose $\bar{\boldsymbol{q}}$ is a stationary point in $\mathcal{R}_{C_\star}$ ($C_\star \geq 10$), and its corresponding $\boldsymbol{\zeta}$ has only one entry $\zeta_l$ with nontrivial magnitude, then this stationary point is a local minimum near $\boldsymbol{a}_l$ that $|\langle \bar{\boldsymbol{q}}, \mathcal{P}_\mathbb{S} [\boldsymbol{a}_l] \rangle| > 1 - 2c_\star \kappa^{-2}$ with $c_\star = 1/C_\star$.*

**Proof** Suppose $\boldsymbol{\zeta}$ has only one big entry $\zeta_l$, and other entries are bounded by $2\beta/\alpha$

$$
\|\boldsymbol{\zeta}\|_4^4 = \zeta_l^4 + \sum_{j \neq l} \zeta_j^4 \leq \zeta_l^4 + \max_{j \neq l} \zeta_j^2 \cdot \sum_{j \neq i} \zeta_j^2 \leq \zeta_l^4 + \frac{4\mu^2 \|\boldsymbol{\zeta}\|_3^6}{\|\boldsymbol{\zeta}\|_4^8}, \quad\quad (3.1)
$$

with $\|\boldsymbol{\zeta}\|_4^6 \geq C_\star \mu \kappa^2 \|\boldsymbol{\zeta}\|_3^3$, and for simplicity let $c_\star = 1/C_\star$, we have

$$
\zeta_l^4 \geq \|\boldsymbol{\zeta}\|_4^4 - \frac{4\mu^2 \|\boldsymbol{\zeta}\|_3^6}{\|\boldsymbol{\zeta}\|_4^8} \geq \left( 1 - 4c_\star^2 \kappa^{-4} \right) \|\boldsymbol{\zeta}\|_4^4. \quad\quad (3.2)
$$

On the other hand, we also have

$$
\zeta_l^2 \leq (\sqrt{\alpha_l} + \frac{2\beta_l}{\alpha_l})^2 \leq \frac{\|\boldsymbol{\zeta}\|_4^4}{\|\boldsymbol{a}_i\|_2^2} + \frac{4\mu \|\boldsymbol{\zeta}\|_3^3}{\|\boldsymbol{a}_i\|_2 \|\boldsymbol{\zeta}\|_4^2} + \frac{4\mu^2 \|\boldsymbol{\zeta}\|_3^6}{\|\boldsymbol{\zeta}\|_4^8} \leq \frac{\|\boldsymbol{\zeta}\|_4^4}{\|\boldsymbol{a}_i\|_2^2} \left( 1 + 4c_\star \kappa^{-2} + 4c_\star^2 \kappa^{-4} \right). \quad\quad (3.3)
$$

Combining above two inequalities, we have

$$\zeta_l^2 \le \frac{1 + 4c_\star \kappa^{-2} + 4c_\star^2 \kappa^{-4}}{1 - 4c_\star^2 \kappa^{-4}} \frac{\zeta_l^4}{\|\boldsymbol{a}_i\|_2^2}, \tag{3.4}$$

thus the local minimum $\bar{\boldsymbol{q}}$ is close to $\boldsymbol{a}_l$:

$$\frac{|\langle \bar{\boldsymbol{q}}, \boldsymbol{a}_l \rangle|}{\|\boldsymbol{a}_l\|_2} \ge \frac{\sqrt{1 - 4c_\star^2 \kappa^{-4}}}{1 + 2c_\star \kappa^{-2}} \ge 1 - 2c_\star \kappa^{-2}. \tag{3.5}$$

Next, we need to verify that the Riemannian Hessian at $\bar{\boldsymbol{q}}$ is definite positive, recall that

$$\operatorname{Hess} \varphi\left(\bar{\boldsymbol{q}}\right) = -\boldsymbol{P}_{\bar{\boldsymbol{q}}^\perp} \left[ 3\boldsymbol{A} \operatorname{diag}(\boldsymbol{\zeta}^{\circ 2})\boldsymbol{A}^T - \|\boldsymbol{\zeta}\|_4^4 \, \boldsymbol{I} \right] \boldsymbol{P}_{\bar{\boldsymbol{q}}^\perp}. \tag{3.6}$$

Let $\boldsymbol{v}$ be a unit vector such that $\boldsymbol{v} \perp \bar{\boldsymbol{q}}$, then

$$\begin{align}
\boldsymbol{v}^T \operatorname{Hess} \varphi\left(\bar{\boldsymbol{q}}\right) \boldsymbol{v} &= -\boldsymbol{v}^T \left( 3\boldsymbol{A} \operatorname{diag}(\boldsymbol{\zeta}^{\circ 2})\boldsymbol{A}^T - \|\boldsymbol{\zeta}\|_4^4 \, \boldsymbol{I} \right) \boldsymbol{v} \tag{3.7} \\
&= \|\boldsymbol{\zeta}\|_4^4 - 3\boldsymbol{v}^T \boldsymbol{A} \operatorname{diag}(\boldsymbol{\zeta}^{\circ 2})\boldsymbol{A}^T \boldsymbol{v} \tag{3.8} \\
&= \|\boldsymbol{\zeta}\|_4^4 - 3 \langle \boldsymbol{a}_l, \boldsymbol{v} \rangle^2 \zeta_l^2 - 3 \sum_{i \ne l} \langle \boldsymbol{a}_i, \boldsymbol{v} \rangle^2 \zeta_i^2 \tag{3.9} \\
&\ge \|\boldsymbol{\zeta}\|_4^4 - 3 \langle \boldsymbol{a}_l, \boldsymbol{v} \rangle^2 \zeta_l^2 - 3 \max_{i \ne l} \zeta_i^2. \tag{3.10}
\end{align}$$

The last inequality is due to $\sum_{i \ne l} \langle \boldsymbol{a}_i, \boldsymbol{v} \rangle^2 \le \|\boldsymbol{A}^T \boldsymbol{v}\|_2^2 = 1$. Since $\boldsymbol{v} \perp \bar{\boldsymbol{q}}$ and $\zeta_l$ is the only entry with nontrivial magnitude, then derive from eq. (3.5):

$$\langle \boldsymbol{a}_l, \boldsymbol{v} \rangle^2 \zeta_l^2 \le 2c_\star \|\boldsymbol{a}_l\|_2^2 \left( \sqrt{\alpha_l} + \frac{2\beta_l}{\alpha_l} \right)^2 \le 2c_\star \|\boldsymbol{a}_l\|_2^2 \cdot (1 + 2c_\star)^2 \alpha_l \le 2c_\star \left(1 + 2c_\star^2\right)^2 \|\boldsymbol{\zeta}\|_4^4, \tag{3.11}$$

$$\max_{i \ne l} \zeta_i^2 \le \frac{4\beta^2}{\alpha^2} \le \frac{4\mu^2 \|\boldsymbol{\zeta}\|_3^6}{\|\boldsymbol{\zeta}\|_4^8} \le \frac{4c_\star^2 \|\boldsymbol{\zeta}\|_4^{12}}{\|\boldsymbol{\zeta}\|_4^8} \le 4c_\star^2 \|\boldsymbol{\zeta}\|_4^4. \tag{3.12}$$

Hence, the inequality $\boldsymbol{v}^T \operatorname{Hess} \varphi\left(\bar{\boldsymbol{q}}\right) \boldsymbol{v} \ge \left(1 - 6c_\star - 36c_\star^2 - 24c_\star^3\right) \|\boldsymbol{\zeta}\|_4^4$ holds for any choice of $\boldsymbol{v} \perp \bar{\boldsymbol{q}}$, thus when $C_\star \ge 10$ this implies positive curvature along any tangent direction at such stationary point $\bar{\boldsymbol{q}}$. ∎

## 3.2 Saddle Points

**Lemma 3.2** *Suppose $\bar{\boldsymbol{q}}$ is a stationary point in $\mathcal{R}_{C_\star}$, and its corresponding $\boldsymbol{\zeta}$ has only at least two entries $\zeta_l$ and $\zeta_{l'}$ with nontrivial magnitude, then the Riemannian Hessian at $\bar{\boldsymbol{q}}$ has negative eigenvalues.*

**Proof** Suppose $\boldsymbol{\zeta}$ has at least two big entries $\zeta_l$ and $\zeta_{l'}$ that

$$\begin{align}
\zeta_l^2 &\ge \left( \sqrt{\alpha_l} - \frac{2\beta_l}{\alpha_l} \right)^2 \tag{3.13} \\
&\ge \frac{\|\boldsymbol{\zeta}\|_4^4}{\|\boldsymbol{a}_l\|_2^2} - \frac{4\mu \|\boldsymbol{\zeta}\|_3^3}{\|\boldsymbol{\zeta}\|_4^2 \|\boldsymbol{a}_l\|_2} + \frac{4\mu^2 \|\boldsymbol{\zeta}\|_3^6}{\|\boldsymbol{\zeta}\|_4^8} \tag{3.14} \\
&> \frac{\|\boldsymbol{\zeta}\|_4^4}{\|\boldsymbol{a}_l\|_2^2} - \frac{4\mu \|\boldsymbol{\zeta}\|_3^3}{\|\boldsymbol{\zeta}\|_4^2 \|\boldsymbol{a}_l\|_2}. \tag{3.15}
\end{align}$$

Since the nontrivial entry $\zeta_l = \langle \boldsymbol{a}_l, \boldsymbol{q} \rangle$, and again let $c_\star = 1/C_\star$, it is easy to show that the norm of $\boldsymbol{a}_l$ is sufficiently large:

$$\|\boldsymbol{a}_l\|_2^2 \geq \zeta_l^2 \geq \left( \sqrt{\alpha_l} - \frac{2\beta_l}{\alpha_l} \right)^2 \geq (1 - 2c_\star)^2 \frac{\|\boldsymbol{\zeta}\|_4^4}{\|\boldsymbol{a}_l\|_2^2} \geq (1-\star)^2 \, C_\star^{2/3} \frac{\mu^{2/3} \|\boldsymbol{\zeta}\|_3^2}{\|\boldsymbol{a}_l\|_2^2}, \tag{3.16}$$

or

$$\|\boldsymbol{a}_l\|_2 \geq (1 - c_\star)^{1/2} \, C_\star^{1/6} \mu^{1/6} \|\boldsymbol{\zeta}\|_3^{1/2}. \tag{3.17}$$

Similar result holds for $\|\boldsymbol{a}_{l'}\|_2$, therefore

$$\frac{\mu}{\|\boldsymbol{a}_l\|_2 \|\boldsymbol{a}_{l'}\|_2} \leq \frac{\mu^{2/3}}{C_\star^{1/3} \|\boldsymbol{\zeta}\|_3} \leq \frac{C_\star^{-2/3} \|\boldsymbol{\zeta}\|_4^4}{C_\star^{1/3} \|\boldsymbol{\zeta}\|_3^3} \leq c_\star. \tag{3.18}$$

Now we are ready to show there exists a unit vector $\boldsymbol{v}$ such that $\boldsymbol{v} \in span(\boldsymbol{a}_l, \boldsymbol{a}_{l'})$ and $\boldsymbol{v} \perp \boldsymbol{q}$, and the Hessian has negative curvature along such $\boldsymbol{v}$:

$$\begin{aligned}
\boldsymbol{v}^T \operatorname{Hess} \varphi(\boldsymbol{q}) \boldsymbol{v} &= -3\boldsymbol{v}^T \boldsymbol{A} \operatorname{diag}(\boldsymbol{\zeta}^2) \boldsymbol{A}^T \boldsymbol{v} + \|\boldsymbol{\zeta}\|_4^4 & (3.19)\\
&\leq -3\boldsymbol{v}^T \left( \boldsymbol{a}_l \zeta_l^2 \boldsymbol{a}_l^T + \boldsymbol{a}_{l'} \zeta_{l'}^2 \boldsymbol{a}_{l'}^T \right) \boldsymbol{v} + \|\boldsymbol{\zeta}\|_4^4 & (3.20)\\
&< -3 \left( \left| \left\langle \frac{\boldsymbol{a}_l}{\|\boldsymbol{a}_l\|_2}, \boldsymbol{v} \right\rangle \right|^2 + \left| \left\langle \frac{\boldsymbol{a}_{l'}}{\|\boldsymbol{a}_{l'}\|_2}, \boldsymbol{v} \right\rangle \right|^2 \right) \|\boldsymbol{\zeta}\|_4^4 & (3.21)\\
&\quad + \frac{4\mu \|\boldsymbol{\zeta}\|_3^3}{\|\boldsymbol{\zeta}\|_4^2} \left( \|\boldsymbol{a}_l\|_2 + \|\boldsymbol{a}_{l'}\|_2 \right) + \|\boldsymbol{\zeta}\|_4^4 & (3.22)\\
&< -3 \left( 1 - \frac{\mu}{\|\boldsymbol{a}_l\|_2 \|\boldsymbol{a}_{l'}\|_2} \right) \|\boldsymbol{\zeta}\|_4^4 + \frac{4\mu \|\boldsymbol{\zeta}\|_3^3}{\|\boldsymbol{\zeta}\|_4^2} \left( \|\boldsymbol{a}_l\|_2 + \|\boldsymbol{a}_{l'}\|_2 \right) + \|\boldsymbol{\zeta}\|_4^4 & (3.23)\\
&\leq (-2 + 11c_\star) \|\boldsymbol{\zeta}\|_4^4. & (3.24)
\end{aligned}$$

The last inequality is implied by Lemma 2.3 and is negative when $C_\star \geq 10$.

∎

## 4  Initialization

**Lemma 4.1** *Suppose $\boldsymbol{x}_0 \sim_{\text{i.i.d.}} \text{BG}(\theta) \in \mathbb{R}^m$. There exists a positive constant $C > 2560^4$ such that whenever*

$$m \geq C\theta^2 \sigma_{\min}^{-2} \kappa^6 k^3 \left( 1 + 36\mu^2 k \log k \right)^4 \log \left( \kappa k/\sigma_{\min} \right) \tag{4.1}$$

*and the sparsity rate*

$$\begin{aligned}
64k^{-1} \log k \leq \theta \leq \min \Big\{ &\tfrac{1}{48^2} \mu^{-2} k^{-1} \log^{-2} k, \\
&\left( \tfrac{1}{4} - \tfrac{640}{C^{1/4}} \right) \left( 3C_\star \mu \kappa^2 \right)^{-2/3} k^{-1} \left( 1 + 36\mu^2 k \log k \right)^{-2} \Big\},
\end{aligned} \tag{4.2}$$

*Then the initialization $\boldsymbol{q}_{\text{init}} = \mathcal{P}_{\mathbb{S}} \left[ \left( \boldsymbol{Y}\boldsymbol{Y}^T \right)^{-1/2} \boldsymbol{y}_i \right]$ satisfies*

$$\left\| \boldsymbol{A}^T \boldsymbol{q}_{\text{init}} \right\|_4^6 \geq 3C_\star \mu \kappa^2, \tag{4.3}$$

*namely $\boldsymbol{q}_{\text{init}} \in \hat{\mathcal{R}}_{3C_\star}$, with probability no smaller than $1 - k^{-1} - 8k^{-2} - 2\exp(-\theta k) - 48k^{-7} - 48m^{-5} - 24k \exp \left( -\frac{1}{144} \min \left\{ k, 3\sqrt{\theta m} \right\} \right)$.*

**Proof**
  Since

$$m \geq C \frac{\theta^2}{\sigma_{\min}^2} \kappa^6 k^3 \left(1 + 36\mu^2 k \log k\right)^4 \log\left(\kappa k / \sigma_{\min}\right) \tag{4.4}$$

with $C \geq 2560^4$, then from Lemma 5.1, then with probability no smaller than $1 - 2\exp\left(-\theta k\right) - 24k\exp\left(-\frac{1}{144}\min\left\{k, 3\sqrt{\theta m}\right\}\right) - 48k^{-7} - 48m^{-5}$, we have

$$\delta \doteq \left\| \frac{1}{\theta m} \boldsymbol{X}_0 \boldsymbol{X}_0^T - \boldsymbol{I} \right\|_2 \tag{4.5}$$

$$\leq 10\sqrt{k \log m / m} \tag{4.6}$$

$$\leq \frac{10\sigma_{\min}}{\theta\sigma_{\min}^{-1}\kappa^3 k \left(1 + 36\mu^2 k \log k\right)^2} \times$$
$$\sqrt{\frac{\log\left(\frac{C\kappa^6 k^3 (1 + 36\mu^2 k \log k)^4}{\sigma_{\min}^2} \log\left(\frac{\kappa k}{\sigma_{\min}}\right)\right)}{C \log\left(\kappa k / \sigma_{\min}\right)}} \tag{4.7}$$

$$\leq \frac{20\sigma_{\min}}{C^{1/4}\theta\kappa^3 k \left(1 + 36\mu^2 k \log k\right)^2} \tag{4.8}$$

obtains, and the last inequality holds when $C \geq 1000$ that

$$\log\left(37^4 C\right) \leq \log 2\sqrt{C}. \tag{4.9}$$

Therefore

$$C\sigma_{\min}^{-2}\kappa^6 k^3 \left(1 + 36\mu^2 k \log k\right)^4 \log\left(\kappa k / \sigma_{\min}\right)$$

$$\leq 37^4 C \left(\kappa k / \sigma_{\min}\right)^7 \log^5\left(\kappa k / \sigma_{\min}\right) \tag{4.10}$$

$$\leq 37^4 C \left(\kappa k / \sigma_{\min}\right)^{12} \tag{4.11}$$

or

$$\sqrt{\frac{\log\left(\frac{C\kappa^6 k^3 (1 + 36\mu^2 k \log k)^4}{\sigma_{\min}^2} \log\left(\frac{\kappa k}{\sigma_{\min}}\right)\right)}{C \log\left(\kappa k / \sigma_{\min}\right)}}$$

$$\leq \sqrt{\frac{\log\left(37^4 C\right) + 12\log\left(\kappa k / \sigma_{\min}\right)}{C \log\left(\kappa k / \sigma_{\min}\right)}} \tag{4.12}$$

$$\leq \sqrt{\frac{\log 2}{\sqrt{C} \log\left(\kappa k / \sigma_{\min}\right)} + \frac{12}{C}} \tag{4.13}$$

$$\leq \frac{2}{C^{1/4}} \qquad (k \geq 2, C \geq 16) \tag{4.14}$$

Moreover, $\kappa^2\delta \leq 1/2$ always holds provided

$$C \geq \left(\frac{40}{\theta k \left(1 + 36\mu^2 k \log k\right)^2}\right)^4. \tag{4.15}$$

Notice that because $\theta$ is lower bounded by $c \log k / k$, the right hand side is indeed bounded by an absolute constant.

Set $\boldsymbol{\zeta}_{\text{init}} = \boldsymbol{A}^T \boldsymbol{q}_{\text{init}}$ and $\hat{\boldsymbol{\zeta}}_{\text{init}} = \mathcal{P}_{\mathbb{S}} \left[ \boldsymbol{A}^T \boldsymbol{A} \boldsymbol{x}_i \right]$. Then using for any nonzero vectors $\boldsymbol{u}$ and $\boldsymbol{v}$,

$$\left\| \frac{\boldsymbol{u}}{\|\boldsymbol{u}\|_2} - \frac{\boldsymbol{v}}{\|\boldsymbol{v}\|_2} \right\|_2 \leq \frac{2}{\|\boldsymbol{v}\|_2} \|\boldsymbol{u} - \boldsymbol{v}\|_2 , \tag{4.16}$$

we have that

$$\left\| \boldsymbol{\zeta}_{\text{init}} - \hat{\boldsymbol{\zeta}}_{\text{init}} \right\|_2$$
$$= \left\| \boldsymbol{A}^T \mathcal{P}_{\mathbb{S}} \left[ \left( \boldsymbol{Y} \boldsymbol{Y}^T \right)^{-1/2} \boldsymbol{A}_0 \boldsymbol{x}_i \right] - \mathcal{P}_{\mathbb{S}} \left[ \boldsymbol{A}^T \boldsymbol{A} \boldsymbol{x}_i \right] \right\|_2 \tag{4.17}$$

$$= \left\| \frac{\boldsymbol{A}^T \left( \frac{1}{\theta m} \boldsymbol{Y} \boldsymbol{Y}^T \right)^{-1/2} \boldsymbol{A}_0 \boldsymbol{x}_i}{\left\| \left( \frac{1}{\theta m} \boldsymbol{Y} \boldsymbol{Y}^T \right)^{-1/2} \boldsymbol{A}_0 \boldsymbol{x}_i \right\|_2} - \frac{\boldsymbol{A}^T \boldsymbol{A} \boldsymbol{x}_i}{\|\boldsymbol{A}^T \boldsymbol{A} \boldsymbol{x}_i\|_2} \right\|_2 \tag{4.18}$$

$$\leq \frac{2}{\|\boldsymbol{A} \boldsymbol{x}_i\|_2} \left\| \left( \frac{1}{\theta m} \boldsymbol{Y} \boldsymbol{Y}^T \right)^{-1/2} \boldsymbol{A}_0 \boldsymbol{x}_i - \boldsymbol{A} \boldsymbol{x}_i \right\|_2 \tag{4.19}$$

$$\leq 2 \|\boldsymbol{A}_0\|_2 \left\| \left( \frac{1}{\theta m} \boldsymbol{Y} \boldsymbol{Y}^T \right)^{-1/2} - \left( \boldsymbol{A}_0 \boldsymbol{A}_0^T \right)^{-1/2} \right\|_2 \tag{4.20}$$

$$\leq \frac{8 \kappa^3 \delta}{\sigma_{\min}}, \tag{4.21}$$

where we have used Lemma 5.3 in the final bound.

Since $\|\cdot\|_4^4$ is convex, $\|\boldsymbol{\zeta}_{\text{init}}\|_4^4$ can be lower bounded via

$$\|\boldsymbol{\zeta}_{\text{init}}\|_4^4 \geq \left\| \hat{\boldsymbol{\zeta}}_{\text{init}} \right\|_4^4 + 4 \left\langle \hat{\boldsymbol{\zeta}}_{\text{init}}^{\circ 3}, \boldsymbol{\zeta}_{\text{init}} - \hat{\boldsymbol{\zeta}}_{\text{init}} \right\rangle \tag{4.22}$$

$$\geq \left\| \hat{\boldsymbol{\zeta}}_{\text{init}} \right\|_4^4 - 4 \left\| \boldsymbol{\zeta}_{\text{init}} - \hat{\boldsymbol{\zeta}}_{\text{init}} \right\|_2 \tag{4.23}$$

$$\geq \left\| \hat{\boldsymbol{\zeta}}_{\text{init}} \right\|_4^4 - \frac{32 \kappa^3 \delta}{\sigma_{\min}}. \tag{4.24}$$

Let $I = \text{supp}(\boldsymbol{x}_i)$, then the vector $\hat{\boldsymbol{\zeta}}_{\text{init}} = \mathcal{P}_{\mathbb{S}} \left[ \boldsymbol{A}^T \boldsymbol{A} \boldsymbol{x}_i \right]$ is composed of $|I|$ large components and small components on the off-support $I^c$ of $\boldsymbol{x}_i$.

**Dense Component of $\hat{\boldsymbol{\zeta}}_{\text{init}}$.** Note that $\left\| \left( \boldsymbol{A}^T \boldsymbol{A} \right)_{I^c, I} \boldsymbol{x}_i \right\|_2 \leq \left\| \text{offdiag} \left( \boldsymbol{A}^T \boldsymbol{A} \right) \boldsymbol{x}_i \right\|_2$ with $\left\| \text{offdiag} \left( \boldsymbol{A}^T \boldsymbol{A} \right) \right\|_\infty \leq \mu$. We have

$$\mathbb{E} \left[ \text{offdiag} \left( \boldsymbol{A}^T \boldsymbol{A} \right) \boldsymbol{x}_i \right] = \boldsymbol{0} \tag{4.25}$$

$$\mathbb{E} \left[ \left| \boldsymbol{e}_j^T \text{offdiag} \left( \boldsymbol{A}^T \boldsymbol{A} \right) \boldsymbol{x}_i \right|^2 \right] = \theta \left\| \boldsymbol{e}_j^T \text{offdiag} \left( \boldsymbol{A}^T \boldsymbol{A} \right) \right\|_2^2$$
$$\leq \mu^2 \theta k \tag{4.26}$$

With Bernstein's Inequality, the summation of moment-bounded independent random variables can be controlled via

$$\mathbb{P} \left[ \left| \boldsymbol{e}_j^T \text{offdiag} \left( \boldsymbol{A}^T \boldsymbol{A} \right) \boldsymbol{x}_i \right| \geq \mu t \right] \leq 2 \exp \left( - \frac{t^2}{2 \theta k + 2t} \right) \tag{4.27}$$

and via union bound

$$\mathbb{P} \left[ \left\| \text{offdiag} \left( \boldsymbol{A}^T \boldsymbol{A} \right) \boldsymbol{x}_i \right\|_2^2 \geq 2k \left( \mu t \right)^2 \right] \leq 4k \exp \left( - \frac{t^2}{2 \theta k + 2t} \right) \tag{4.28}$$

Therefore, setting $t^2 = 9\theta k \log k$, we obtain

$$\left\| \text{offdiag} \left( \boldsymbol{A}^T \boldsymbol{A} \right) \boldsymbol{x}_i \right\|_2^2 \leq 18 \mu^2 \theta k^2 \log k \tag{4.29}$$

with failure probability bounded by

$$4k \exp \left( -\frac{9\theta k \log k}{2\theta k + 2\sqrt{9\theta k \log k}} \right)$$

$$= 4k \exp \left( -\frac{9 \log k}{2 + 6\sqrt{(\theta k)^{-1} \log k}} \right) \tag{4.30}$$

$$\leq 4k^{-2} \tag{4.31}$$

The last inequality is derived under the assumption $(\theta k)^{-1} \log k \leq \frac{1}{64}$.

**Spiky Component of $\hat{\zeta}_{\text{init}}$.**   On the other hand,

$$\mathbb{E} \left[ \left\| \text{diag} \left( \boldsymbol{A}^T \boldsymbol{A} \right) \boldsymbol{x}_i \right\|_2^2 \right] = \theta \left\| \text{diag} \left( \boldsymbol{A}^T \boldsymbol{A} \right) \right\|_F^2 \tag{4.32}$$

$$= \theta k. \tag{4.33}$$

For $\text{diag} \left( \boldsymbol{A}^T \boldsymbol{A} \right) \boldsymbol{x}_i$, applying the moment control Bernstein Inequality, we have

$$\mathbb{P} \left[ \left| \left\| \text{diag} \left( \boldsymbol{A}^T \boldsymbol{A} \right) \boldsymbol{x}_i \right\|_2^2 - \mathbb{E} \left[ \cdot \right] \right| \geq t \right] \leq 2 \exp \left( -\frac{t^2}{2\theta k + 2t} \right). \tag{4.34}$$

By setting $t = 2\sqrt{\theta k \log k}$, we obtain that with probability no smaller than $1 - k^{-1}$,

$$\left\| \text{diag} \left( \boldsymbol{A}^T \boldsymbol{A} \right) \boldsymbol{x}_i \right\|_2^2 \geq \theta k - 2\sqrt{\theta k \log k}. \tag{4.35}$$

Denote the following events for the entry-wise magnitude

$$\mathcal{E}_j = \left\{ |\boldsymbol{e}_j^T \text{offdiag}(\boldsymbol{A}^T \boldsymbol{A}) \boldsymbol{x}_i| \leq \mu t \right\}, \tag{4.36}$$

and for the support size

$$\mathcal{E}_{\text{supp}} = \left\{ \|\boldsymbol{x}_i\|_0 \leq 4\theta k \right\}. \tag{4.37}$$

On their intersection $\mathcal{E}_{\text{supp}} \cap \bigcap_{j=1}^{2k} \mathcal{E}_j$, we have

$$\left\| \text{offdiag}(\boldsymbol{A}^T \boldsymbol{A})_{I,I} \boldsymbol{x}_i \right\|_2^2 \leq 4\theta k (\mu t)^2. \tag{4.38}$$

The the failure probability can be bounded from the union bound as

$$\mathbb{P} \left[ \left\| \text{offdiag}(\boldsymbol{A}^T \boldsymbol{A})_{I,I} \boldsymbol{x}_i \right\|_2^2 \geq 4\theta k (\mu t)^2 \right]$$

$$\leq \mathbb{P} \left[ \left( \mathcal{E}_{\text{supp}} \cap \bigcap_j \mathcal{E}_j \right)^c \right] \tag{4.39}$$

$$= \mathbb{P} \left[ \mathcal{E}_{\text{supp}}^c \cup \bigcup_j \mathcal{E}_j^c \right] \tag{4.40}$$

$$\leq \mathbb{P} \left[ \mathcal{E}_{\text{supp}}^c \right] + \sum_j \mathbb{P} \left[ \mathcal{E}_j^c \right] \tag{4.41}$$

$$\leq \exp(-\theta k) + 4k \exp\left(-\frac{t^2}{2\theta k + 2t}\right). \tag{4.42}$$

Therefore, by setting $t^2 = 9\theta k \log k$, we obtain

$$\left\|\text{offdiag}\left(\boldsymbol{A}^T\boldsymbol{A}\right)_{I,I}\boldsymbol{x}_i\right\|_2^2 \leq 36\mu^2\theta^2 k^2 \log k \tag{4.43}$$

with probability no smaller than $1 - \exp\left(-\theta k\right) - 8k^{-2}$.

Therefore, with probability no smaller than $1 - k^{-1} - 8k^{-2} - \exp\left(-\theta k\right)$,

$$\left\|\text{diag}\left(\boldsymbol{A}^T\boldsymbol{A}\right)\boldsymbol{x}_i\right\|_2^2 \geq \theta k - 2\sqrt{\theta k \log k} \tag{4.44}$$

$$\left\|\text{offdiag}\left(\boldsymbol{A}^T\boldsymbol{A}\right)_{I,I}\boldsymbol{x}_i\right\|_2^2 \leq 36\mu^2\theta^2 k^2 \log k \tag{4.45}$$

and via Cauchy-Schwatz inequality, we obtain

$$\left\|\left(\boldsymbol{A}^T\boldsymbol{A}\right)_{I,I}\boldsymbol{x}_i\right\|_2^2 \tag{4.46}$$

$$= \left\|\text{diag}\left(\boldsymbol{A}^T\boldsymbol{A}\right)\boldsymbol{x}_i + \text{offdiag}\left(\boldsymbol{A}^T\boldsymbol{A}\right)_{I,I}\boldsymbol{x}_i\right\|_2^2 \tag{4.47}$$

$$= \left\|\text{diag}\left(\boldsymbol{A}^T\boldsymbol{A}\right)\boldsymbol{x}_i\right\|_2^2 + \left\|\text{offdiag}\left(\boldsymbol{A}^T\boldsymbol{A}\right)_{I,I}\boldsymbol{x}_i\right\|_2^2$$
$$+ 2\left\langle\text{diag}\left(\boldsymbol{A}^T\boldsymbol{A}\right)\boldsymbol{x}_i, \text{offdiag}\left(\boldsymbol{A}^T\boldsymbol{A}\right)_{I,I}\boldsymbol{x}_i\right\rangle \tag{4.48}$$

$$\geq \left\|\text{diag}\left(\boldsymbol{A}^T\boldsymbol{A}\right)\boldsymbol{x}_i\right\|_2^2$$
$$- 2\left\|\text{diag}\left(\boldsymbol{A}^T\boldsymbol{A}\right)\boldsymbol{x}_i\right\|_2\left\|\text{offdiag}\left(\boldsymbol{A}^T\boldsymbol{A}\right)_{I,I}\boldsymbol{x}_i\right\|_2 \tag{4.49}$$

$$\geq \theta k\left(1 - 2\sqrt{(\theta k)^{-1}\log k} - 12\mu\sqrt{\theta k \log k}\right) \tag{4.50}$$

$$\geq \theta k/2. \tag{4.51}$$

The last equation is derived by plugging in

$$(\theta k)^{-1}\log k \leq \tfrac{1}{64}, \quad \mu^2\theta k \log k \leq \tfrac{1}{48^2} \tag{4.52}$$

under the assumption

$$64k^{-1}\log k \leq \theta \leq \tfrac{1}{48^2}\mu^{-2}k^{-1}\log^{-1}k. \tag{4.53}$$

**Lower Bound of** $\|\cdot\|_4^4$**.** Since with probability no smaller than $1 - 4k^{-2}$, $\left\|\text{offdiag}\left(\boldsymbol{A}^T\boldsymbol{A}\right)\boldsymbol{x}_i\right\|_2^2 \leq 36\mu^2\theta k^2 \log k$ obtains and the relative $\|\cdot\|_2^2$ norm between the flat entries to the spiky entries in $\boldsymbol{A}^T\boldsymbol{A}\boldsymbol{x}_i$ can be bounded as

$$\frac{\left\|\left(\boldsymbol{A}^T\boldsymbol{A}\right)_{I^c,I}\boldsymbol{x}_i\right\|_2^2}{\left\|\left(\boldsymbol{A}^T\boldsymbol{A}\right)_{I,I}\boldsymbol{x}_i\right\|_2^2} \leq \frac{\left\|\text{offdiag}\left(\boldsymbol{A}^T\boldsymbol{A}\right)\boldsymbol{x}_i\right\|_2^2}{\left\|\left(\boldsymbol{A}^T\boldsymbol{A}\right)_{I,I}\boldsymbol{x}_i\right\|_2^2} \tag{4.54}$$

$$\leq 36\mu^2 k \log k \doteq r. \tag{4.55}$$

Since

$$\left\|\hat{\boldsymbol{\zeta}}_{\text{init}}\right\|_4^4 = \left\|\mathcal{P}_{\mathbb{S}}\left[\boldsymbol{A}^T\boldsymbol{A}\boldsymbol{x}_i\right]\right\|_4^4 \tag{4.56}$$

$$= \frac{1}{\left\| \boldsymbol{A}^T \boldsymbol{A} \boldsymbol{x}_i \right\|_2^4} \left\| \left( \boldsymbol{A}^T \boldsymbol{A} \right)_{I^c, I} \boldsymbol{x}_i \right\|_4^4$$

$$+ \frac{1}{\left\| \boldsymbol{A}^T \boldsymbol{A} \boldsymbol{x}_i \right\|_2^4} \left\| \left( \boldsymbol{A}^T \boldsymbol{A} \right)_{I, I} \boldsymbol{x}_i \right\|_4^4 \tag{4.57}$$

$$\geq \frac{1}{\left\| \boldsymbol{A}^T \boldsymbol{A} \boldsymbol{x}_i \right\|_2^4} \left\| \left( \boldsymbol{A}^T \boldsymbol{A} \right)_{I, I} \boldsymbol{x}_i \right\|_4^4 \tag{4.58}$$

$$= \frac{\left\| \left( \boldsymbol{A}^T \boldsymbol{A} \right)_{I, I} \boldsymbol{x}_i \right\|_2^4 \left\| \mathcal{P}_{\mathbb{S}} \left[ \left( \boldsymbol{A}^T \boldsymbol{A} \right)_{I, I} \boldsymbol{x}_i \right] \right\|_4^4}{\left\| \left( \boldsymbol{A}^T \boldsymbol{A} \right)_{I, I} \boldsymbol{x}_i + \left( \boldsymbol{A}^T \boldsymbol{A} \right)_{I^c, I} \boldsymbol{x}_i \right\|_2^4} \tag{4.59}$$

$$\geq \frac{1}{(1+r)^2} \left\| \mathcal{P}_{\mathbb{S}} \left[ \left( \boldsymbol{A}^T \boldsymbol{A} \right)_{I, I} \boldsymbol{x}_i \right] \right\|_4^4 \tag{4.60}$$

and with high probability $1 - \exp\left(-\theta k\right)$ according to Lemma 2.4, $\mathcal{P}_{\mathbb{S}} \left[ \left( \boldsymbol{A}^T \boldsymbol{A} \right)_{I, I} \boldsymbol{x}_i \right]$ satisfies

$$\left\| \mathcal{P}_{\mathbb{S}} \left[ \left( \boldsymbol{A}^T \boldsymbol{A} \right)_{I, I} \boldsymbol{x}_i \right] \right\|_4^4 \geq \frac{1}{\left\| \boldsymbol{x}_i \right\|_0} \geq \frac{1}{2\theta\left(2k - 1\right)}, \tag{4.61}$$

Together, we have

$$\left\| \boldsymbol{\zeta}_{\text{init}} \right\|_4^4 \geq \left\| \hat{\boldsymbol{\zeta}}_{\text{init}} \right\|_4^4 - \frac{32\kappa^3 \delta}{\sigma_{\min}} \tag{4.62}$$

$$\geq \frac{1}{(1+r)^2} \left\| \mathcal{P}_{\mathbb{S}} \left[ \left( \boldsymbol{A}^T \boldsymbol{A} \right)_{I, I} \boldsymbol{x}_i \right] \right\|_4^4$$

$$- \frac{640 C^{-1/4}}{\theta k \left(1 + 36\mu^2 k \log k\right)^2} \tag{4.63}$$

$$\geq \left( \frac{1}{4} - \frac{640}{C^{1/4}} \right) \frac{1}{\theta k \left(1 + 36\mu^2 k \log k\right)^2} \tag{4.64}$$

holds with probability no smaller than $1 - k^{-1} - 8k^{-2} - 2\exp\left(-\theta k\right) - 24k \exp\left(-\frac{1}{144} \min\left\{k, 3\sqrt{\theta m}\right\}\right) - 48k^{-7} - 48m^{-5}$. To make sure $\left\| \boldsymbol{\zeta}_{\text{init}} \right\|_4^6 \geq 3C_\star \mu \kappa^2$ as desired, we require the sparsity to satisfy

$$\theta \leq \left( \frac{1}{4} - \frac{640}{C^{1/4}} \right) \left( 3C_\star \mu \kappa^2 \right)^{-2/3} k^{-1} \left( 1 + 36\mu^2 k \log k \right)^{-2}, \tag{4.65}$$

then the initialization $\boldsymbol{q}_{\text{init}} \in \hat{\mathcal{R}}_{3C_\star}$ follows by **??**. ∎

# 5 Preconditioning

**Lemma 5.1** *Suppose* $\boldsymbol{x}_0 \sim_{\text{i.i.d.}} \text{BG}\left(\theta\right) \in \mathbb{R}^m$, *then following inequality holds*

$$\left\| \frac{1}{\theta m} \boldsymbol{X}_0 \boldsymbol{X}_0^T - \boldsymbol{I} \right\|_2 \leq 10\sqrt{k \log m / m}, \tag{5.1}$$

*with probability no smaller than* $1 - 2\exp\left(-\theta k\right) - 24k \exp\left(-\frac{1}{144} \min\left\{k, 3\sqrt{\theta m}\right\}\right) - 48k^{-7} - 48m^{-5}$.

**Proof** Since

$$\left\| \frac{1}{\theta m} \boldsymbol{X}_0 \boldsymbol{X}_0^T - \boldsymbol{I} \right\|_2$$

$$\leq \left\| \mathrm{diag} \left( \frac{1}{\theta m} \boldsymbol{X}_0 \boldsymbol{X}_0^T \right) - \boldsymbol{I} \right\|_2$$

$$+ \left\| \mathrm{offdiag} \left( \frac{1}{\theta m} \boldsymbol{X}_0 \boldsymbol{X}_0^T \right) \right\|_2, \tag{5.2}$$

which is bounded by $\delta$ with probability no smaller than $1 - \varepsilon_d - \varepsilon_o$ whenever the probability that each of the terms is upper bounded by $\delta/2$ satisfies

$$\mathbb{P} \left[ \left\| \mathrm{diag} \left( \frac{1}{\theta m} \boldsymbol{X}_0 \boldsymbol{X}_0^T \right) - \boldsymbol{I} \right\|_2 \geq \delta/2 \right] \leq \varepsilon_d, \tag{5.3}$$

$$\mathbb{P} \left[ \left\| \mathrm{offdiag} \left( \frac{1}{\theta m} \boldsymbol{X}_0 \boldsymbol{X}_0^T \right) - \boldsymbol{I} \right\|_2 \geq \delta/2 \right] \leq \varepsilon_o. \tag{5.4}$$

**Diagonal of $\frac{1}{\theta m} \boldsymbol{X}_0 \boldsymbol{X}_0^T$.** Note that $\mathrm{diag} \left( \boldsymbol{X}_0 \boldsymbol{X}_0^T \right) = \| \boldsymbol{x}_0 \|_2^2 \, \boldsymbol{I}$, so

$$\left\| \mathrm{diag} \left( \frac{1}{\theta m} \boldsymbol{X}_0 \boldsymbol{X}_0^T \right) - \boldsymbol{I} \right\|_2 = \left| \frac{1}{\theta m} \| \boldsymbol{x}_0 \|_2^2 - 1 \right|. \tag{5.5}$$

We calculate the moment for each summand of $\| \boldsymbol{x}_0 \|_2^2$. The summands can be seen as a $\chi_1^2$ random variable but populated with probability $\theta$, whence

$$\mathbb{E}_{x_i \sim \mathrm{BG}(\theta)} \left[ \left( x_i^2 \right)^p \right] = \theta \, \mathbb{E}_{X_i \sim \chi_1^2} \left[ X_i^p \right] \tag{5.6}$$

$$= \theta \frac{\Gamma \left( p + \frac{1}{2} \right)}{\Gamma \left( \frac{1}{2} \right)} \tag{5.7}$$

$$\leq \frac{\theta p! \, (2)^p}{2} \tag{5.8}$$

$$= \frac{p!}{2} \sigma^2 R^{p-2}. \tag{5.9}$$

Apply Bernstein's inequality for moment bounded random variables (9.4) with $R = 2, \sigma^2 = 4\theta$, then

$$\mathbb{P} \left[ \left| \frac{1}{m} \| \boldsymbol{x}_0 \|_2^2 - \theta \right| \geq t \right] \leq 2 \exp \left( -\frac{mt^2}{8\theta + 4t} \right). \tag{5.10}$$

By taking $t = \frac{1}{2} \theta \delta$, we obtain

$$\mathbb{P} \left[ \left\| \mathrm{diag} \left( \frac{1}{\theta m} \boldsymbol{X}_0 \boldsymbol{X}_0^T \right) - \boldsymbol{I} \right\|_2 \geq \delta/2 \right]$$

$$\leq 2 \exp \left( -\frac{\theta m \delta^2}{32 + 8\delta} \right) \tag{5.11}$$

$$\leq 2 \exp \left( -\frac{100 \theta k \log m}{32 + 80 \sqrt{k \log m / m}} \right) \tag{5.12}$$

$$\leq 2 \exp \left( -\theta k \right). \tag{5.13}$$

**Off-diagonal of $\frac{1}{\theta m} \boldsymbol{X}_0 \boldsymbol{X}_0^T$.** Note that $\mathrm{offdiag} \left( \boldsymbol{X}_0 \boldsymbol{X}_0^T \right)$ is a sub-circulant matrix generated by

$$\boldsymbol{r}_{\boldsymbol{x}_0} = \left[ r_{\boldsymbol{x}_0} \left( 2k - 2 \right), \cdots, 0, \cdots, r_{\boldsymbol{x}_0} \left( 2k - 2 \right) \right]^T \tag{5.14}$$

with $r_{\boldsymbol{x}_0} (\tau) = \langle \boldsymbol{x}_0, s_\tau [\boldsymbol{x}_0] \rangle$ for $\tau = 1, \cdots, 2k - 2$. Equivalently, we can write

$$\boldsymbol{r}_{\boldsymbol{x}_0} = \boldsymbol{R}_{\boldsymbol{x}_0}^T \boldsymbol{x}_0, \tag{5.15}$$

with

$$\boldsymbol{R}_{\boldsymbol{x}_0} = [s_{2k-2}[\boldsymbol{x}_0], \cdots, \boldsymbol{0}, \cdots, s_{2k-2}[\boldsymbol{x}_0]] \in \mathbb{R}^{m \times (4k-3)}. \tag{5.16}$$

Operator norm of a circulant matrix is defined as the following

$$\left\| \text{offdiag}\left( \frac{1}{\theta m} \boldsymbol{X}_0 \boldsymbol{X}_0^T \right) \right\|_2 = \max_{l=0,\dots,4k-4} \left| \left\langle \boldsymbol{v}_l, \frac{1}{\theta m} \boldsymbol{r}_{\boldsymbol{x}_0} \right\rangle \right|, \tag{5.17}$$

where $\boldsymbol{v}_l$ is the $l$-th (discrete) Fourier basis vector

$$\boldsymbol{v}_l = \left[ 1,\ e^{l\frac{2\pi j}{4k-3}}, \cdots,\ e^{l(4k-4)\frac{2\pi j}{4k-3}} \right]^T, \tag{5.18}$$

and $j$ is the imaginary unit. Let $v_{l,\tau} = \boldsymbol{v}_l\left(2k-2-\tau\right) + \boldsymbol{v}_l\left(2k-2+\tau\right)$, then

$$\langle \boldsymbol{v}_l, \boldsymbol{r}_{\boldsymbol{x}_0} \rangle = \sum_{\tau=1}^{2k-2} v_{l,\tau} \langle \boldsymbol{x}_0, s_\tau[\boldsymbol{x}_0] \rangle \tag{5.19}$$

$$= \sum_{\tau=1}^{2k-2} v_{l,\tau} \sum_{i=0}^{m-1} \boldsymbol{x}_0\left(i\right) \boldsymbol{x}_0\left([i+\tau]_m\right). \tag{5.20}$$

By decoupling (Theorem 3.4.1 of [DlPG99]), the tail probability of the weighted autocorrelation $\langle \boldsymbol{v}_l, \boldsymbol{r}_{\boldsymbol{x}_0} \rangle$ can be upper bounded via

$$\mathbb{P}\left[|\langle \boldsymbol{v}_l, \boldsymbol{r}_{\boldsymbol{x}_0} \rangle| > t\right]$$

$$= \mathbb{P}\left[ \left| \sum_{\tau=1}^{2k-2} v_{l,\tau} \langle \boldsymbol{x}_0, s_\tau[\boldsymbol{x}_0] \rangle \right| > t \right] \tag{5.21}$$

$$\leq 6\,\mathbb{P}\left[ \left| \sum_{\tau=1}^{2k-2} v_{l,\tau} \langle \boldsymbol{x}_0, s_\tau[\boldsymbol{x}_0'] \rangle \right| > \frac{t}{6} \right], \tag{5.22}$$

where $\boldsymbol{x}_0' \sim_{\text{i.i.d.}} \mathrm{BG}\left(\theta\right)$ is an independent copy of the random vector $\boldsymbol{x}_0$, we have Plugging in $\langle \boldsymbol{v}_l, \boldsymbol{r}_{\boldsymbol{x}_0} \rangle = \left\langle \boldsymbol{v}_l, \boldsymbol{R}_{\boldsymbol{x}_0}^T \boldsymbol{x}_0 \right\rangle = \langle \boldsymbol{R}_{\boldsymbol{x}_0} \boldsymbol{v}_l, \boldsymbol{x}_0 \rangle$.

$$\mathbb{P}\left[ \left| \left\langle \boldsymbol{v}_l, \frac{1}{\theta m} \boldsymbol{r}_{\boldsymbol{x}_0} \right\rangle \right| > t \right] \leq 6\,\mathbb{P}\left[ \left| \frac{1}{\theta m} \langle \boldsymbol{R}_{\boldsymbol{x}_0'} \boldsymbol{v}_l, \boldsymbol{x}_0 \rangle \right| > \frac{t}{6} \right]. \tag{5.23}$$

Again with Bernstein's inequality for moment bounded random variable, we have

$$\mathbb{P}\left[ \left| \frac{1}{\theta m} \langle \boldsymbol{R}_{\boldsymbol{x}_0'} \boldsymbol{v}_l, \boldsymbol{x}_0 \rangle \right| \geq t \right]$$

$$\leq 2\exp\left( -\frac{\theta m^2 t^2}{2\left\| \boldsymbol{R}_{\boldsymbol{x}_0'} \boldsymbol{v}_l \right\|_2^2 + 2\left\| \boldsymbol{R}_{\boldsymbol{x}_0'} \boldsymbol{v}_l \right\|_\infty m t} \right) \tag{5.24}$$

**Control $\left\| \boldsymbol{R}_{\boldsymbol{x}_0'} \boldsymbol{v}_l \right\|_2$.**

$$\left\| \boldsymbol{R}_{\boldsymbol{x}_0} \boldsymbol{v}_l \right\|_2^2 \leq \left\| \boldsymbol{R}_{\boldsymbol{x}_0} \right\|_2^2 \left\| \boldsymbol{v}_l \right\|_2^2 = k \left\| \boldsymbol{R}_{\boldsymbol{x}_0} \right\|_2^2 \tag{5.25}$$

With tail bound of the operator norm of a circulant matrix in Lemma 2.6, we have

$$\mathbb{P}\left[ \left\| \boldsymbol{R}_{\boldsymbol{x}_0} \right\|_2 \geq t \right] \leq 4m \exp\left( -\frac{t^2}{2\theta m + 2t} \right) \tag{5.26}$$

**Control $\left\| \boldsymbol{R}_{\boldsymbol{x}_0'} \boldsymbol{v}_l \right\|_\infty$.** For a discrete Fourier basis $\boldsymbol{v}_l$ as defined, we have

$$\left\| \boldsymbol{v}_l \right\|_2^2 = \left\| \boldsymbol{v}_l \right\|_0 = 4k-3, \quad \left\| \boldsymbol{v}_l \right\|_\infty = 1 \tag{5.27}$$

Note that

$$\left\| \boldsymbol{R}_{\boldsymbol{x}_0} \boldsymbol{v}_l \right\|_\infty = \max_{\tau=1,\ldots,2k-2} |\langle s_\tau[\boldsymbol{x}_0], \boldsymbol{v}_l \rangle| \tag{5.28}$$

and moment control Bernstein inequality implies that

$$\mathbb{P}\left[ |\langle s_\tau[\boldsymbol{x}_0], \boldsymbol{v}_l \rangle| \geq t \right] \leq 2 \exp\left( -\frac{t^2}{2\theta \left\| \boldsymbol{v}_l \right\|_2^2 + 2 \left\| \boldsymbol{v}_l \right\|_\infty t} \right). \tag{5.29}$$

with union bound, we obtain

$$\mathbb{P}\left[ \left\| \boldsymbol{R}_{\boldsymbol{x}_0} \boldsymbol{v}_l \right\|_\infty \geq t \right] \leq \sum_{\tau=1}^{2k-2} \mathbb{P}\left[ |\langle s_\tau[\boldsymbol{x}_0], \boldsymbol{v}_l \rangle| \geq t \right] \tag{5.30}$$

$$\leq 4k \exp\left( -\frac{t^2}{8\theta k + 2t} \right) \tag{5.31}$$

Therefore, by plugging in

$$\left\| \boldsymbol{R}_{\boldsymbol{x}_0'} \boldsymbol{v}_l \right\|_\infty \leq t_1 = 10\sqrt{\theta k \log k}, \tag{5.32}$$

$$\left\| \boldsymbol{R}_{\boldsymbol{x}_0'} \boldsymbol{v}_l \right\|_2 \leq t_2 = 5\sqrt{\theta m \log m}, \tag{5.33}$$

we obtain the following probabilities

$$\mathbb{P}\left[ \left\| \boldsymbol{R}_{\boldsymbol{x}_0'} \boldsymbol{v}_l \right\|_\infty \geq t_1 \right] \leq 4k \exp\left( -\frac{t_1^2}{8\theta k + 2t_1} \right)$$

$$\leq 4k^{-8}, \tag{5.34}$$

$$\mathbb{P}\left[ \left\| \boldsymbol{R}_{\boldsymbol{x}_0'} \right\|_2 \geq t_2 \right] \leq 4m \exp\left( -\frac{t_2^2}{2\theta m + 2t_2} \right)$$

$$\leq 4m^{-6}. \tag{5.35}$$

Denoting event

$$\boldsymbol{E} = \left\{ \left\| \boldsymbol{R}_{\boldsymbol{x}_0'} \boldsymbol{v}_l \right\|_\infty \leq t_1, \left\| \boldsymbol{R}_{\boldsymbol{x}_0'} \right\|_2 \leq t_2 \right\}, \tag{5.36}$$

and combining these bounds with (5.23), we obtain

$$\mathbb{P}\left[ \left\| \text{offdiag}\left( \frac{1}{\theta m} \boldsymbol{X}_0 \boldsymbol{X}_0^T \right) \right\|_2 \geq \delta/2 \right]$$

$$\leq 6\,\mathbb{P}\left[ \max_l \left| \frac{1}{\theta m} \left\langle \boldsymbol{R}_{\boldsymbol{x}_0'} \boldsymbol{v}_l, \boldsymbol{x}_0 \right\rangle \right| \geq \frac{\delta}{12} \right] \tag{5.37}$$

$$\leq 12k\,\mathbb{P}\left[ \left| \frac{1}{\theta m} \left\langle \boldsymbol{R}_{\boldsymbol{x}_0'} \boldsymbol{v}_l, \boldsymbol{x}_0 \right\rangle \right| \geq \frac{\delta}{12} \right] \tag{5.38}$$

$$\leq 12k\,\mathbb{P}\left[ \left\| \boldsymbol{R}_{\boldsymbol{x}_0'} \boldsymbol{v}_l \right\|_\infty > t_1 \right] + 12k\,\mathbb{P}\left[ \left\| \boldsymbol{R}_{\boldsymbol{x}_0'} \right\|_2 > t_2 \right]$$

$$+ 12k\,\mathbb{P}\left[ \left| \frac{1}{\theta m} \left\langle \boldsymbol{R}_{\boldsymbol{x}_0'} \boldsymbol{v}_l, \boldsymbol{x}_0 \right\rangle \right| \geq \frac{\delta}{12} \mid \boldsymbol{E} \right] \tag{5.39}$$

$$\leq 24k \exp\left( -\frac{100\theta k m \log m / 144}{50\theta m \log m + \frac{200}{12} k \sqrt{\theta m \log k \log m}} \right)$$

$$+ 12k\left( 4k^{-8} + 4m^{-6} \right) \tag{5.40}$$

$$\left( t_1 = 10\sqrt{\theta k \log k},\ t_2 = 5\sqrt{\theta m \log m} \right)$$

$$\leq 24k \exp\left(-\tfrac{1}{144} \min\left\{k, 3\sqrt{\theta m}\right\}\right) + 48k^{-7} + 48m^{-5} \tag{5.41}$$

At last, by combining the control for both the diagonal and off-diagonal term, we obtain that with probability no smaller than $1 - 2\exp\left(-\theta k\right) - 24k \exp\left(-\tfrac{1}{144} \min\left\{k, 3\sqrt{\theta m}\right\}\right) - 48k^{-7} - 48m^{-5}$,

$$\left\| \frac{1}{\theta m} \boldsymbol{X}_0 \boldsymbol{X}_0^T - \boldsymbol{I} \right\|_2 \leq 10\sqrt{k \log m/m}, \tag{5.42}$$

holds and completes the proof. ∎

**Lemma 5.2** *Suppose* $\delta = \left\| \frac{1}{\theta m} \boldsymbol{X}_0 \boldsymbol{X}_0^T - \boldsymbol{I} \right\|_2 \leq 1/\left(2\kappa^2\right)$, *then*

$$\left\| \left( \frac{1}{\theta m} \boldsymbol{Y}\boldsymbol{Y}^T \right)^{1/2} \left( \boldsymbol{A}_0 \boldsymbol{A}_0^T \right)^{-1/2} - \boldsymbol{I} \right\|_2 \leq \kappa^2 \delta / \sigma_{\min}. \tag{5.43}$$

**Proof** As in by [?], we denote the directional derivative of $f$ at direction $\boldsymbol{\Delta}$ with

$$Df(\boldsymbol{M})\left(\boldsymbol{\Delta}\right) = \frac{d}{dt}\bigg|_{t=0} f(\boldsymbol{M} + t\boldsymbol{\Delta}), \tag{5.44}$$

Denote symmetric matrix $\boldsymbol{M} = \boldsymbol{A}_0 \boldsymbol{A}_0^T = \boldsymbol{U}\boldsymbol{\Lambda}\boldsymbol{U}^T$, with $\lambda_{\max}$ and $\lambda_{\min}$ being its maximum and minimum eigenvalue. Then we have

$$\frac{1}{\theta m} \boldsymbol{Y}\boldsymbol{Y}^T = \boldsymbol{M} + \boldsymbol{\Delta}, \quad \|\boldsymbol{\Delta}\|_2 \leq \lambda_{\max}\delta. \tag{5.45}$$

Then derivative of $f$ with $Df(\boldsymbol{M})$. By differential calculus, we can obtain that

$$\left\| \left( \frac{1}{\theta m} \boldsymbol{Y}\boldsymbol{Y}^T \right)^{1/2} \left( \boldsymbol{A}_0 \boldsymbol{A}_0^T \right)^{-1/2} - \boldsymbol{I} \right\|_2$$

$$= \left\| \left( \boldsymbol{A}_0 \boldsymbol{A}_0^T + \boldsymbol{\Delta} \right)^{1/2} \left( \boldsymbol{A}_0 \boldsymbol{A}_0^T \right)^{-1/2} - \boldsymbol{I} \right\|_2 \tag{5.46}$$

$$= \left\| \left( \boldsymbol{A}_0 \boldsymbol{A}_0^T \right)^{-1/2} \int_{t=0}^{1} Df\left( \boldsymbol{A}_0 \boldsymbol{A}_0^T + t\boldsymbol{\Delta} \right)(\boldsymbol{\Delta})\, dt \right\|_2 \tag{5.47}$$

$$\leq \sup_{t \in [0,1]} \left\| Df\left( \boldsymbol{A}_0 \boldsymbol{A}_0^T + t\boldsymbol{\Delta} \right) \right\|_2 \|\boldsymbol{\Delta}\|_2 \left\| \left( \boldsymbol{A}_0 \boldsymbol{A}_0^T \right)^{-1/2} \right\|_2 \tag{5.48}$$

$$\leq \sup_{t \in [0,1]} \left\| Df\left( \boldsymbol{A}_0 \boldsymbol{A}_0^T + t\boldsymbol{\Delta} \right) \right\|_2 \lambda_{\max}\delta / \sigma_{\min} \tag{5.49}$$

Moreover, we denote $f(t) = t^{1/2}$ and $g(t) = t^2$, then $f = g^{-1}$. The directional derivative of $g$ has following form

$$Dg\left(\boldsymbol{M}\right)\left(\boldsymbol{X}\right) = \boldsymbol{M}\boldsymbol{X} + \boldsymbol{X}\boldsymbol{M}, \tag{5.50}$$

and directional derivative $\boldsymbol{Z} = Df\left(\boldsymbol{M}\right)\left(\boldsymbol{X}\right)$ satisfies

$$\boldsymbol{M}\boldsymbol{Z} + \boldsymbol{Z}\boldsymbol{M} = \boldsymbol{X}. \tag{5.51}$$

Denote $\boldsymbol{M} = \boldsymbol{U}\boldsymbol{\Lambda}\boldsymbol{U}^T$ with $\boldsymbol{U}$ orthogonal, without loss of generality,

$$\boldsymbol{\Lambda}\boldsymbol{Z} + \boldsymbol{Z}\boldsymbol{\Lambda} = \boldsymbol{X}. \tag{5.52}$$

Applying Theorem VII.2.3 of [?], we have

$$\left\| Df\left(\boldsymbol{M}\right)\left(\boldsymbol{X}\right) \right\|_2 = \sup_{\|\boldsymbol{X}\|_2 \leq 1} \|\boldsymbol{Z}\|_2 \tag{5.53}$$

$$\leq \int_{t=0}^{\infty} \left\| e^{-\mathbf{\Lambda} t} \mathbf{X} e^{-\mathbf{\Lambda} t} \right\|_2 dt \tag{5.54}$$

$$\leq \int_{t=0}^{\infty} e^{-2\lambda_{\min} t} \left\| \mathbf{X} \right\|_2 dt \tag{5.55}$$

and

$$\sup_{t \in [0,1]} \left\| Df \left( \mathbf{A}_0 \mathbf{A}_0^T + t\mathbf{\Delta} \right) \right\|_2$$

$$\leq \frac{\left\| \mathbf{X} \right\|_2}{2 \left( \lambda_{\min} - \lambda_{\max} \delta \right)} \tag{5.56}$$

$$\leq 1/\lambda_{\min}. \tag{5.57}$$

Therefore,

$$\left\| \left( \frac{1}{\theta m} \mathbf{Y} \mathbf{Y}^T \right)^{1/2} \left( \mathbf{A}_0 \mathbf{A}_0^T \right)^{-1/2} - \mathbf{I} \right\|_2 \leq \kappa^2 \delta / \sigma_{\min}. \tag{5.58}$$

$\blacksquare$

**Lemma 5.3** *Suppose $\mathbf{A}_0$ has condition number $\kappa$ and*

$$\delta = \left\| \frac{1}{\theta m} \mathbf{X}_0 \mathbf{X}_0^T - \mathbf{I} \right\|_2 \leq 1/\left( 2\kappa^2 \right) \tag{5.59}$$

*then*

$$\left\| \left( \frac{1}{\theta m} \mathbf{Y} \mathbf{Y}^T \right)^{-1/2} - \left( \mathbf{A}_0 \mathbf{A}_0^T \right)^{-1/2} \right\|_2 \leq 4\kappa^2 \delta / \sigma_{\min}^2. \tag{5.60}$$

**Proof** Denote symmetric matrix

$$\mathbf{M} = \mathbf{A}_0 \mathbf{A}_0^T = \mathbf{U} \mathbf{\Lambda} \mathbf{U}^T, \tag{5.61}$$

with $\lambda_{\max}$ and $\lambda_{\min}$ being its maximum and minimum eigenvalue. Then we have

$$\frac{1}{\theta m} \mathbf{Y} \mathbf{Y}^T = \mathbf{M} + \mathbf{\Delta}, \quad \left\| \mathbf{\Delta} \right\|_2 \leq \lambda_{\max} \delta. \tag{5.62}$$

Then

$$\left\| \left( \frac{1}{\theta m} \mathbf{Y} \mathbf{Y}^T \right)^{-1/2} - \left( \mathbf{A}_0 \mathbf{A}_0^T \right)^{-1/2} \right\|_2$$

$$= \left\| \left( \mathbf{M} + \mathbf{\Delta} \right)^{-1/2} - \mathbf{M}^{-1/2} \right\|_2 \tag{5.63}$$

$$\leq \left\| \mathbf{\Delta} \right\|_2 \cdot \sup_{0 \leq t \leq 1} \left\| Df \left( \mathbf{M} + t\mathbf{\Delta} \right) \right\|_2. \tag{5.64}$$

Here, $f(t) = t^{-1/2}$ and $Df$ is the derivative of function $f$. In addition, we define function $g(t) = t^{-2}$, $h(t) = t^{-1}$, $w(t) = t^2$, and following function compositions hold

$$f = g^{-1}, \quad g = h \circ w. \tag{5.65}$$

For differential function $g$ and if $Dg \left( f \left( \mathbf{M} \right) \right) \neq 0$, we have

$$Df \left( \mathbf{M} \right) = \left[ Dg \left( f \left( \mathbf{M} \right) \right) \right]^{-1}. \tag{5.66}$$

The derivative of function $g$ satisfies the chain rule that

$$Dg\left(\boldsymbol{M}\right) = Dh\left(w\left(\boldsymbol{M}\right)\right)\left(Dw\left(\boldsymbol{M}\right)\right). \tag{5.67}$$

Plug in

$$Dh\left(\boldsymbol{M}\right)\left(\boldsymbol{X}\right) = -\boldsymbol{M}^{-1}\boldsymbol{X}\boldsymbol{M}^{-1}, \tag{5.68}$$
$$Dw\left(\boldsymbol{M}\right)\left(\boldsymbol{X}\right) = \boldsymbol{M}\boldsymbol{X} + \boldsymbol{X}\boldsymbol{M}, \tag{5.69}$$

we obtain that

$$Dg\left(\boldsymbol{M}\right)\left(\boldsymbol{X}\right) \tag{5.70}$$
$$= Dh\left(w\left(\boldsymbol{M}\right)\right)\left(Dw\left(\boldsymbol{M}\right)\left(\boldsymbol{X}\right)\right) \tag{5.71}$$
$$= Dh\left(w\left(\boldsymbol{M}\right)\right)\left[\boldsymbol{M}\boldsymbol{X} + \boldsymbol{X}\boldsymbol{M}\right] \tag{5.72}$$
$$= Dh\left(\boldsymbol{M}^2\right)\left[\boldsymbol{M}\boldsymbol{X} + \boldsymbol{X}\boldsymbol{M}\right] \tag{5.73}$$
$$= -\boldsymbol{M}^{-2}\left[\boldsymbol{M}\boldsymbol{X} + \boldsymbol{X}\boldsymbol{M}\right]\boldsymbol{M}^{-2} \tag{5.74}$$
$$= -\left[\boldsymbol{M}^{-1}\boldsymbol{X}\boldsymbol{M}^{-2} + \boldsymbol{M}^{-2}\boldsymbol{X}\boldsymbol{M}^{-1}\right]. \tag{5.75}$$

Since the function $g$ is differentiable and $Dg(\boldsymbol{M}) \neq \boldsymbol{0}$, then

$$Df\left(\boldsymbol{M}\right) = \left[Dg\left(f\left(\boldsymbol{M}\right)\right)\right]^{-1} \tag{5.76}$$
$$= \left[Dg\left(\boldsymbol{M}^{-1/2}\right)\right]^{-1}. \tag{5.77}$$

Hence, directional derivative $\boldsymbol{Z} \doteq Df\left(\boldsymbol{M}\right)\left(\boldsymbol{X}\right)$ satisfies

$$\boldsymbol{M}^{1/2}\boldsymbol{Z}\boldsymbol{M} + \boldsymbol{M}\boldsymbol{Z}\boldsymbol{M}^{1/2} = -\boldsymbol{X}. \tag{5.78}$$

Denote $\boldsymbol{M} = \boldsymbol{U}\boldsymbol{\Lambda}\boldsymbol{U}^T$ with $\boldsymbol{\Lambda} \succ 0$ and $\boldsymbol{U}$ orthogonal, without loss of generality

$$\boldsymbol{\Lambda}\boldsymbol{Z}\boldsymbol{\Lambda}^{1/2} + \boldsymbol{\Lambda}^{1/2}\boldsymbol{Z}\boldsymbol{\Lambda} = -\boldsymbol{X}. \tag{5.79}$$

Above equation can be reformulated as a Sylvester equation as following

$$\boldsymbol{\Lambda}^{1/2}\boldsymbol{Z} - \boldsymbol{Z}\left(-\boldsymbol{\Lambda}^{1/2}\right) = -\boldsymbol{\Lambda}^{-1/2}\boldsymbol{X}\boldsymbol{\Lambda}^{-1/2}. \tag{5.80}$$

From Theorem VII.2.3 of [?], when there are no common eigenvalues of $\boldsymbol{\Lambda}^{1/2}$ and $-\boldsymbol{\Lambda}^{1/2}$, then there exists a closed form solution for matrix $\boldsymbol{Z}$ that

$$\boldsymbol{Z} = \int_{t=0}^{\infty} e^{-\boldsymbol{\Lambda}^{1/2}t}\left(-\boldsymbol{\Lambda}^{-1/2}\boldsymbol{X}\boldsymbol{\Lambda}^{-1/2}\right)e^{-\boldsymbol{\Lambda}^{1/2}t}dt \tag{5.81}$$

Therefore, the operator norm of $Df\left(\boldsymbol{M}\right)$ can be obtained as

$$\|Df(\boldsymbol{M})(\boldsymbol{X})\|_2 = \sup_{\|\boldsymbol{X}\|_2 \leq 1} \|\boldsymbol{Z}\|_2 \tag{5.82}$$
$$\leq \int_{t=0}^{\infty}\left\|e^{-\boldsymbol{\Lambda}^{1/2}t}\left(\boldsymbol{\Lambda}^{-1/2}\boldsymbol{X}\boldsymbol{\Lambda}^{-1/2}\right)e^{-\boldsymbol{\Lambda}^{1/2}t}\right\|dt \tag{5.83}$$
$$\leq \int_{t=0}^{\infty} e^{-\lambda_{\min}t}\left\|\boldsymbol{\Lambda}^{-1/2}\boldsymbol{X}\boldsymbol{\Lambda}^{-1/2}\right\|dt \tag{5.84}$$
$$\leq \frac{\|\boldsymbol{X}\|}{\lambda_{\min}^2}. \tag{5.85}$$

Therefore

$$\left\| (\boldsymbol{M} + \boldsymbol{\Delta})^{-1/2} - \boldsymbol{M}^{-1/2} \right\|_2$$

$$\leq \frac{\|\boldsymbol{\Delta}\|_2}{\left( \lambda_{\min} - \|\boldsymbol{\Delta}\|_2 \right)^2} \tag{5.86}$$

$$\leq \frac{4 \|\boldsymbol{\Delta}\|_2}{\lambda_{\min}^2} \qquad \left( \delta \leq 1/\left( 2\kappa^2 \right) \right) \tag{5.87}$$

$$\leq \frac{4 \lambda_{\max} \delta}{\lambda_{\min}^2} \tag{5.88}$$

$$= \frac{4 \kappa^2 \delta}{\sigma_{\min}^2} \tag{5.89}$$

∎

# 6 Proof of the Main Theorem and Corollary

## 6.1 Proof of the Main Theorem

**Lemma 6.1** *If following inequalities hold*

$$\left\| \operatorname{grad}[\psi](\boldsymbol{q}) - \frac{3(1-\theta)}{\theta m^2} \operatorname{grad}[\varphi](\boldsymbol{q}) \right\|_2$$

$$\leq \frac{3 c_\star}{2 \kappa^2} \frac{1-\theta}{\theta m^2} \left\| \boldsymbol{A}^T \boldsymbol{q} \right\|_4^6, \tag{6.1}$$

$$\left\| \operatorname{Hess}[\psi](\boldsymbol{q}) - \frac{3(1-\theta)}{\theta m^2} \operatorname{Hess}[\varphi](\boldsymbol{q}) \right\|_2$$

$$\leq 3 \left( 1 - 6 c_\star - 36 c_\star^2 - 24 c_\star^3 \right) \frac{1-\theta}{\theta m^2} \left\| \boldsymbol{A}^T \boldsymbol{q} \right\|_4^4. \tag{6.2}$$

*for all $\boldsymbol{q} \in \mathcal{R}_{2C_\star}$ with $C_\star \geq 10$ and $c_\star = 1/C_\star$, then any local minimum $\bar{\boldsymbol{q}}$ of $\psi(\boldsymbol{q})$ in $\mathcal{R}_{2C_\star}$ satisfies $|\langle \bar{\boldsymbol{q}}, \mathcal{P}_{\mathbb{S}}[\boldsymbol{a}_l] \rangle| \geq 1 - 2 c_\star \kappa^{-2}$ for some index $l$.*

**Proof** Let

$$\boldsymbol{\delta}_{\mathrm{grad}} = \operatorname{grad}[\psi](\boldsymbol{q}) - \frac{3(1-\theta)}{\theta m^2} \operatorname{grad}[\varphi](\boldsymbol{q}), \tag{6.3}$$

and let

$$\bar{\boldsymbol{\delta}}_{\mathrm{grad}} = \frac{\theta m^2}{3(1-\theta)} \boldsymbol{\delta}_{\mathrm{grad}}. \tag{6.4}$$

Then at any stationary point of $\psi(\boldsymbol{q})$, we have

$$\boldsymbol{0} = \boldsymbol{A}^T \operatorname{grad}[\psi](\boldsymbol{q}) \tag{6.5}$$

$$= \frac{3(1-\theta)}{\theta m^2} \boldsymbol{A}^T \operatorname{grad}[\varphi](\boldsymbol{q}) + \boldsymbol{A}^T \boldsymbol{\delta}_{\mathrm{grad}}. \tag{6.6}$$

Hence for any index $i$, following equality always holds

$$0 = \|\boldsymbol{a}_i\|_2^2 \zeta_i^3 + \sum_{j \neq i} \langle \boldsymbol{a}_i, \boldsymbol{a}_j \rangle \zeta_j^3 - \zeta_i \|\boldsymbol{\zeta}\|_4^4 + \langle \boldsymbol{a}_i, \bar{\boldsymbol{\delta}}_{\mathrm{grad}} \rangle$$

$$= \zeta_i^3 - \zeta_i \underbrace{\frac{\|\boldsymbol{\zeta}\|_4^4}{\|\boldsymbol{a}_i\|_2^2}}_{\alpha_i} + \underbrace{\frac{\sum_{j\neq i}\langle \boldsymbol{a}_i, \boldsymbol{a}_j\rangle \zeta_j^3 + \langle \boldsymbol{a}_i, \bar{\boldsymbol{\delta}}_{\mathrm{grad}}\rangle}{\|\boldsymbol{a}_i\|_2^2}}_{\beta_i'} \tag{6.7}$$

with $\boldsymbol{\zeta} = \boldsymbol{A}^T\boldsymbol{q}$. Under the assumption that

$$\left\| \mathrm{grad}[\psi](\boldsymbol{q}) - \frac{3(1-\theta)}{\theta m^2}\mathrm{grad}[\varphi](\boldsymbol{q}) \right\|_2 \leq \frac{3c_\star}{2\kappa^2}\frac{1-\theta}{\theta m^2}\|\boldsymbol{\zeta}\|_4^6, \tag{6.8}$$

the perturbed part can be bounded via

$$\left|\langle \boldsymbol{a}_i, \bar{\boldsymbol{\delta}}_{\mathrm{grad}}\rangle\right| \leq \|\boldsymbol{a}_i\|_2 \|\bar{\boldsymbol{\delta}}_{\mathrm{grad}}\|_2 \leq \frac{c_\star}{2\kappa^2}\|\boldsymbol{a}_i\|_2\|\boldsymbol{\zeta}\|_4^6, \tag{6.9}$$

and also

$$\frac{\beta_i'}{\alpha_i^{3/2}} \leq \frac{\mu\|\boldsymbol{\zeta}\|_3^3 + \frac{1}{2}c_\star\kappa^{-2}\|\boldsymbol{\zeta}\|_4^6}{\|\boldsymbol{\zeta}\|_4^6} \leq c_\star\kappa^{-2} \leq \frac{1}{4}. \tag{6.10}$$

Then by Lemma 2.2, at every stationary point $\bar{\boldsymbol{q}}$, the $i$-th entry of $\boldsymbol{\zeta}$ resides in the set $\bigcup_{x\in\{0,\pm\sqrt{\alpha_i}\}}[x-\frac{2\beta_i'}{\alpha_i}, x+\frac{2\beta_i'}{\alpha_i}]$ – i.e., $\boldsymbol{\zeta}$ is nearly a trinary vector.

Moreover, we can characterize the curvature of critical points in terms of the number of large entries of $\boldsymbol{\zeta}$. Indeed, whenever $\boldsymbol{\zeta}$ has at least two entries in

$$\bigcup_{x\in\{\pm\sqrt{\alpha_i}\}}\left[x-\frac{2\beta_i'}{\alpha_i}, x+\frac{2\beta_i'}{\alpha_i}\right],$$

using (3.24), there exists a direction of strict negative curvature, provided

$$\mathrm{Hess}[\psi](\boldsymbol{q}) \prec \frac{3(1-\theta)}{\theta m^2}\mathrm{Hess}[\varphi](\boldsymbol{q})$$
$$+ 3(2-11c_\star)\frac{1-\theta}{\theta m^2}\|\boldsymbol{\zeta}\|_4^4\boldsymbol{I}. \tag{6.11}$$

Similarly, whenever $\boldsymbol{\zeta}$ has only one entry in

$$\bigcup_{x\in\{\pm\sqrt{\alpha_i}\}}\left[x-\frac{2\beta_i'}{\alpha_i}, x+\frac{2\beta_i'}{\alpha_i}\right],$$

using (3.10), we have that $\mathrm{Hess}[\psi](\boldsymbol{q}) \succ \boldsymbol{0}$, provided

$$\mathrm{Hess}[\psi](\boldsymbol{q}) \succ \frac{3(1-\theta)}{\theta m^2}\mathrm{Hess}[\varphi](\boldsymbol{q})$$
$$-3\left(1-6c_\star-36c_\star^2-24c_\star^3\right)\frac{1-\theta}{\theta m^2}\|\boldsymbol{\zeta}\|_4^4\boldsymbol{I}. \tag{6.12}$$

When $C_\star \geq 10$ and $c_\star \leq 0.1$, we have $2-11c_\star > 1-6c_\star-36c_\star^2-24c_\star^3 \geq 0.016$, and so above characterization obtains. ∎

**Theorem 6.2 (Main Result)** *Assume the observation $\boldsymbol{y} \in \mathbb{R}^m$ is the cyclic convolution of $\boldsymbol{a}_0 \in \mathbb{R}^k$ and $\boldsymbol{x}_0 \sim_{\mathrm{i.i.d.}}$ BG$(\theta) \in \mathbb{R}^m$, where the convolution matrix $\boldsymbol{A}_0 \in \mathbb{R}^{k\times(2k-1)}$ has minimum singular value $\sigma_{\min} > 0$ and condition number $\kappa \geq 1$, and $\boldsymbol{A}$ has column incoherence $\mu$. If*

$$m \geq C\frac{\min\left\{(2C_\star\mu)^{-1}, \kappa^2k^2\right\}}{(1-\theta)^2\sigma_{\min}^2}\kappa^8k^4\log^3(\kappa k) \tag{6.13}$$

*and $\theta \geq \log k / k$, then with probability no smaller than $1 - \exp(-k) - \theta^2 (1-\theta)^2 k^{-4} - 2\exp(-\theta k) - 48k^{-7} - 48m^{-5} - 24k \exp\left(-\frac{1}{144}\min\left\{k, 3\sqrt{\theta m}\right\}\right)$, any local minimum $\bar{q}$ of $\psi$ in $\hat{\mathcal{R}}_{2C_\star}$ satisfies $|\langle \bar{q}, \mathcal{P}_{\mathbb{S}}[a_\tau]\rangle| \geq 1 - c_\star \kappa^{-2}$ for some integer $\tau$.*

**Proof** From the concentration analysis for the Riemannian gradient (Lemma 7.1) and Hessian (Lemma 8.1), if

$$m \geq C \frac{\min\left\{(2C_\star\mu)^{-1}, \kappa^2 k^2\right\}}{(1-\theta)^2 \sigma_{\min}^2} \kappa^8 k^4 \log^3(\kappa k), \tag{6.14}$$

then with probability no smaller than $1 - \exp(-k) - \theta^2 (1-\theta)^2 k^{-4} - 2\exp(-\theta k) - 24k\exp\left(-\frac{1}{144}\min\left\{k, 3\sqrt{\theta m}\right\}\right) - 48k^{-7} - 48m^{-5}$,

$$\left\|\mathrm{grad}[\psi](q) - \frac{3(1-\theta)}{\theta m^2}\mathrm{grad}[\varphi](q)\right\|_2$$
$$\leq \frac{3c_\star}{2\kappa^2}\frac{1-\theta}{\theta m^2}\|A^T q\|_4^6, \tag{6.15}$$
$$\left\|\mathrm{Hess}[\psi](q) - \frac{3(1-\theta)}{\theta m^2}\mathrm{Hess}[\varphi](q)\right\|_2$$
$$\leq 3\left(1 - 6c_\star - 36c_\star^2 - 24c_\star^3\right)\frac{1-\theta}{\theta m^2}\|A^T q\|_4^4. \tag{6.16}$$

hold for all $q \in \hat{\mathcal{R}}_{2C_\star}$ with $C_\star \geq 10$ and $c_\star = 1/C_\star$. Therefore, by Lemma 6.1 any local minimum $\bar{q}$ of $\psi(q)$ in $\mathcal{R}_{2C_\star}$ satisfies $|\langle \bar{q}, \mathcal{P}_{\mathbb{S}}[a_l]\rangle| \geq 1 - 2c_\star \kappa^{-2}$ for some index $l$. ∎

## 6.2 Proof of the Main Corollary

**Corollary 6.3** *Suppose the ground truth kernel $a_0$ has induces coherence $0 \leq \mu \leq \frac{1}{8 \times 48}\log^{-3/2}(k)$ and sparse coefficient $x_0 \sim_{\text{i.i.d.}} \mathrm{BG}(\theta) \in \mathbb{R}^m$. there exist positive constants $C \geq 2560^4$ and $C'$ such that whenever the sparsity level*

$$64k^{-1}\log k \leq \theta \leq \min\left\{\tfrac{1}{48^2}\mu^{-2}k^{-1}\log^{-2}k, \tag{6.17}\right.$$
$$\left.\left(\tfrac{1}{4} - \tfrac{640}{C^{1/4}}\right)\left(3C_\star\mu\kappa^2\right)^{-2/3}k^{-1}\left(1 + 36\mu^2 k\log k\right)^{-2}\right\},$$

*and signal length*

$$m \geq \max\left\{C\theta^2\sigma_{\min}^{-2}\kappa^6 k^3\left(1 + 36\mu^2 k\log k\right)^4\log(\kappa k), \tag{6.18}\right.$$
$$\left.C'(1-\theta)^{-2}\sigma_{\min}^{-2}\min\left\{\mu^{-1}, \kappa^2 k^2\right\}\kappa^8 k^4\log^3(\kappa k)\right\},$$

*then Algorithm 1 recovers $\bar{a}$ such that*

$$\|\bar{a} \pm \mathcal{P}_{\mathbb{S}}[\iota_k s_\tau[\widetilde{a_0}]]\|_2 \leq 4\sqrt{c_\star} + ck^{-1} \tag{6.19}$$

*for some integer shift $\tau \in [-(k-1), k-1]$ with probability no smaller than $1 - k^{-1} - 8k^{-2} - \exp(-k) - \theta^2(1-\theta)^2 k^{-4} - 2\exp(-\theta k) - 24k\exp\left(-\frac{1}{144}\min\left\{k, 3\sqrt{\theta m}\right\}\right) - 48k^{-7} - 48m^{-5}$.*

**Proof** From the concentration results for the Riemannian gradient, at every point $q \in \hat{\mathcal{R}}_{2C_\star}$, the objective value of $\psi(q)$ satisfies

$$\left|\psi(q) - \frac{3(1-\theta)}{\theta m^2}\varphi(q) + \frac{3}{4m^2}\right|$$

$$
\leq \left| \frac{\left\| \boldsymbol{Y}^T \left(\boldsymbol{Y}\boldsymbol{Y}^T\right)^{-1/2}\boldsymbol{q} \right\|_4^4}{4m} - \frac{3\left(1-\theta\right)\left\|\boldsymbol{\zeta}\right\|_4^4}{4\theta m^2} - \frac{3}{4m^2} \right| \tag{6.20}
$$

$$
\leq \left| \left\langle \boldsymbol{q}, \frac{\left(\boldsymbol{Y}\boldsymbol{Y}^T\right)^{-1/2}\boldsymbol{Y}\boldsymbol{\eta}^{\circ 3}}{4m} - \frac{3\left(1-\theta\right)}{4\theta m^2}\boldsymbol{A}\boldsymbol{\zeta}^{\circ 3} - \frac{3}{4m^2}\boldsymbol{q} \right\rangle \right| \tag{6.21}
$$

$$
\leq \left\| \frac{\left(\boldsymbol{Y}\boldsymbol{Y}^T\right)^{-1/2}\boldsymbol{Y}\boldsymbol{\eta}^{\circ 3}}{4m} - \frac{3\left(1-\theta\right)}{4\theta m^2}\boldsymbol{A}\boldsymbol{\zeta}^{\circ 3} - \frac{3}{4m^2}\boldsymbol{q} \right\|_2 \tag{6.22}
$$

$$
\leq \frac{1}{4m}\left\| \left(\boldsymbol{Y}\boldsymbol{Y}^T\right)^{-1/2}\boldsymbol{Y}\boldsymbol{\eta}^{\circ 3} - \left(\theta m \boldsymbol{A}_0\boldsymbol{A}_0^T\right)^{-1/2}\boldsymbol{Y}\boldsymbol{\eta}^{\circ 3} \right\|_2
$$
$$
+ \frac{1}{4\theta^{1/2}m^{3/2}}\left\| \left(\boldsymbol{A}_0\boldsymbol{A}_0^T\right)^{-1/2}\boldsymbol{Y}\left(\boldsymbol{\eta}^{\circ 3} - \bar{\boldsymbol{\eta}}^{\circ 3}\right) \right\|_2
$$
$$
+ \left\| \frac{1}{4\theta^{1/2}m^{3/2}}\left(\boldsymbol{A}_0\boldsymbol{A}_0^T\right)^{-1/2}\boldsymbol{Y}\bar{\boldsymbol{\eta}}^{\circ 3} \right.
$$
$$
\left. - \frac{3\left(1-\theta\right)}{4\theta m^2}\boldsymbol{A}\boldsymbol{\zeta}^{\circ 3} - \frac{3}{4m^2}\boldsymbol{q} \right\|_2 \tag{6.23}
$$

$$
\leq \frac{3c_\star}{8\kappa^2}\frac{1-\theta}{\theta m^2}\min_{\boldsymbol{q}\in\hat{\mathcal{R}}_{2C_\star}}\left\|\boldsymbol{A}^T\boldsymbol{q}\right\|_4^6 \tag{6.24}
$$

with probability no smaller than $1 - 2\exp\left(-\theta k\right) - 24k\exp\left(-\frac{1}{144}\min\left\{k, 3\sqrt{\theta m}\right\}\right) - 48k^{-7} - 48m^{-5}$. The last inequality is derived with similar arguments in Lemma 7.1, for simplicity, we do not present them here. Moreover, with Lemma 4.1, we can obtain an initialization point $\boldsymbol{q}_{\text{init}}$ such that

$$
\left\|\boldsymbol{A}^T\boldsymbol{q}_{\text{init}}\right\|_4^4 \geq \left(3C_\star\mu\kappa^2\right)^{2/3} \tag{6.25}
$$
$$
\geq \left(2C_\star\mu\kappa^2\right)^{2/3} + \mu/2. \tag{6.26}
$$

Consider any descent method for $\psi$, which generates a sequence of iterates $\boldsymbol{q}^{(0)} = \boldsymbol{q}_{\text{init}}, \boldsymbol{q}^{(1)}, \ldots, \boldsymbol{q}^{(k)}, \ldots$ such that $\psi(\boldsymbol{q}^{(k)})$ is non-increasing with $k$. Then

$$
\psi\left(\boldsymbol{q}^{(k)}\right) \leq \psi\left(\boldsymbol{q}_{\text{init}}\right) \tag{6.27}
$$
$$
\leq \frac{3\left(1-\theta\right)}{\theta m^2}\varphi\left(\boldsymbol{q}_{\text{init}}\right) + \frac{3}{4m^2}
$$
$$
+ \frac{3c_\star}{8\kappa^2}\frac{1-\theta}{\theta m^2}\min_{\boldsymbol{q}\in\hat{\mathcal{R}}_{2C_\star}}\left\|\boldsymbol{A}^T\boldsymbol{q}\right\|_4^6. \tag{6.28}
$$

On the other hand, the finite sample objective function value $\psi$ is close to that of $\frac{3(1-\theta)}{\theta m^2}\varphi\left(\boldsymbol{q}\right) - \frac{3}{4m^2}$,

$$
\frac{3\left(1-\theta\right)}{\theta m^2}\varphi\left(\boldsymbol{q}^{(k)}\right)
$$
$$
\leq \psi\left(\boldsymbol{q}^{(k)}\right) + \frac{3}{4m^2} + \frac{3c_\star}{8\kappa^2}\frac{1-\theta}{\theta m^2}\min_{\boldsymbol{q}\in\hat{\mathcal{R}}_{2C_\star}}\left\|\boldsymbol{A}^T\boldsymbol{q}\right\|_4^6 \tag{6.29}
$$
$$
\leq \frac{3\left(1-\theta\right)}{\theta m^2}\varphi\left(\boldsymbol{q}_{\text{init}}\right) + \frac{3c_\star}{4\kappa^2}\frac{1-\theta}{\theta m^2}\min_{\boldsymbol{q}\in\hat{\mathcal{R}}_{2C_\star}}\left\|\boldsymbol{A}^T\boldsymbol{q}\right\|_4^6, \tag{6.30}
$$

Therefore, we obtain that

$$
\varphi\left(\boldsymbol{q}^{(k)}\right) \leq \varphi\left(\boldsymbol{q}_{\text{init}}\right) + \frac{\mu}{2} \tag{6.31}
$$

$$\leq \varphi\left(\boldsymbol{q}_{\mathrm{init}}\right) + \frac{c_{\star}}{4\kappa^2} \min_{\boldsymbol{q}\in\hat{\mathcal{R}}_{2C_{\star}}} \left\|\boldsymbol{A}^T\boldsymbol{q}\right\|_4^6, \tag{6.32}$$

which implies that $\boldsymbol{q}^{(k)} \in \hat{\mathcal{R}}_{2C_{\star}}$ always holds. At last, Theorem 6.2 says that any local minimum $\bar{\boldsymbol{q}}$ is close to $\pm\boldsymbol{a}_i$ for some $i$, in the sense that

$$\left|\langle \bar{\boldsymbol{q}}, \mathcal{P}_{\mathbb{S}}\left[\boldsymbol{a}_i\right]\rangle\right| \geq 1 - c_{\star}\kappa^{-2}. \tag{6.33}$$

Write $\frac{1}{\theta m}\boldsymbol{Y}\boldsymbol{Y}^T = \boldsymbol{A}_0\left(\boldsymbol{I}+\boldsymbol{\Delta}\right)\boldsymbol{A}_0^T$ with $\|\boldsymbol{\Delta}\|_2 \leq \delta$, and let

$$\bar{\boldsymbol{q}} = \pm\frac{\boldsymbol{a}_i}{\|\boldsymbol{a}_i\|_2} + \sqrt{2\left(1-\left|\left\langle\bar{\boldsymbol{q}},\frac{\boldsymbol{a}_i}{\|\boldsymbol{a}_i\|_2}\right\rangle\right|\right)}\boldsymbol{\delta}, \tag{6.34}$$

with $\|\boldsymbol{\delta}\|_2 = 1$. Since

$$\boldsymbol{a}_i = \left(\boldsymbol{A}_0\boldsymbol{A}_0^T\right)^{-1/2}\boldsymbol{\iota}_k^* s_{-(k-i)}[\widetilde{\boldsymbol{a}_0}], \tag{6.35}$$

we have

$$\left(\frac{\boldsymbol{Y}\boldsymbol{Y}^T}{\theta m}\right)^{1/2}\bar{\boldsymbol{q}}$$

$$= \pm\left(\frac{\boldsymbol{Y}\boldsymbol{Y}^T}{\theta m}\right)^{1/2}\left[\frac{\boldsymbol{a}_i}{\|\boldsymbol{a}_i\|_2}+\sqrt{2\left(1-\left|\left\langle\bar{\boldsymbol{q}},\frac{\boldsymbol{a}_i}{\|\boldsymbol{a}_i\|_2}\right\rangle\right|\right)}\boldsymbol{\delta}\right] \tag{6.36}$$

$$= \pm\left(\frac{\boldsymbol{Y}\boldsymbol{Y}^T}{\theta m}\right)^{1/2}\left(\boldsymbol{A}_0\boldsymbol{A}_0^T\right)^{-1/2}\frac{\boldsymbol{\iota}_k^* s_{-(k-i)}[\widetilde{\boldsymbol{a}_0}]}{\|\boldsymbol{a}_i\|_2}$$

$$+ \sqrt{2\left(1-\left|\left\langle\bar{\boldsymbol{q}},\frac{\boldsymbol{a}_i}{\|\boldsymbol{a}_i\|_2}\right\rangle\right|\right)}\left(\frac{\boldsymbol{Y}\boldsymbol{Y}^T}{\theta m}\right)^{1/2}\boldsymbol{\delta} \tag{6.37}$$

therefore the error can be bounded as

$$\left\|\left(\frac{\boldsymbol{Y}\boldsymbol{Y}^T}{\theta m}\right)^{1/2}\bar{\boldsymbol{q}} \pm \frac{\boldsymbol{\iota}_k^* s_{-(k-i)}[\widetilde{\boldsymbol{a}_0}]}{\|\boldsymbol{a}_i\|_2}\right\|_2$$

$$\leq \left\|\left(\frac{\boldsymbol{Y}\boldsymbol{Y}^T}{\theta m}\right)^{1/2}\left(\boldsymbol{A}_0\boldsymbol{A}_0^T\right)^{-1/2}-\boldsymbol{I}\right\|_2\left\|\frac{\boldsymbol{\iota}_k^* s_{-(k-i)}[\widetilde{\boldsymbol{a}_0}]}{\|\boldsymbol{a}_i\|_2}\right\|_2$$

$$+ \sqrt{2\left(1-\left|\left\langle\bar{\boldsymbol{q}},\frac{\boldsymbol{a}_i}{\|\boldsymbol{a}_i\|_2}\right\rangle\right|\right)}\left\|\left(\frac{\boldsymbol{Y}\boldsymbol{Y}^T}{\theta m}\right)^{1/2}\right\|_2. \tag{6.38}$$

Finally, using the fact that for any nonzero vectors $\boldsymbol{u}$ and $\boldsymbol{v}$ that $\langle\boldsymbol{u},\boldsymbol{v}\rangle \geq 0$,

$$\left\|\frac{\boldsymbol{u}}{\|\boldsymbol{u}\|_2}-\frac{\boldsymbol{v}}{\|\boldsymbol{v}\|_2}\right\|_2 \leq \frac{\sqrt{2}}{\|\boldsymbol{v}\|_2}\|\boldsymbol{u}-\boldsymbol{v}\|_2 \tag{6.39}$$

always holds. Therefore,

$$\left\|\bar{\boldsymbol{a}} \pm \mathcal{P}_{\mathbb{S}}\left[\boldsymbol{\iota}_k s_i[\widetilde{\boldsymbol{a}_0}]\right]\right\|_2$$

$$= \left\|\mathcal{P}_{\mathbb{S}}\left[\left(\boldsymbol{Y}\boldsymbol{Y}^T\right)^{1/2}\bar{\boldsymbol{q}}\right] \pm \mathcal{P}_{\mathbb{S}}\left[\boldsymbol{\iota}_k s_i[\widetilde{\boldsymbol{a}_0}]\right]\right\|_2 \tag{6.40}$$

$$\leq \frac{\sqrt{2}\|\boldsymbol{a}_i\|_2}{\|\boldsymbol{\iota}_k^* s_{i-k}[\widetilde{\boldsymbol{a}_0}]\|_2}\left\|\left(\frac{\boldsymbol{Y}\boldsymbol{Y}^T}{\theta m}\right)^{1/2}\bar{\boldsymbol{q}} \pm \frac{\boldsymbol{\iota}_k^* s_{i-k}[\widetilde{\boldsymbol{a}_0}]}{\|\boldsymbol{a}_i\|_2}\right\|_2 \tag{6.41}$$

$$\leq \kappa \sqrt{2\left(1+\delta\right)\left(1-\left|\left\langle \bar{\boldsymbol{q}}, \frac{\boldsymbol{a}_i}{\|\boldsymbol{a}_i\|_2}\right\rangle\right|\right)} \tag{6.42}$$

$$+\sqrt{2}\kappa \left\|\left(\frac{\boldsymbol{Y}\boldsymbol{Y}^T}{\theta m}\right)^{1/2}\left(\boldsymbol{A}_0\boldsymbol{A}_0^T\right)^{-1/2} - \boldsymbol{I}\right\|_2$$

$$\leq 2\kappa \sqrt{2\left(1-\left|\left\langle \bar{\boldsymbol{q}}, \frac{\boldsymbol{a}_i}{\|\boldsymbol{a}_i\|_2}\right\rangle\right|\right)} + \sqrt{2}\kappa^3 \delta/\sigma_{\min} \tag{6.43}$$

$$\text{(Lemma 5.2)}$$

$$\leq 4\sqrt{c_\star} + 10\sqrt{2}\kappa^3 \sigma_{\min}^{-1}\sqrt{k\log m/m} \tag{6.44}$$

$$\leq 4\sqrt{c_\star} + ck^{-1}, \tag{6.45}$$

completing the proof. ∎

# 7 Concentration for Gradient (Lemma 7.1)

**Lemma 7.1** *Suppose $\boldsymbol{x}_0 \sim_{\text{i.i.d.}} \mathrm{BG}\left(\theta\right)$. There exists a positive constant $C$ such that whenever*

$$m \geq C \frac{\min\left\{\left(2C_\star\mu\right)^{-1}, \kappa^2 k^2\right\}}{\left(1-\theta\right)^2 \sigma_{\min}^2} \kappa^8 k^4 \log^3\left(\frac{\kappa k}{\left(1-\theta\right)\sigma_{\min}}\right) \tag{7.1}$$

*and $\theta > \log k/k$, then with probability no smaller than $1 - c_1 \exp\left(-k\right) - c_2 k^{-4} - 2\exp\left(-\theta k\right) - 24k\exp\left(-\frac{1}{144}\min\left\{k, 3\sqrt{\theta m}\right\}\right) - 48k^{-7} - 48m^{-5}$,*

$$\left\|\mathrm{grad}[\psi]\left(\boldsymbol{q}\right) - \frac{3\left(1-\theta\right)}{\theta m^2}\mathrm{grad}[\varphi]\left(\boldsymbol{q}\right)\right\|_2 \leq c\frac{1-\theta}{\theta m^2}\frac{\left\|\boldsymbol{A}^T\boldsymbol{q}\right\|_4^6}{\kappa^2}, \tag{7.2}$$

*holds for all $\boldsymbol{q} \in \hat{\mathcal{R}}_{2C_\star}$ with positive constant $c \leq 3/\left(2C_\star\right)$.*

**Proof** Denote $\boldsymbol{\eta} = \boldsymbol{Y}^T\left(\boldsymbol{Y}\boldsymbol{Y}^T\right)^{-1/2}\boldsymbol{q}$ and $\bar{\boldsymbol{\eta}} = \boldsymbol{Y}^T\left(\theta m\boldsymbol{A}_0\boldsymbol{A}_0^T\right)^{-1/2}\boldsymbol{q} = \left(\theta m\right)^{-1/2}\boldsymbol{X}_0^T\boldsymbol{\zeta}$, then

$$\left\|\mathrm{grad}\left[\psi\right]\left(\boldsymbol{q}\right) - \frac{3\left(1-\theta\right)}{\theta m^2}\mathrm{grad}\left[\varphi\right]\left(\boldsymbol{q}\right)\right\|_2$$

$$= \left\|\boldsymbol{P}_{\boldsymbol{q}^\perp}\left[\frac{1}{m}\left(\boldsymbol{Y}\boldsymbol{Y}^T\right)^{-1/2}\boldsymbol{Y}\boldsymbol{\eta}^{\circ 3} - \frac{3\left(1-\theta\right)}{\theta m^2}\boldsymbol{A}\boldsymbol{\zeta}^{\circ 3}\right]\right\|_2$$

$$\leq \underbrace{\frac{1}{m}\left\|\left(\boldsymbol{Y}\boldsymbol{Y}^T\right)^{-1/2}\boldsymbol{Y}\boldsymbol{\eta}^{\circ 3} - \left(\theta m\right)^{-1/2}\boldsymbol{A}\boldsymbol{X}_0\boldsymbol{\eta}^{\circ 3}\right\|_2}_{\Delta_1^g}$$

$$+ \underbrace{\frac{1}{\theta^{1/2}m^{3/2}}\left\|\boldsymbol{A}\boldsymbol{X}_0\boldsymbol{\eta}^{\circ 3} - \boldsymbol{A}\boldsymbol{X}_0\bar{\boldsymbol{\eta}}^{\circ 3}\right\|_2}_{\Delta_2^g}$$

$$+ \underbrace{\left\|\boldsymbol{P}_{\boldsymbol{q}^\perp}\left[\frac{1}{\theta^{1/2}m^{3/2}}\boldsymbol{A}\boldsymbol{X}_0\bar{\boldsymbol{\eta}}^{\circ 3} - \frac{3\left(1-\theta\right)}{\theta m^2}\boldsymbol{A}\boldsymbol{\zeta}^{\circ 3}\right]\right\|_2}_{\Delta_3^g}.$$

First, let us note that

$$C\left(1-\theta\right)^{-2}\sigma_{\min}^{-2}\kappa^{10}k^6 \log^3\left(\frac{\kappa k}{\left(1-\theta\right)\sigma_{\min}}\right)$$

$$\leq C\left(\frac{\kappa k}{\sigma_{\min}\left(1-\theta\right)}\right)^{10}\log^3\left(\frac{\kappa k}{\left(1-\theta\right)\sigma_{\min}}\right) \tag{7.3}$$

$$\leq C\left(\frac{\kappa k}{\left(1-\theta\right)\sigma_{\min}}\right)^{13}, \tag{7.4}$$

hence

$$\frac{\log^3\left(C\left(1-\theta\right)^{-2}\sigma_{\min}^{-2}\kappa^{10}k^6\log^3\left(\left(1-\theta\right)^{-1}\sigma_{\min}^{-1}\kappa k\right)\right)}{C\log^3\left(\left(1-\theta\right)^{-1}\sigma_{\min}^{-1}\kappa k\right)}$$

$$\leq\left(\frac{\log C+13\log\left(\left(1-\theta\right)^{-1}\sigma_{\min}^{-1}\kappa k\right)}{C^{1/3}\log\left(\left(1-\theta\right)^{-1}\sigma_{\min}^{-1}\kappa k\right)}\right)^3 \tag{7.5}$$

$$\leq\left(\frac{\log C}{C^{1/3}\log\left(\left(1-\theta\right)^{-1}\sigma_{\min}^{-1}\kappa k\right)}+\frac{13}{C^{1/3}}\right)^3 \tag{7.6}$$

$$\leq\left(\frac{1}{C^{1/6}}+\frac{1}{2}\frac{1}{C^{1/6}}\right)^3\qquad\left(C\geq10^8\right) \tag{7.7}$$

$$\leq\frac{4}{C^{1/2}}. \tag{7.8}$$

Given

$$m\geq C\frac{\min\left\{\left(2C_\star\mu\right)^{-1},\kappa^2k^2\right\}}{\left(1-\theta\right)^2\sigma_{\min}^2}\kappa^8k^4\log^3\left(\frac{\kappa k}{\sigma_{\min}\left(1-\theta\right)}\right), \tag{7.9}$$

as the ratio $\log^3 m/m$ decreases with increasing $m$, then

$$\frac{\log^3 m}{m}\leq\frac{\log^3\left(\frac{C\kappa^{10}k^6}{(1-\theta)^2\sigma_{\min}^2}\log^3\left(\frac{\kappa k}{(1-\theta)\sigma_{\min}}\right)\right)}{C\log^3\left(\frac{\kappa k}{(1-\theta)\sigma_{\min}}\right)}$$

$$\times\frac{\left(1-\theta\right)^2\sigma_{\min}^2}{\min\left\{\left(2C_\star\mu\right)^{-1},\kappa^2k^2\right\}\kappa^8k^4} \tag{7.10}$$

$$\leq\frac{4}{C^{1/2}}\frac{\left(1-\theta\right)^2\sigma_{\min}^2}{\min\left\{\left(2C_\star\mu\right)^{-1},\kappa^2k^2\right\}\kappa^8k^4} \tag{7.11}$$

According to Lemma 5.1, following inequality always holds

$$\left\|\frac{1}{\theta m}\boldsymbol{X}_0\boldsymbol{X}_0^T-\boldsymbol{I}\right\|_2\leq\delta \tag{7.12}$$

$$\leq 10\sqrt{k\log m/m} \tag{7.13}$$

$$\leq\frac{20\left(1-\theta\right)\sigma_{\min}\max\left\{\left(2C_\star\mu\right)^{1/2},\left(\kappa k\right)^{-1}\right\}}{C^{1/4}\kappa^4k^{3/2}\log m} \tag{7.14}$$

$$\leq\frac{20\sigma_{\min}}{C^{1/4}\kappa^3}\frac{\left(1-\theta\right)\left\|\boldsymbol{A}^T\boldsymbol{q}\right\|_4^6}{\kappa^2k\log m},\qquad\forall\boldsymbol{q}\in\hat{\mathcal{R}}_{2C_\star}. \tag{7.15}$$

with probability no smaller than $1-\varepsilon_0$ with $\varepsilon_0=2\exp\left(-\theta k\right)+24k\exp\left(-\frac{1}{144}\min\left\{k,3\sqrt{\theta m}\right\}\right)+48k^{-7}+48m^{-5}$.

Moreover, $4\kappa^3\delta/\sigma_{\min} \le 1/2$ whenever

$$C \ge \left(\frac{160\,(1-\theta)}{k\log m}\right)^4,\tag{7.16}$$

whence $\delta \le 1/\left(8\kappa^2\right)$, and Lemma 5.3 implies that

$$\left\|\left(\frac{1}{\theta m}\boldsymbol{Y}\boldsymbol{Y}^T\right)^{-1/2}\boldsymbol{A}_0 - \left(\boldsymbol{A}_0\boldsymbol{A}_0^T\right)^{-1/2}\boldsymbol{A}_0\right\|_2$$

$$\le 4\kappa^3\delta/\sigma_{\min}\tag{7.17}$$

$$\le \frac{80\,(1-\theta)}{C^{1/4}k\log m}\frac{\left\|\boldsymbol{A}^T\boldsymbol{q}\right\|_4^6}{\kappa^2}, \qquad \forall \boldsymbol{q} \in \hat{\mathcal{R}}_{2C_\star}.\tag{7.18}$$

At the same time,

$$\|\boldsymbol{X}_0\|_2 \le (\theta m)^{1/2}\sqrt{1+\delta} \le (\theta m)^{1/2}\,(1+\delta/2)\,.\tag{7.19}$$

Moreover, Lemma 2.5 implies that with probability no smaller than $1 - \varepsilon_B$, we have

$$\|\boldsymbol{x}_0\|_\infty \le \sqrt{2}\log^{1/2}\left(\frac{2\theta m}{\varepsilon_B}\right).\tag{7.20}$$

**Upper Bound for $\Delta_1^g$.** Using Lemma 2.7, on the an event of probability at least $1 - \varepsilon_0 - \varepsilon_B$,

$$\left\|\boldsymbol{\eta}^{\circ 3}\right\|_2 = \|\boldsymbol{\eta}\|_6^3\tag{7.21}$$

$$\le \left(1 + \frac{4\kappa^3\delta}{\sigma_{\min}}\right)^2\frac{2k}{\theta m}\|\boldsymbol{x}_0\|_\infty^2\tag{7.22}$$

$$\le \frac{9k}{\theta m}\log\left(\frac{2\theta m}{\varepsilon_B}\right).\tag{7.23}$$

Therefore, we can obtain following upper bound

$$\Delta_1^g = \frac{1}{m}\left\|\left(\boldsymbol{Y}\boldsymbol{Y}^T\right)^{-1/2}\boldsymbol{Y}\boldsymbol{\eta}^{\circ 3} - (\theta m)^{-1/2}\boldsymbol{A}\boldsymbol{X}_0\boldsymbol{\eta}^{\circ 3}\right\|_2\tag{7.24}$$

$$\le \frac{1}{\theta^{1/2}m^{3/2}}\|\boldsymbol{X}_0\|_2\left\|\boldsymbol{\eta}^{\circ 3}\right\|_2 \times$$

$$\left\|\left(\frac{1}{\theta m}\boldsymbol{Y}\boldsymbol{Y}^T\right)^{-1/2}\boldsymbol{A}_0 - \boldsymbol{A}\right\|_2\tag{7.25}$$

$$\le \frac{5}{4m}\cdot\frac{4\kappa^3\delta}{\sigma_{\min}}\cdot\frac{9k}{\theta m}\log\left(\frac{2\theta m}{\varepsilon_B}\right)\tag{7.26}$$

$$\le \frac{900\,(1-\theta)\log\left(2\theta m/\varepsilon_B\right)}{C^{1/4}\theta m^2\log m}\frac{\left\|\boldsymbol{A}^T\boldsymbol{q}\right\|_4^6}{\kappa^2} \quad \forall \boldsymbol{q} \in \hat{\mathcal{R}}_{2C_\star}.$$

**Upper Bound for $\Delta_2^g$.** Similarly, with probability no smaller than $1 - \varepsilon_0 - \varepsilon_B$, together with Lemma 2.7, following upper bound can be obtained

$$\left\|\boldsymbol{\eta}^{\circ 3} - \bar{\boldsymbol{\eta}}^{\circ 3}\right\|_2$$

$$= \left\|\boldsymbol{\eta}^{\circ 3} - \operatorname{diag}\left(\boldsymbol{\eta}^{\circ 2}\right)\bar{\boldsymbol{\eta}} + \operatorname{diag}\left(\boldsymbol{\eta}^{\circ 2}\right)\bar{\boldsymbol{\eta}} - \bar{\boldsymbol{\eta}}^{\circ 3}\right\|_2\tag{7.27}$$

$$\le \|\boldsymbol{\eta} - \bar{\boldsymbol{\eta}}\|_2\left\|\operatorname{diag}\left(\boldsymbol{\eta}^{\circ 2}\right)\right\|_2 + \|\bar{\boldsymbol{\eta}}\|_2\left\|\operatorname{diag}\left(\boldsymbol{\eta}^{\circ 2} - \bar{\boldsymbol{\eta}}^{\circ 2}\right)\right\|_2\tag{7.28}$$

$$= \|\boldsymbol{\eta} - \bar{\boldsymbol{\eta}}\|_2\|\boldsymbol{\eta}\|_\infty^2 + \|\bar{\boldsymbol{\eta}}\|_2\left\|\boldsymbol{\eta}^{\circ 2} - \bar{\boldsymbol{\eta}}^{\circ 2}\right\|_\infty\tag{7.29}$$

$$\leq \|\boldsymbol{\eta} - \bar{\boldsymbol{\eta}}\|_2 \|\boldsymbol{\eta}\|_\infty^2 + \|\bar{\boldsymbol{\eta}}\|_2 \|\boldsymbol{\eta} - \bar{\boldsymbol{\eta}}\|_\infty \|\boldsymbol{\eta} + \bar{\boldsymbol{\eta}}\|_\infty \tag{7.30}$$

$$\leq 4\left(1 + \delta/2\right) \frac{4\kappa^3 \delta}{\sigma_{\min}} \frac{k}{\theta m} \log\left(2\theta m/\varepsilon_B\right) \times$$

$$\left[\left(1 + \frac{4\kappa^3 \delta}{\sigma_{\min}}\right)^2 + \left(2 + \frac{4\kappa^3 \delta}{\sigma_{\min}}\right)\right] \tag{7.31}$$

$$\leq \frac{24k}{\theta m} \log\left(2\theta m/\varepsilon_B\right) \cdot \frac{4\kappa^3 \delta}{\sigma_{\min}}. \tag{7.32}$$

Therefore, we can obtain following upper bound

$$\Delta_2^g = \frac{1}{\theta^{1/2} m^{3/2}} \left\| \boldsymbol{A} \boldsymbol{X}_0^T \boldsymbol{\eta}^{\circ 3} - \boldsymbol{A} \boldsymbol{X}_0^T \bar{\boldsymbol{\eta}}^{\circ 3} \right\|_2 \tag{7.33}$$

$$\leq \frac{1}{\theta^{1/2} m^{3/2}} \|\boldsymbol{A}\|_2 \|\boldsymbol{X}_0\|_2 \left\| \boldsymbol{\eta}^{\circ 3} - \bar{\boldsymbol{\eta}}^{\circ 3} \right\|_2 \tag{7.34}$$

$$\leq \frac{5}{4m} \cdot \frac{24k}{\theta m} \log\left(2\theta m/\varepsilon_B\right) \cdot \frac{4\kappa^3 \delta}{\sigma_{\min}} \tag{7.35}$$

$$\leq \frac{2400}{C^{1/4}} \frac{1 - \theta}{\theta m^2} \frac{\left\| \boldsymbol{A}^T \boldsymbol{q} \right\|_4^6}{\kappa^2} \cdot \frac{\log\left(2\theta m/\varepsilon_B\right)}{\log m}. \tag{7.36}$$

For both $\Delta_1^g$ and $\Delta_2^g$ to be bounded by $\frac{1}{2C_\star} \frac{1-\theta}{\theta m^2} \frac{\|\boldsymbol{A}^T \boldsymbol{q}\|_4^6}{\kappa^2}$, we set

$$C \geq \left(4800 C_\star \frac{\log\left(2\theta m/\varepsilon_B\right)}{\log m}\right)^4. \tag{7.37}$$

Notice that the right hand side is indeed bounded by a numerical constant for all $m$.

**Tail Bound for $\Delta_3^g$.** Note that

$$\left(\boldsymbol{A}_0 \boldsymbol{A}_0^T\right)^{-1/2} \boldsymbol{Y} \bar{\boldsymbol{\eta}}^{\circ 3}$$

$$= \left(\boldsymbol{A}_0 \boldsymbol{A}_0^T\right)^{-1/2} \boldsymbol{A}_0 \boldsymbol{X}_0 \left(\boldsymbol{Y}^T \left(\theta m \boldsymbol{A}_0 \boldsymbol{A}_0^T\right)^{-1/2} \bar{\boldsymbol{q}}\right)^{\circ 3} \tag{7.38}$$

$$= \left(\theta m\right)^{-3/2} \boldsymbol{A} \boldsymbol{X}_0 \left(\boldsymbol{X}_0^T \boldsymbol{A}^T \boldsymbol{q}\right)^{\circ 3}, \tag{7.39}$$

and its expectation with respect to $\boldsymbol{x}_0$

$$\mathbb{E}\left[\frac{1}{m} \boldsymbol{A} \boldsymbol{X}_0 \left(\boldsymbol{X}_0^T \boldsymbol{A}^T \boldsymbol{q}\right)^{\circ 3}\right]$$

$$= \mathbb{E}\left[\boldsymbol{A} \boldsymbol{x}_i \left(\boldsymbol{x}_i^T \boldsymbol{A}^T \boldsymbol{q}\right)^3\right] \tag{7.40}$$

$$= 3\theta\left(1 - \theta\right) \boldsymbol{A} \boldsymbol{\zeta}^{\circ 3} + 3\theta^2 \left\| \boldsymbol{A}^T \boldsymbol{q} \right\|_2^2 \boldsymbol{A} \boldsymbol{A}^T \boldsymbol{q} \tag{7.41}$$

$$= 3\theta\left(1 - \theta\right) \boldsymbol{A} \boldsymbol{\zeta}^{\circ 3} + 3\theta^2 \boldsymbol{q}, \tag{7.42}$$

hence

$$\boldsymbol{P}_{\boldsymbol{q}^\perp} \left[\mathbb{E}\left[\frac{1}{m} \boldsymbol{A} \boldsymbol{X}_0 \left(\boldsymbol{X}_0^T \boldsymbol{A}^T \boldsymbol{q}\right)^{\circ 3}\right]\right]$$

$$= \boldsymbol{P}_{\boldsymbol{q}^\perp} \left[3\theta\left(1 - \theta\right) \boldsymbol{A} \boldsymbol{\zeta}^{\circ 3}\right]. \tag{7.43}$$

Therefore, the $\Delta_3^g$ term can be simplified as

$$\Delta_3^g = \left\| \boldsymbol{P}_{\boldsymbol{q}^\perp} \left[\frac{1}{\theta^{1/2} m^{3/2}} \boldsymbol{A} \boldsymbol{X}_0^T \bar{\boldsymbol{\eta}}^{\circ 3} - \frac{3\left(1 - \theta\right)}{\theta m^2} \boldsymbol{A} \boldsymbol{\zeta}^{\circ 3}\right] \right\|_2 \tag{7.44}$$

$$= \frac{1}{\theta^2 m^2} \left\| \boldsymbol{P}_{\boldsymbol{q}^\perp} \left[ \frac{\boldsymbol{A} \boldsymbol{X}_0 \left( \boldsymbol{X}_0^T \boldsymbol{\zeta} \right)^{\circ 3}}{m} - 3\theta \left( 1 - \theta \right) \boldsymbol{A} \boldsymbol{\zeta}^{\circ 3} \right] \right\|_2 \tag{7.45}$$

$$\leq \frac{1}{\theta^2 m^2} \left\| \boldsymbol{P}_{\boldsymbol{q}^\perp} \left[ \frac{\boldsymbol{A} \boldsymbol{X}_0 \left( \boldsymbol{X}_0^T \boldsymbol{\zeta} \right)^{\circ 3}}{m} - \mathbb{E} \left[ \cdot \right] \right] \right\|_2$$

$$+ \frac{1}{\theta^2 m^2} \left\| \boldsymbol{P}_{\boldsymbol{q}^\perp} \left[ 3\theta^2 \boldsymbol{q} \right] \right\|_2 \tag{7.46}$$

$$\leq \frac{1}{\theta^2 m^2} \left\| \frac{1}{m} \boldsymbol{X}_0 \left( \boldsymbol{X}_0^T \boldsymbol{\zeta} \right)^{\circ 3} - \mathbb{E} \left[ \cdot \right] \right\|_2 . \tag{7.47}$$

Under the assumption that

$$m \geq \frac{C}{\left( 1 - \theta \right)^2} \min \left\{ \mu^{-1}, \kappa^2 k^2 \right\} \kappa^2 k^4 \log^3 \left( \kappa k \right), \tag{7.48}$$

applying Lemma 7.2, we have

$$\left\| \frac{1}{m} \boldsymbol{X}_0 \left( \boldsymbol{X}_0^T \boldsymbol{A}^T \boldsymbol{q} \right)^{\circ 3} - \mathbb{E} \left[ \cdot \right] \right\|_2 \leq c\theta \left( 1 - \theta \right) \frac{\left\| \boldsymbol{A}^T \boldsymbol{q} \right\|_4^6}{\kappa^2}. \tag{7.49}$$

with probability larger than $1 - c_2 \exp \left( -k \right) - c_2 k^{-4}$. At last, taking $\varepsilon_B = \theta^2 k^{-4}$, we obtain that

$$\left\| \operatorname{grad} \left[ \psi \right] \left( \boldsymbol{q} \right) - \frac{3 \left( 1 - \theta \right)}{\theta m^2} \operatorname{grad} \left[ \varphi \right] \left( \boldsymbol{q} \right) \right\|_2$$

$$\leq c \frac{1 - \theta}{\theta m^2} \frac{\left\| \boldsymbol{A}^T \boldsymbol{q} \right\|_4^6}{\kappa^2}, \qquad \forall \boldsymbol{q} \in \hat{\mathcal{R}}_{2C_\star} \tag{7.50}$$

with probability larger than $1 - c_2 \exp \left( -k \right) - c_2 k^{-4} - \varepsilon_B - \varepsilon_0$ as desired. $\blacksquare$

## 7.1 Proof of Lemma 7.2

**Lemma 7.2** *Suppose* $\boldsymbol{x}_0 \sim_{\text{i.i.d.}} \operatorname{BG} \left( \theta \right) \in \mathbb{R}^m$. *There exist positive constant* $C$ *such that whenever*

$$m \geq \frac{C}{\left( 1 - \theta \right)^2} \min \left\{ \left( 2C_\star \mu \right)^{-1}, \kappa^2 k^2 \right\} \kappa^2 k^4 \log^3 \left( \kappa k \right) \tag{7.51}$$

*and* $\theta k \geq 1$, *then with probability no smaller than* $1 - c_1 \exp \left( -k \right) - c_2 k^{-4}$,

$$\left\| \frac{1}{m} \boldsymbol{X}_0 \left( \boldsymbol{X}_0^T \boldsymbol{A}^T \boldsymbol{q} \right)^{\circ 3} - \mathbb{E} \left[ \cdot \right] \right\|_2 \leq c\theta \left( 1 - \theta \right) \frac{\left\| \boldsymbol{A}^T \boldsymbol{q} \right\|_4^6}{\kappa^2} \tag{7.52}$$

*holds for all* $\boldsymbol{q} \in \hat{\mathcal{R}}_{2C_\star}$ *with positive constant* $c \leq 1 / \left( 2C_\star \right)$.

**Proof** Let $\bar{\boldsymbol{x}}_i \in \mathbb{R}^{2k-1}$ be generated via

$$\bar{\boldsymbol{x}}_i = \begin{cases} \boldsymbol{x}_i & \left\| \boldsymbol{x}_i \right\|_\infty \leq B \text{ and } \left\| \boldsymbol{x}_i \right\|_0 \leq 4\theta k \log m \\ \boldsymbol{0} & \text{else} \end{cases} \tag{7.53}$$

Let $\bar{\boldsymbol{X}}_0 \in \mathbb{R}^{(2k-1) \times m}$ denote the circulant submatrix generated by $\bar{\boldsymbol{x}}_0$. Then $\bar{\boldsymbol{X}}_0 = \boldsymbol{X}_0$ obtains whenever

1. $\left\| \boldsymbol{x}_0 \right\|_\infty \leq B$, which happens with probability no smaller than $1 - 2\theta m e^{-B^2/2}$ according to Lemma 2.5;

2. $\|\boldsymbol{x}_i\|_0 \le 4\theta k \log m$ holds for any index $i$, applying Lemma 2.4 and Boole's inequality we have

$$\mathbb{E}\left[\mathbf{1}_{\bigcup_i \|\boldsymbol{x}_i\|_0 > 4\theta k \log m}\right]$$

$$\le m\mathbb{P}\left[\|\boldsymbol{x}_i\|_0 > 4\theta k \log m\right] \tag{7.54}$$

$$\le 2m \exp\left(-\tfrac{3}{4}\theta k \log m\right). \tag{7.55}$$

Denote $\boldsymbol{\zeta} = \boldsymbol{A}^T \boldsymbol{q}$ and

$$\boldsymbol{g}_E = \mathbb{E}\left[\frac{1}{m}\boldsymbol{X}_0\left(\boldsymbol{X}_0^T \boldsymbol{A}^T \boldsymbol{q}\right)^{\circ 3}\right], \tag{7.56}$$

$$\bar{\boldsymbol{g}}_E = \mathbb{E}\left[\frac{1}{m}\bar{\boldsymbol{X}}_0\left(\bar{\boldsymbol{X}}_0^T \boldsymbol{A}^T \boldsymbol{q}\right)^{\circ 3}\right], \tag{7.57}$$

then,

$$\mathbb{P}\left[\left\|\frac{1}{m}\boldsymbol{X}_0\left(\boldsymbol{X}_0^T \boldsymbol{\zeta}\right)^{\circ 3} - \boldsymbol{g}_E\right\|_2 \ge c\theta\left(1-\theta\right)\frac{\|\boldsymbol{\zeta}\|_4^6}{\kappa^2}\right]$$

$$\le \mathbb{P}\left[\left\|\frac{1}{m}\bar{\boldsymbol{X}}_0\left(\bar{\boldsymbol{X}}_0^T \boldsymbol{\zeta}\right)^{\circ 3} - \boldsymbol{g}_E\right\|_2 \ge c\theta\left(1-\theta\right)\frac{\|\boldsymbol{\zeta}\|_4^6}{\kappa^2}\right]$$

$$+ 2\theta m e^{-B^2/2} + 2m \exp\left(-\tfrac{3}{4}\theta k \log m\right) \tag{7.58}$$

With triangle inequality, we have

$$\left\|\frac{1}{m}\bar{\boldsymbol{X}}_0\left(\bar{\boldsymbol{X}}_0^T \boldsymbol{\zeta}\right)^{\circ 3} - \boldsymbol{g}_E\right\|_2 \tag{7.59}$$

$$\le \left\|\mathbb{E}\left[\frac{1}{m}\bar{\boldsymbol{X}}_0\left(\bar{\boldsymbol{X}}_0^T \boldsymbol{\zeta}\right)^{\circ 3}\right] - \bar{\boldsymbol{g}}_E\right\|_2 + \|\bar{\boldsymbol{g}}_E - \boldsymbol{g}_E\|_2 .$$

Hence, provided

$$\|\bar{\boldsymbol{g}}_E - \boldsymbol{g}_E\|_2 \le \frac{c}{2}\theta(1-\theta)\frac{\|\boldsymbol{\zeta}\|_4^6}{\kappa^2}, \tag{7.60}$$

we have

$$\mathbb{P}\left[\left\|\frac{1}{m}\bar{\boldsymbol{X}}_0\left(\bar{\boldsymbol{X}}_0^T \boldsymbol{\zeta}\right)^{\circ 3} - \boldsymbol{g}_E\right\|_2 \ge c\theta\left(1-\theta\right)\frac{\|\boldsymbol{\zeta}\|_4^6}{\kappa^2}\right]$$

$$\le \mathbb{P}\left[\left\|\frac{1}{m}\bar{\boldsymbol{X}}_0\left(\bar{\boldsymbol{X}}_0^T \boldsymbol{\zeta}\right)^{\circ 3} - \bar{\boldsymbol{g}}_E\right\|_2 \ge \frac{c}{2}\theta\left(1-\theta\right)\frac{\|\boldsymbol{\zeta}\|_4^6}{\kappa^2}\right]. \tag{7.61}$$

**Truncation Level** Next, we choose a large enough entry-wise truncation level $B$ such that the expectation of the gradient $\mathbb{E}\left[\frac{1}{m}\boldsymbol{X}_0\left(\boldsymbol{X}_0^T \boldsymbol{\zeta}\right)^{\circ 3}\right]$ is close to that of its truncation $\mathbb{E}\left[\frac{1}{m}\bar{\boldsymbol{X}}_0\left(\bar{\boldsymbol{X}}_0^T \boldsymbol{\zeta}\right)^{\circ 3}\right]$. Moreover, we introduce following events notation

$$\mathcal{E}_i \doteq \{\|\boldsymbol{x}_i\|_\infty > B \ \cup \ \|\boldsymbol{x}_i\|_0 > 4\theta k \log m\}, \tag{7.62}$$

then

$$\|\bar{\boldsymbol{g}}_E - \boldsymbol{g}_E\|_2$$

$$= \left\|\mathbb{E}\left[\frac{1}{m}\sum_i \boldsymbol{x}_i \langle \boldsymbol{x}_i, \boldsymbol{\zeta}\rangle^3 \cdot \mathbf{1}_{\mathcal{E}_i}\right]\right\|_2 \tag{7.63}$$

$$\leq \frac{1}{m} \sum_i \left\| \mathbb{E}\left[\boldsymbol{x}_i \left\langle \boldsymbol{x}_i, \boldsymbol{\zeta} \right\rangle^3 \cdot \mathbf{1}_{\mathcal{E}_i} \right] \right\|_2 \tag{7.64}$$

$$\leq \frac{1}{m} \sum_i \left( \mathbb{E}\left[ \left\| \boldsymbol{x}_i \left(\boldsymbol{x}_i^T \boldsymbol{\zeta}\right)^{\circ 3} \right\|_2^2 \right] \cdot \mathbb{E}\left[\mathbf{1}_{\mathcal{E}_i}\right] \right)^{1/2} \tag{7.65}$$

$$\leq \left( \mathbb{E}\left[ \left\| \boldsymbol{x}_i \right\|_2^8 \right] \right)^{1/2} \times$$
$$\sqrt{\mathbb{E}\left[\mathbf{1}_{\|\boldsymbol{x}_i\|_\infty > B}\right] + \mathbb{E}\left[\mathbf{1}_{\|\boldsymbol{x}_i\|_0 > 4\theta k \log m}\right]} \tag{7.66}$$

$$\leq 50k^2 \sqrt{4\theta k e^{-B^2/2} + \exp\left(-\tfrac{3}{4}\theta k \log m\right)} \tag{7.67}$$

By setting

$$B \geq C' \log^{1/2}\left( \frac{\kappa^4 k^8}{\theta\left(1-\theta\right)^2} \right), \tag{7.68}$$

we have

$$\theta k e^{-B^2/2} \leq \frac{1}{2}\left(\frac{c}{100}\right)^2 \theta^2\left(1-\theta\right)^2 \frac{\|\boldsymbol{\zeta}\|_4^{12}}{\kappa^4 k^4}. \tag{7.69}$$

In addition, whenever

$$\theta k \geq \frac{4}{3\log m} \log\left( \frac{400^2 \kappa^4 k^4}{c^2 \theta^2\left(1-\theta\right)^2 \|\boldsymbol{\zeta}\|_4^{12}} \right), \tag{7.70}$$

we have

$$\exp\left(-\tfrac{3}{4}\theta k \log m\right) \leq \frac{1}{2}\left(\frac{c}{100}\right)^2 \theta^2\left(1-\theta\right)^2 \frac{\|\boldsymbol{\zeta}\|_4^{12}}{\kappa^4 k^4}. \tag{7.71}$$

Therefore,

$$\sqrt{4\theta k e^{-B^2/2} + \exp\left(-\tfrac{3}{4}\theta k \log m\right)} \leq \frac{c}{2}\theta\left(1-\theta\right) \frac{\|\boldsymbol{\zeta}\|_4^6}{50\kappa^2 k^2}. \tag{7.72}$$

In addition,

$$\left( \mathbb{E}\left[ \left\| \boldsymbol{x}_i \right\|_2^8 \right] \right)^{1/2} \leq \left(7!! \cdot 2^4 k^4\right)^{1/2} < 50k^2. \tag{7.73}$$

Plugging in Eq [(7.73)](#) and [(7.72)](#) back to [(7.67)](#), we obtain that

$$\|\bar{\boldsymbol{g}}_E - \boldsymbol{g}_E\|_2 \leq \frac{c}{2}\theta\left(1-\theta\right) \frac{\left\|\boldsymbol{A}^T\boldsymbol{q}\right\|_4^6}{\kappa^2}, \tag{7.74}$$

and hence

$$\mathbb{P}\left[ \left\| \frac{1}{m}\bar{\boldsymbol{X}}_0 \left(\bar{\boldsymbol{X}}_0^T\boldsymbol{\zeta}\right)^{\circ 3} - \boldsymbol{g}_E \right\|_2 \geq c\theta\left(1-\theta\right) \frac{\|\boldsymbol{\zeta}\|_4^6}{\kappa^2} \right]$$
$$\leq \mathbb{P}\left[ \left\| \frac{1}{m}\bar{\boldsymbol{X}}_0 \left(\bar{\boldsymbol{X}}_0^T\boldsymbol{\zeta}\right)^{\circ 3} - \bar{\boldsymbol{g}}_E \right\|_2 \geq \frac{c}{2}\theta\left(1-\theta\right) \frac{\|\boldsymbol{\zeta}\|_4^6}{\kappa^2} \right]. \tag{7.75}$$

**Independent Submatrices.** To deal with the complicated dependence within the random circulant matrix $\boldsymbol{X}_0$, we break $\boldsymbol{X}_0$ into submatrices $\boldsymbol{X}_1, \ldots, \boldsymbol{X}_{2k-1}$, each of which is (marginally) distributed as a $(2k-1) \times \frac{m}{2k-1}$ i.i.d. $\mathrm{BG}(\theta)$ random matrix. Indeed, there exists a permutation $\boldsymbol{\Pi}$ such that

$$\boldsymbol{X}_0\boldsymbol{\Pi} = \left[\boldsymbol{X}_1, \boldsymbol{X}_2, \cdots, \boldsymbol{X}_{2k-1}\right], \tag{7.76}$$

with

$$\boldsymbol{X}_i = \left[\boldsymbol{x}_i, \boldsymbol{x}_{i+(2k-1)}, \cdots, \boldsymbol{x}_{i+(m-2k-1)}\right]. \tag{7.77}$$

We apply similar matrix breaking approach for the truncated matrix $\bar{X}$. The summands within each term $\bar{X}_i \left( \bar{X}_i^T \zeta \right)^{\circ 3}$ are mutually independent and hence is amenable to classical concentration results.

$$\frac{1}{m} \bar{X}_0 \left( \bar{X}_0^T \zeta \right)^{\circ 3} = \frac{1}{m} \sum_{l=1}^{m} \langle \bar{x}_l, \zeta \rangle^3 \bar{x}_l \tag{7.78}$$

$$= \sum_{i=1}^{2k-1} \frac{1}{m} \left( \sum_{j=0}^{\frac{m}{2k-1}-1} \langle \bar{x}_{i+(2k-1)j}, \zeta \rangle^3 \bar{x}_{i+(2k-1)j} \right) \tag{7.79}$$

$$= \sum_{i=1}^{2k-1} \frac{1}{m} \bar{X}_i \left( \bar{X}_i^T \zeta \right)^{\circ 3}. \tag{7.80}$$

We conservatively bound the quantity of interest, $\frac{1}{m} \bar{X}_0 \left( \bar{X}_0^T \zeta \right)^{\circ 3}$, by ensuring that for each $k$, $\bar{X}_k \left( \bar{X}_k^T \zeta \right)^{\circ 3}$ be close to its expectation.

$$\mathbb{P} \left[ \left\| \frac{1}{m} \bar{X}_0 \left( \bar{X}_0^T \zeta \right)^{\circ 3} - \bar{g}_E \right\|_2 \geq \frac{c}{2} \theta \left( 1 - \theta \right) \frac{\|\zeta\|_4^6}{\kappa^2} \right]$$

$$\leq \sum_{i=1}^{2k-1} \mathbb{P} \left[ \left\| \frac{1}{m} \bar{X}_i \left( \bar{X}_i^T \zeta \right)^{\circ 3} - \frac{\bar{g}_E}{2k-1} \right\|_2 \geq \frac{c}{2} \frac{\theta \left( 1 - \theta \right) \|\zeta\|_4^6}{\kappa^2 \left( 2k - 1 \right)} \right]$$

$$= \sum_{i=1}^{2k-1} \mathbb{P} \left[ \left\| \frac{1}{m} \bar{X}_i \left( \bar{X}_i^T \zeta \right)^{\circ 3} - \bar{g}_E \right\|_2 \geq \frac{c}{2} \frac{\theta \left( 1 - \theta \right) \|\zeta\|_4^6}{\kappa^2 \left( 2k - 1 \right)} \right]$$

Applying Bernstein inequality for matrix variables as in Lemma 9.7, with $d_1 = 2k - 1$, $d_2 = 1$, we can obtain that for independent random vectors $v_1, \ldots, v_n$ with

$$\sigma^2 = \sum_{i=1}^{n} \mathbb{E}[\|v_i\|_2^2] \tag{7.81}$$

and ensuring that

$$\|v_i\|_2 \leq R \qquad a.s. \tag{7.82}$$

we obtain that

$$\mathbb{P} \left[ \left\| \sum_i v_i - \mathbb{E}[\cdot] \right\| > t \right] \leq 4k \exp \left( \frac{-t^2/2}{\sigma^2 + 2Rt/3} \right) \tag{7.83}$$

Here, we have used that

$$\left\| \sum_{i=1}^{n} \mathbb{E}[v_i v_i^*] \right\| \leq \operatorname{tr} \sum_{i=1}^{n} \mathbb{E}[v_i v_i^*] \tag{7.84}$$

$$= \sum_{i=1}^{n} \mathbb{E} \left[ \|v_i\|_2^2 \right]. \tag{7.85}$$

and

$$w_i = \bar{x}_i \langle \bar{x}_i, \zeta \rangle^3. \tag{7.86}$$

Notice that

$$\|w_i\|_2 \leq \|\bar{x}_i\|_2^4 \tag{7.87}$$

$$\leq \left( 4B^2 \theta k \log m \right)^2 \tag{7.88}$$

$$= 16B^4\theta^2 k^2 \log m. \tag{7.89}$$

Let us further note that

$$\sum_{\substack{j_1,\\ j_2\neq j_3\neq j_4}} \mathbb{E}\left[\bar{\boldsymbol{x}}_i(j_1)^2 \bar{\boldsymbol{x}}_i(j_2)^2 \boldsymbol{\zeta}_{j_2}^2 \bar{\boldsymbol{x}}_i(j_3)^2 \boldsymbol{\zeta}_{j_3}^2 \bar{\boldsymbol{x}}_i(j_4)^2 \boldsymbol{\zeta}_{j_4}^2\right]$$

$$= 3 \sum_{j_1\neq j_2\neq j_3} \mathbb{E}\left[\bar{\boldsymbol{x}}_i(j_1)^4 \boldsymbol{\zeta}_{j_1}^2 \bar{\boldsymbol{x}}_i(j_2)^2 \boldsymbol{\zeta}_{j_2}^2 \bar{\boldsymbol{x}}_i(j_3)^2 \boldsymbol{\zeta}_{j_3}^2\right]$$

$$+ \sum_{j_1=1}^{2k-1} \mathbb{E}\left[\bar{\boldsymbol{x}}_i(j_1)^2\right] \times$$

$$\sum_{j_1\neq j_2\neq j_3\neq j_4} \mathbb{E}\left[\bar{\boldsymbol{x}}_i(j_2)^2 \boldsymbol{\zeta}_{j_2}^2 \bar{\boldsymbol{x}}_i(j_3)^2 \boldsymbol{\zeta}_{j_3}^2 \bar{\boldsymbol{x}}_i(j_4)^2 \boldsymbol{\zeta}_{j_4}^2\right] \tag{7.90}$$

$$\leq 2\theta k \times \theta^3 \|\boldsymbol{\zeta}\|_2^6 + 3 \times 3\theta^3 \|\boldsymbol{\zeta}\|_2^6 \tag{7.91}$$

In similar vein, we can obtain that

$$\sum_{j_1,j_2\neq j_3} \mathbb{E}\left[\bar{\boldsymbol{x}}_i(j_1)^2 \bar{\boldsymbol{x}}_i(j_2)^2 \boldsymbol{\zeta}_{j_2}^2 \bar{\boldsymbol{x}}_i(j_3)^4 \boldsymbol{\zeta}_{j_3}^4\right]$$

$$= \sum_{j_1} \mathbb{E}\left[\bar{\boldsymbol{x}}_i(j_1)^2\right] \sum_{j_2\neq j_3\neq j_1} \mathbb{E}\left[\bar{\boldsymbol{x}}_i(j_2)^2 \boldsymbol{\zeta}_{j_2}^2 \bar{\boldsymbol{x}}_i(j_3)^4 \boldsymbol{\zeta}_{j_3}^4\right]$$

$$+ \sum_{j_1\neq j_2} \mathbb{E}\left[\bar{\boldsymbol{x}}_i(j_1)^4 \boldsymbol{\zeta}_{j_1}^2 \bar{\boldsymbol{x}}_i(j_2)^4 \boldsymbol{\zeta}_{j_2}^4\right]$$

$$+ \sum_{j_1\neq j_2} \mathbb{E}\left[\bar{\boldsymbol{x}}_i(j_1)^2 \boldsymbol{\zeta}_{j_1}^2 \bar{\boldsymbol{x}}_i(j_2)^6 \boldsymbol{\zeta}_{j_2}^4\right] \tag{7.92}$$

$$\leq 2\theta k \times 3\theta^2 \|\boldsymbol{\zeta}\|_2^2 \|\boldsymbol{\zeta}\|_4^4 + (9+15)\,\theta^2 \|\boldsymbol{\zeta}\|_2^2 \|\boldsymbol{\zeta}\|_4^4 \tag{7.93}$$

and

$$\sum_{j_1,j_2} \mathbb{E}\left[\bar{\boldsymbol{x}}_i(j_1)^2 \bar{\boldsymbol{x}}_i(j_2)^6 \boldsymbol{\zeta}_{j_2}^6\right]$$

$$= \sum_{j_1} \mathbb{E}\left[\bar{\boldsymbol{x}}_i(j_1)^2\right] \sum_{j_2\neq j_1} \mathbb{E}\left[\bar{\boldsymbol{x}}_i(j_2)^6 \boldsymbol{\zeta}_{j_2}^6\right]$$

$$+ \sum_{j_1} \mathbb{E}\left[\bar{\boldsymbol{x}}_i(j_1)^8 \boldsymbol{\zeta}_{j_1}^6\right] \tag{7.94}$$

$$\leq 2\theta k \times 15\theta \|\boldsymbol{\zeta}\|_6^6 + 105\theta \|\boldsymbol{\zeta}\|_6^6 \tag{7.95}$$

Now we calculate

$$\mathbb{E}\left[\|\boldsymbol{w}_i\|_2^2\right] = \mathbb{E}\left[\|\bar{\boldsymbol{x}}_i\|_2^2 \langle \bar{\boldsymbol{x}}_i, \boldsymbol{\zeta}\rangle^6\right] \tag{7.96}$$

$$= \mathbb{E}\left[\sum_{j_1,\ldots,j_7} \bar{\boldsymbol{x}}_i(j_1)^2 \prod_{\ell=2}^{7} \bar{\boldsymbol{x}}_i(j_\ell)\boldsymbol{\zeta}_{j_\ell}\right] \tag{7.97}$$

$$= 15 \sum_{\substack{j_1,\\ j_2\neq j_3\neq j_4}} \mathbb{E}\left[\bar{\boldsymbol{x}}_i(j_1)^2 \bar{\boldsymbol{x}}_i(j_2)^2 \boldsymbol{\zeta}_{j_2}^2 \bar{\boldsymbol{x}}_i(j_3)^2 \boldsymbol{\zeta}_{j_3}^2 \bar{\boldsymbol{x}}_i(j_4)^2 \boldsymbol{\zeta}_{j_4}^2\right]$$

$$+ 15 \sum_{j_1,j_2\neq j_3} \mathbb{E}\left[\bar{\boldsymbol{x}}_i(j_1)^2 \bar{\boldsymbol{x}}_i(j_2)^2 \boldsymbol{\zeta}_{j_2}^2 \bar{\boldsymbol{x}}_i(j_3)^4 \boldsymbol{\zeta}_{j_3}^4\right]$$

$$+ \sum_{j_1, j_2} \mathbb{E}\left[\bar{\boldsymbol{x}}_i(j_1)^2 \bar{\boldsymbol{x}}_i(j_2)^6 \boldsymbol{\zeta}_{j_2}^6\right] \tag{7.98}$$

$$\leq 15\theta^3 \|\boldsymbol{\zeta}\|_2^6 (2\theta k + 9)$$
$$+ 15\theta^2 \|\boldsymbol{\zeta}\|_4^4 (6 + 24)$$
$$+ \theta \|\boldsymbol{\zeta}\|_6^6 (30\theta k + 105) \tag{7.99}$$
$$\leq 150\theta^2 k + 600\theta \tag{7.100}$$

whence for $\theta > 1/k$,

$$\mathbb{E}\left[\|\boldsymbol{w}_i\|_2^2\right] \leq C\theta^2 k, \tag{7.101}$$

and hence

$$\sigma^2 \leq C'\theta^2 m. \tag{7.102}$$

Matrix Bernstein gives that

$$\mathbb{P}\left[\left\|\bar{\boldsymbol{X}}_i(\bar{\boldsymbol{X}}_i^T \boldsymbol{\zeta})^{\circ 3} - \mathbb{E}[\cdot]\right\|_2 \geq t\right]$$
$$\leq 4k \exp\left(\frac{-t^2/2}{C\theta^2 m + C' B^4 \theta^2 k^2 \log^2 kt}\right). \tag{7.103}$$

Setting $t = \frac{c}{4} \frac{m\theta(1-\theta)\|\boldsymbol{\zeta}\|_4^6}{\kappa^2 (2k-1)}$, we obtain that

$$\mathbb{P}\left[\left\|\frac{1}{m}\bar{\boldsymbol{X}}_i(\bar{\boldsymbol{X}}_i^T \boldsymbol{\zeta})^{\circ 3} - \mathbb{E}[\cdot]\right\|_2 \geq \frac{c}{4} \frac{\theta(1-\theta)\|\boldsymbol{\zeta}\|_4^6}{\kappa^2 (2k-1)}\right]$$
$$\leq 4k \exp\left(-\frac{c''m (1-\theta)^2 \|\boldsymbol{\zeta}\|_4^{12}}{\kappa^4 k^2 + \theta (1-\theta) B^4 \kappa^2 k^3 \|\boldsymbol{\zeta}\|_4^6}\right) \tag{7.104}$$

$\varepsilon$**-Net Covering** To obtain a probability bound for all $\boldsymbol{q} \in \mathbb{S}^{k-1}$, we choose a set of $\boldsymbol{\zeta}_n = \boldsymbol{A}^T \boldsymbol{q}_n$ with $n = 1, \cdots, N$. Suppose for any $\boldsymbol{q} \in \mathbb{S}^{k-1}$, there exists $\boldsymbol{q}_n$ such that $\|\boldsymbol{q} - \boldsymbol{q}_n\|_2 \leq \varepsilon$, then

$$\left\|\frac{1}{m}\bar{\boldsymbol{X}}_i\left(\bar{\boldsymbol{X}}_i^T \boldsymbol{\zeta}\right)^{\circ 3} - \frac{1}{m}\bar{\boldsymbol{X}}_i\left(\bar{\boldsymbol{X}}_i^T \boldsymbol{\zeta}_n\right)^{\circ 3}\right\|_2 \leq L \|\boldsymbol{q} - \boldsymbol{q}_n\|_2. \tag{7.105}$$

For entry wise bounded $\bar{\boldsymbol{X}}_i \in \mathbb{R}^{(2k-1) \times \frac{m}{2k-1}}$, we have

$$\left\|\bar{\boldsymbol{X}}_i\right\|_2 \leq \sqrt{2\theta m}B, \quad \left\|\bar{\boldsymbol{X}}_i \boldsymbol{e}_j\right\|_2 \leq \sqrt{4\theta k}B, \tag{7.106}$$

then the Lipschitz constant $L$ can be bounded as

$$L \leq \frac{1}{m}\left\|\bar{\boldsymbol{X}}_i\right\|_2 \left\|\mathrm{diag}\left(\bar{\boldsymbol{X}}_i^T \boldsymbol{\zeta}\right)^{\circ 2}\right\|_2 \left\|\bar{\boldsymbol{X}}_i^T \boldsymbol{A}^T\right\|_2 \tag{7.107}$$
$$\leq 8\theta^2 k B^4. \tag{7.108}$$

With triangle inequality, we have

$$\left\|\frac{1}{m}\bar{\boldsymbol{X}}_i\left(\bar{\boldsymbol{X}}_i^T \boldsymbol{\zeta}\right)^{\circ 3} - \mathbb{E}\left[\frac{1}{m}\bar{\boldsymbol{X}}_i\left(\bar{\boldsymbol{X}}_i^T \boldsymbol{\zeta}\right)^{\circ 3}\right]\right\|_2$$
$$\leq \left\|\mathbb{E}\left[\frac{1}{m}\bar{\boldsymbol{X}}_i\left(\bar{\boldsymbol{X}}_i^T \boldsymbol{\zeta}\right)^{\circ 3}\right] - \mathbb{E}\left[\frac{1}{m}\bar{\boldsymbol{X}}_i\left(\bar{\boldsymbol{X}}_i^T \boldsymbol{\zeta}_n\right)^{\circ 3}\right]\right\|_2$$
$$+ \left\|\frac{1}{m}\bar{\boldsymbol{X}}_i\left(\bar{\boldsymbol{X}}_i^T \boldsymbol{\zeta}_n\right)^{\circ 3} - \mathbb{E}\left[\frac{1}{m}\bar{\boldsymbol{X}}_i\left(\bar{\boldsymbol{X}}_i^T \boldsymbol{\zeta}_n\right)^{\circ 3}\right]\right\|_2$$

$$+ \left\| \frac{1}{m} \bar{\boldsymbol{X}}_i \left( \bar{\boldsymbol{X}}_i^T \boldsymbol{\zeta} \right)^{\circ 3} - \frac{1}{m} \bar{\boldsymbol{X}}_i \left( \bar{\boldsymbol{X}}_i^T \boldsymbol{\zeta}_n \right)^{\circ 3} \right\|_2 \tag{7.109}$$

$$\leq \left\| \frac{1}{m} \bar{\boldsymbol{X}}_i \left( \bar{\boldsymbol{X}}_i^T \boldsymbol{\zeta}_n \right)^{\circ 3} - \mathbb{E} \left[ \frac{1}{m} \bar{\boldsymbol{X}}_i \left( \bar{\boldsymbol{X}}_i^T \boldsymbol{\zeta}_n \right)^{\circ 3} \right] \right\|_2$$

$$+ 2L\varepsilon. \tag{7.110}$$

Hence we need to choose the $\varepsilon$-net to cover the sphere of $\boldsymbol{q}$ with

$$\varepsilon = \frac{c}{4} \frac{\theta \left( 1 - \theta \right)}{\kappa^2 \left( 2k - 1 \right) L} \min_{\boldsymbol{q} \in \mathbb{S}^{k-1}} \|\boldsymbol{\zeta}\|_4^6, \tag{7.111}$$

plug in $L \leq 4\theta^2 k B^4$ and number of sample $N$ suffice

$$N \leq \left( \frac{3}{\varepsilon} \right)^k \tag{7.112}$$

$$\leq \exp \left( k \ln \left( \frac{3}{\varepsilon} \right) \right) \tag{7.113}$$

$$\leq \exp \left[ k \ln \left( C \frac{\theta^2 \kappa^2 k^4 B^4}{\theta \left( 1 - \theta \right)} \right) \right] \tag{7.114}$$

For $n = 1, \cdots, N$, denote

$$P_i \left( \boldsymbol{q}_n \right) = \mathbb{P} \left[ \left\| \frac{\bar{\boldsymbol{X}}_i \left( \bar{\boldsymbol{X}}_i^T \boldsymbol{\zeta}_n \right)^{\circ 3}}{m} - \mathbb{E}[\cdot] \right\|_2 \geq \frac{c\theta(1-\theta)\|\boldsymbol{\zeta}_n\|_4^6}{4\kappa^2 \left( 2k - 1 \right)} \right], \tag{7.115}$$

then together with union bound over all $\boldsymbol{q}_n$, we obtain that,

$$\mathbb{P} \left[ \sup_{\boldsymbol{q} \in \hat{\mathcal{R}}_{2C_\star}} \frac{\left\| \frac{1}{m} \bar{\boldsymbol{X}}_i \left( \bar{\boldsymbol{X}}_i^T \boldsymbol{\zeta} \right)^{\circ 3} - \mathbb{E}[\cdot] \right\|_2}{\|\boldsymbol{\zeta}\|_4^6} \geq \frac{c}{2} \frac{\theta \left( 1 - \theta \right)}{\kappa^2 \left( 2k - 1 \right)} \right]$$

$$\leq \sum_{\boldsymbol{q}_n \in \hat{\mathcal{R}}_{2C_\star}} P_i \left( \boldsymbol{q}_n \right) \tag{7.116}$$

$$\leq N \max_{\boldsymbol{q}_n \in \hat{\mathcal{R}}_{2C_\star}} P_i \left( \boldsymbol{q}_n \right) \tag{7.117}$$

$$\leq 4k \sup_{\boldsymbol{q} \in \hat{\mathcal{R}}_{2C_\star}} \exp \left( - \frac{cm \left( 1 - \theta \right)^2 \|\boldsymbol{\zeta}\|_4^{12}}{\kappa^4 k^2 + \theta \left( 1 - \theta \right) B^4 \kappa^2 k^3 \|\boldsymbol{\zeta}\|_4^6} \right) \times$$

$$\exp \left( k \ln \left( \frac{3}{\varepsilon} \right) \right). \tag{7.118}$$

Hence

$$\mathbb{P} \left[ \sup_{\boldsymbol{q} \in \hat{\mathcal{R}}_{2C_\star}} \frac{\left\| \frac{1}{m} \bar{\boldsymbol{X}}_0 \left( \bar{\boldsymbol{X}}_0^T \boldsymbol{\zeta} \right)^{\circ 3} - \mathbb{E}[\cdot] \right\|_2}{\|\boldsymbol{\zeta}\|_4^6} \geq \frac{c}{2} \frac{\theta \left( 1 - \theta \right)}{\kappa^2} \right]$$

$$\leq \sum_i \mathbb{P} \left[ \sup_{\boldsymbol{q} \in \hat{\mathcal{R}}_{2C_\star}} \frac{\left\| \frac{1}{m} \bar{\boldsymbol{X}}_i \left( \bar{\boldsymbol{X}}_i^T \boldsymbol{\zeta} \right)^{\circ 3} - \mathbb{E}[\cdot] \right\|_2}{\|\boldsymbol{\zeta}\|_4^6} \geq \frac{c\theta \left( 1 - \theta \right)}{2\kappa^2 \left( 2k - 1 \right)} \right] \tag{7.119}$$

$$\leq \left( 2k - 1 \right) \max_i$$

$$\mathbb{P}\left[\sup_{\boldsymbol{q}\in\hat{\mathcal{R}}_{2C_\star}}\frac{\left\|\frac{1}{m}\bar{\boldsymbol{X}}_i\left(\bar{\boldsymbol{X}}_i^T\boldsymbol{\zeta}\right)^{\circ 3}-\mathbb{E}\left[\cdot\right]\right\|_2}{\|\boldsymbol{\zeta}\|_4^6}\geq\frac{c\theta\left(1-\theta\right)}{2\kappa^2\left(2k-1\right)}\right] \tag{7.120}$$

$$\leq 8k^2\sup_{\boldsymbol{q}\in\hat{\mathcal{R}}_{2C_\star}}\exp\left(-\frac{cm\left(1-\theta\right)^2\|\boldsymbol{\zeta}\|_4^{12}}{\kappa^4 k^2+\theta\left(1-\theta\right)B^4\kappa^2 k^3\|\boldsymbol{\zeta}\|_4^6}\right)\times$$
$$\exp\left(k\ln\left(\frac{3}{\varepsilon}\right)\right), \tag{7.121}$$

which is bounded by $\exp\left(-k\right)$ as long as

$$m\geq C\frac{\min\left\{\left(2C_\star\mu\right)^{-2},\kappa^2 k^2\right\}}{\left(1-\theta\right)^2}\kappa^2 k^4\log^3\left(\kappa k\right) \tag{7.122}$$

$$\geq C'k\log\left(\frac{\theta\kappa^2 k^2 B^4}{\left(1-\theta\right)\|\boldsymbol{\zeta}\|_4^6}\right)\times$$
$$\max\left\{\frac{\kappa^4 k^2}{\left(1-\theta\right)^2\|\boldsymbol{\zeta}\|_4^{12}},\frac{\theta B^4\kappa^2 k^3}{\left(1-\theta\right)\|\boldsymbol{\zeta}\|_4^6}\right\}. \tag{7.123}$$

To sum up, we obtain that for all $\boldsymbol{q}\in\hat{\mathcal{R}}_{2C_\star}$, inequality

$$\left\|\frac{1}{m}\boldsymbol{X}_0\left(\boldsymbol{X}_0^T\boldsymbol{A}^T\boldsymbol{q}\right)^{\circ 3}-\mathbb{E}\left[\cdot\right]\right\|_2\leq c\theta\left(1-\theta\right)\frac{\left\|\boldsymbol{A}^T\boldsymbol{q}\right\|_4^6}{\kappa^2} \tag{7.124}$$

holds with probability no smaller than $1-c_1\exp\left(-k\right)-c_2 k^{-4}-c_3\exp\left(-\theta k\right)$. ∎

# 8  Concentration for Hessian (Lemma 8.1)

**Lemma 8.1** *Suppose $\boldsymbol{x}_0\sim_{\text{i.i.d.}}\text{BG}\left(\theta\right)$. There exists positive constant $C$ that whenever*

$$m\geq C\theta\frac{\min\left\{\left(2C_\star\mu\kappa^2\right)^{-4/3},k^2\right\}}{\left(1-\theta\right)^2\sigma_{\min}^2}\kappa^6 k^4\log^3\left(\frac{\kappa k}{\left(1-\theta\right)\sigma_{\min}}\right) \tag{8.1}$$

*and $\theta\geq\log k/k$, then with probability no smaller than $1-c_1\exp\left(-k\right)-c_2 k^{-4}-48k^{-7}-48m^{-5}-24k\exp\left(-\frac{1}{144}\min\left\{k,3\sqrt{\theta m}\right\}\right)$,*

$$\left\|\text{Hess}[\psi]\left(\boldsymbol{q}\right)-\frac{3\left(1-\theta\right)}{\theta m^2}\text{Hess}[\varphi]\left(\boldsymbol{q}\right)\right\|_2\leq c\frac{1-\theta}{\theta m^2}\left\|\boldsymbol{A}^T\boldsymbol{q}\right\|_4^4, \tag{8.2}$$

*holds for all $\boldsymbol{q}\in\hat{\mathcal{R}}_{2C_\star}$ with positive constant $c\leq 0.048\leq 3\left(1-6c_\star-36c_\star^2-24c_\star^3\right)$.*

**Proof** Denote $\boldsymbol{\eta}=\boldsymbol{Y}^T\left(\boldsymbol{Y}\boldsymbol{Y}^T\right)^{-1/2}\boldsymbol{q}$ and $\bar{\boldsymbol{\eta}}=\boldsymbol{Y}^T\left(\theta m\boldsymbol{A}_0\boldsymbol{A}_0^T\right)^{-1/2}\boldsymbol{q}=\left(\theta m\right)^{-1/2}\boldsymbol{X}_0^T\boldsymbol{\zeta}$, and

$$\boldsymbol{W}=\left(\frac{1}{\theta m}\boldsymbol{Y}\boldsymbol{Y}^T\right)^{-1/2}-\left(\boldsymbol{A}_0\boldsymbol{A}_0^T\right)^{-1/2}, \tag{8.3}$$

$$\widehat{\boldsymbol{Y}}=\left(\boldsymbol{Y}\boldsymbol{Y}^T\right)^{-1/2}\boldsymbol{Y}. \tag{8.4}$$

Then we have

$$\left\|\text{Hess}\left[\psi\right]\left(\boldsymbol{q}\right)-\frac{3\left(1-\theta\right)}{\theta m^2}\text{Hess}\left[\varphi\right]\left(\boldsymbol{q}\right)\right\|_2$$

$$= \left\| P_{q^\perp} \left[ \frac{3}{m} \widehat{Y} \operatorname{diag}\left(\eta^{\circ 2}\right) \widehat{Y}^T - \langle q, \nabla\psi\left(q\right)\rangle I \right] P_{q^\perp} \right.$$

$$\left. - \frac{3\left(1-\theta\right)}{\theta m^2} P_{q^\perp} \left[ 3A \operatorname{diag}\left(\zeta^{\circ 2}\right) A^T - \|\zeta\|_4^4 I \right] P_{q^\perp} \right\|_2 \tag{8.5}$$

$$\leq \left\| P_{q^\perp} \left[ \frac{3}{m} \widehat{Y} \operatorname{diag}\left(\eta^{\circ 2}\right) \widehat{Y}^T \right] P_{q^\perp} \right.$$

$$\left. - P_{q^\perp} \left[ \frac{9\left(1-\theta\right)}{\theta m^2} A \operatorname{diag}\left(\zeta^{\circ 2}\right) A^T - \frac{3}{m^2} I \right] P_{q^\perp} \right\|_2$$

$$+ \left\| \left[ \langle q, \nabla\psi\left(q\right)\rangle - \frac{3\left(1-\theta\right)}{\theta m^2} \|\zeta\|_4^4 - \frac{3}{m^2} \right] P_{q^\perp} \right\|_2 \tag{8.6}$$

$$\leq \frac{3}{\theta m^2} \underbrace{\left\| WY \operatorname{diag}\left(\eta^{\circ 2}\right) Y^T \left( \frac{1}{\theta m} YY^T \right)^{-1/2} \right\|_2}_{\Delta_1^H}$$

$$+ \frac{3}{\theta m^2} \underbrace{\left\| AX_0 \operatorname{diag}\left(\eta^{\circ 2}\right) Y^T W \right\|_2}_{\Delta_2^H}$$

$$+ \frac{3}{\theta m^2} \underbrace{\left\| AX_0 \operatorname{diag}\left(\eta^{\circ 2} - \bar{\eta}^{\circ 2}\right) X_0^T A^T \right\|_2}_{\Delta_3^H}$$

$$+ \frac{3}{\theta m^2} \underbrace{\left\| P_{q^\perp} \left[ AX_0 \operatorname{diag}\left(\bar{\eta}^{\circ 2}\right) X_0^T A^T \right] P_{q^\perp} \right.}_{}$$

$$\underbrace{\left. - P_{q^\perp} \left[ 3\left(1-\theta\right) A \operatorname{diag}\left(\zeta^{\circ 2}\right) A^T + \theta I \right] P_{q^\perp} \right\|_2}_{\Delta_4^H}$$

$$+ \underbrace{\left\| \left[ \langle q, \nabla\psi\left(q\right)\rangle - \frac{3\left(1-\theta\right)}{\theta m^2} \|\zeta\|_4^4 - \frac{3}{m^2} \right] P_{q^\perp} \right\|_2}_{\Delta_5^H} \tag{8.7}$$

In the rest of the proof, we prove that

$$\Delta_i^H \leq \frac{c}{9} \frac{1-\theta}{\theta m^2} \|\zeta\|_4^4, \quad i = 1, 2, 3. \tag{8.8}$$

and

$$\Delta_i^H \leq \frac{c}{3} \frac{1-\theta}{\theta m^2} \|\zeta\|_4^4, \quad i = 4, 5. \tag{8.9}$$

First, let us note that

$$C\left(1-\theta\right)^{-2} \sigma_{\min}^{-2} \kappa^6 k^5 \log^3\left( \frac{\kappa k}{\left(1-\theta\right)\sigma_{\min}} \right) \tag{8.10}$$

$$\leq C\left( \frac{\kappa k}{\left(1-\theta\right)\sigma_{\min}} \right)^6 \log^3\left( \frac{\kappa k}{\left(1-\theta\right)\sigma_{\min}} \right) \tag{8.11}$$

$$\leq C\left( \frac{\kappa k}{\left(1-\theta\right)\sigma_{\min}} \right)^9 \tag{8.12}$$

or

$$\frac{\log^3 \left( C \left( 1 - \theta \right)^{-2} \sigma_{\min}^{-2} \kappa^6 k^5 \log^3 \left( \frac{\kappa k}{(1-\theta)\sigma_{\min}} \right) \right)}{C \log^3 \left( \frac{\kappa k}{(1-\theta)\sigma_{\min}} \right)}$$

$$\leq \left( \frac{\log C + 9 \log \left( \frac{\kappa k}{(1-\theta)\sigma_{\min}} \right)}{C^{1/3} \log \left( \frac{\kappa k}{(1-\theta)\sigma_{\min}} \right)} \right)^3 \tag{8.13}$$

$$\leq \left( \frac{\log C}{C^{1/3} \log \left( \frac{\kappa k}{(1-\theta)\sigma_{\min}} \right)} + \frac{9}{C^{1/3}} \right)^3 \tag{8.14}$$

$$\leq \left( \frac{1}{C^{1/6}} + \frac{1}{2} \frac{1}{C^{1/6}} \right)^3 \qquad (C \geq 10^8) \tag{8.15}$$

$$\leq \frac{4}{C^{1/2}}. \tag{8.16}$$

Since

$$m \geq C \frac{\min \left\{ \left( 2C_\star \mu \kappa^2 \right)^{-4/3}, k^2 \right\}}{\left( 1 - \theta \right)^2 \sigma_{\min}^2} \kappa^6 k^4 \log^3 \left( \frac{\kappa k}{(1 - \theta) \sigma_{\min}} \right), \tag{8.17}$$

as the ratio $\log^3 m/m$ decreases with increasing $m$, then

$$\frac{\log^3 m}{m}$$

$$\leq \frac{\log^3 \left( C \frac{\kappa^6 k^5}{(1-\theta)^2 \sigma_{\min}^2} \log^3 \left( \frac{\kappa k}{\sigma_{\min}(1-\theta)} \right) \right)}{C \log^3 \left( \frac{\kappa k}{\sigma_{\min}(1-\theta)} \right)}$$

$$\times \frac{\left( 1 - \theta \right)^2 \sigma_{\min}^2}{\min \left\{ \left( 2C_\star \mu \kappa^2 \right)^{-2/3}, k \right\} \kappa^6 k^4} \tag{8.18}$$

$$\leq \frac{4}{C^{1/2}} \frac{\left( 1 - \theta \right)^2 \sigma_{\min}^2}{\min \left\{ \left( 2C_\star \mu \kappa^2 \right)^{-2/3}, k \right\} \kappa^6 k^4} \tag{8.19}$$

According to Lemma 5.1, following inequality obtains

$$\left\| \frac{1}{\theta m} \boldsymbol{X}_0 \boldsymbol{X}_0^T - \boldsymbol{I} \right\|_2 \leq \delta \tag{8.20}$$

$$\leq 10 \sqrt{k \log m / m} \tag{8.21}$$

$$\leq \frac{20 \left( 1 - \theta \right) \sigma_{\min} \max \left\{ \left( 2C_\star \mu \kappa^2 \right)^{2/3}, k^{-1} \right\}}{C^{1/4} \kappa^3 k^{3/2} \log m} \tag{8.22}$$

$$\leq \frac{20 \sigma_{\min}}{C^{1/4} \kappa^3} \cdot \frac{\left( 1 - \theta \right) \left\| \boldsymbol{A}^T \boldsymbol{q} \right\|_4^4}{k^{3/2} \log m}, \quad \forall \boldsymbol{q} \in \hat{\mathcal{R}}_{2C_\star} \tag{8.23}$$

with probability no smaller than $1 - \varepsilon_0$ with $\varepsilon_0 = 2 \exp \left( -\theta k \right) + 24k \exp \left( -\frac{1}{144} \min \left\{ k, 3\sqrt{\theta m} \right\} \right) + 48k^{-7} + 48m^{-5}$.

We have $4\kappa^3\delta/\sigma_{\min} \leq 1/2$ whenever

$$C \geq \left(\frac{160\,(1-\theta)}{k^{3/2}\log m}\right)^4 \tag{8.24}$$

whence $\delta \leq 1/\left(8\kappa^2\right)$, and Lemma 5.3 implies that

$$\left\|\left(\frac{1}{\theta m}\boldsymbol{Y}\boldsymbol{Y}^T\right)^{-1/2}\boldsymbol{A}_0 - \left(\boldsymbol{A}_0\boldsymbol{A}_0^T\right)^{-1/2}\boldsymbol{A}_0\right\|_2$$

$$\leq 4\kappa^3\delta/\sigma_{\min} \tag{8.25}$$

$$\leq \frac{80\,(1-\theta)\left\|\boldsymbol{A}^T\boldsymbol{q}\right\|_4^4}{C^{1/4}k^{3/2}\log m}, \quad \forall \boldsymbol{q} \in \hat{\mathcal{R}}_{2C_\star}. \tag{8.26}$$

Moreover,

$$\|\boldsymbol{X}_0\|_2 \leq (\theta m)^{1/2}\sqrt{1+\delta} \tag{8.27}$$

$$\leq (\theta m)^{1/2}\,(1+\delta/2) \tag{8.28}$$

$$\leq \frac{17}{16}\,(\theta m)^{1/2}. \tag{8.29}$$

Finally, Lemma 2.5 implies that with probability no smaller than $1 - \varepsilon_B$, we have

$$\|\boldsymbol{x}_0\|_\infty \leq \sqrt{2}\log^{1/2}\left(\frac{2\theta m}{\varepsilon_B}\right). \tag{8.30}$$

**Upper Bound for $\Delta_1^H$ and $\Delta_2^H$.** With probability no smaller than $1 - \varepsilon_0 - \varepsilon_B$, the norms of $\boldsymbol{\eta}$ are upper bounded as in Lemma 2.7,

$$\Delta_1^H \leq \frac{3}{\theta m^2}\left\|\left(\frac{1}{\theta m}\boldsymbol{Y}\boldsymbol{Y}^T\right)^{-1/2}\boldsymbol{A}_0 - \boldsymbol{A}\right\|_2 \times$$

$$\|\boldsymbol{X}_0\|_2^2\,\|\boldsymbol{\eta}\|_\infty^2\left\|\boldsymbol{A}_0^T\left(\frac{1}{\theta m}\boldsymbol{Y}\boldsymbol{Y}^T\right)^{-1/2}\right\|_2 \tag{8.31}$$

$$\leq \frac{3}{\theta m^2}\cdot\frac{4\kappa^3\delta}{\sigma_{\min}}\cdot(1+\delta/2)^2\,\theta m\cdot$$

$$\left(1+\frac{4\kappa^3\delta}{\sigma_{\min}}\right)^3\frac{4k}{\theta m}\log\left(2\theta m/\varepsilon_B\right) \tag{8.32}$$

$$\leq \frac{3660}{C^{1/4}}\frac{1-\theta}{\theta m^2}\,\|\boldsymbol{\zeta}\|_4^4\cdot\frac{\log\left(2\theta m/\varepsilon_B\right)}{k^{1/2}\log m}. \tag{8.33}$$

A similar result holds for

$$\Delta_2^H \leq \frac{3}{\theta m^2}\,\|\boldsymbol{X}_0\|_2^2\,\left\|\mathrm{diag}\left(\boldsymbol{\eta}^{\circ 2}\right)\right\|_2 \times$$

$$\left\|\left(\frac{1}{\theta m}\boldsymbol{Y}\boldsymbol{Y}^T\right)^{-1/2}\boldsymbol{A}_0 - \boldsymbol{A}\right\|_2 \tag{8.34}$$

$$\leq \frac{2440}{C^{1/4}}\frac{1-\theta}{\theta m^2}\,\|\boldsymbol{\zeta}\|_4^4\cdot\frac{\log\left(2\theta m/\varepsilon_B\right)}{k^{1/2}\log m}. \tag{8.35}$$

To make $\Delta_1^H \leq \frac{c}{9}\frac{1-\theta}{\theta m^2}\left\|\boldsymbol{\zeta}\right\|_4^4$ and $\Delta_2^H \leq \frac{c}{9}\frac{1-\theta}{\theta m^2}\left\|\boldsymbol{\zeta}\right\|_4^4$, we require

$$C \geq \left(9 \times 3660 c^{-1}\frac{\log\left(2\theta m/\varepsilon_B\right)}{k^{1/2}\log m}\right)^4. \tag{8.36}$$

The right hand side is bounded by an absolute constant for all $m$.

**Upper Bound for $\Delta_3^H$.** With probability no smaller than $1 - \varepsilon_0 - \varepsilon_B$, the difference between $\bar{\boldsymbol{\eta}}^{\circ 2}$ and $\boldsymbol{\eta}^{\circ 2}$ is upper bounded as in Lemma 2.7,

$$\left\|\boldsymbol{\eta}^{\circ 2} - \bar{\boldsymbol{\eta}}^{\circ 2}\right\|_\infty$$
$$\leq \left\|\boldsymbol{\eta} - \bar{\boldsymbol{\eta}}\right\|_\infty \left\|\boldsymbol{\eta} + \bar{\boldsymbol{\eta}}\right\|_\infty \tag{8.37}$$
$$\leq \frac{4\kappa^3\delta}{\sigma_{\min}}\left(2 + \frac{4\kappa^3\delta}{\sigma_{\min}}\right)\frac{2k}{\theta m}\log\left(2\theta m/\varepsilon_B\right) \tag{8.38}$$
$$\leq \frac{5k}{\theta m}\log\left(2\theta m/\varepsilon_B\right) \cdot \frac{4\kappa^3\delta}{\sigma_{\min}}. \tag{8.39}$$

Therefore

$$\Delta_3^H = \frac{3}{\theta m^2}\left\|\boldsymbol{A}\boldsymbol{X}_0\operatorname{diag}\left(\boldsymbol{\eta}^{\circ 2} - \bar{\boldsymbol{\eta}}^{\circ 2}\right)\boldsymbol{X}_0^T\boldsymbol{A}^T\right\|_2 \tag{8.40}$$
$$\leq \frac{3}{\theta m^2}\left\|\boldsymbol{A}\right\|_2^2\left\|\boldsymbol{X}_0\right\|_2^2\left\|\operatorname{diag}\left(\boldsymbol{\eta}^{\circ 2} - \bar{\boldsymbol{\eta}}^{\circ 2}\right)\right\|_2 \tag{8.41}$$
$$\leq \frac{15k}{\theta m^2}\left(1 + \delta/2\right)^2\log\left(2\theta m/\varepsilon_B\right) \cdot \frac{4\kappa^3\delta}{\sigma_{\min}} \tag{8.42}$$
$$\leq \frac{1400\left(1-\theta\right)\log\left(2\theta m/\varepsilon_B\right)}{C^{1/4}\theta k^{1/2}m^2\log m}\left\|\boldsymbol{\zeta}\right\|_4^4. \tag{8.43}$$

Again, $\Delta_3^H$ is bounded by $\frac{c}{9}\frac{1-\theta}{\theta m^2}\left\|\boldsymbol{\zeta}\right\|_4^4$ whenever

$$C \geq \left(9 \times 1400 c^{-1}\frac{\log\left(2\theta m/\varepsilon_B\right)}{k^{1/2}\log m}\right)^4 \tag{8.44}$$

**Upper Bound for $\Delta_4^H$.** Recall that

$$\bar{\boldsymbol{\eta}} = \boldsymbol{Y}^T\left(\theta m \boldsymbol{A}_0\boldsymbol{A}_0^T\right)^{-1/2}\boldsymbol{q}, \tag{8.45}$$

then

$$\mathbb{E}\left[\boldsymbol{X}_0\operatorname{diag}\left(\bar{\boldsymbol{\eta}}^{\circ 2}\right)\boldsymbol{X}_0^T\right]$$
$$= \mathbb{E}\left[\frac{1}{\theta m}\boldsymbol{X}_0\operatorname{diag}\left(\boldsymbol{X}_0^T\boldsymbol{A}^T\boldsymbol{q}\right)^{\circ 2}\boldsymbol{X}_0^T\right] \tag{8.46}$$
$$= 3\left(1-\theta\right)\operatorname{diag}\left(\boldsymbol{A}^T\boldsymbol{q}\right)^{\circ 2} + 2\theta\boldsymbol{A}^T\boldsymbol{q}\boldsymbol{q}^T\boldsymbol{A}$$
$$+ \theta\left\|\boldsymbol{A}^T\boldsymbol{q}\right\|_2^2\boldsymbol{I}, \tag{8.47}$$

once including the projection $\boldsymbol{P}_{\boldsymbol{q}^\perp}$, we have

$$\boldsymbol{P}_{\boldsymbol{q}^\perp}\mathbb{E}\left[\boldsymbol{A}\boldsymbol{X}_0\operatorname{diag}\left(\bar{\boldsymbol{\eta}}^{\circ 2}\right)\boldsymbol{X}_0^T\boldsymbol{A}^T\right]\boldsymbol{P}_{\boldsymbol{q}^\perp} \tag{8.48}$$
$$= \boldsymbol{P}_{\boldsymbol{q}^\perp}\left[3\left(1-\theta\right)\boldsymbol{A}\operatorname{diag}\left(\boldsymbol{\zeta}^{\circ 2}\right)\boldsymbol{A}^T + \theta\boldsymbol{I}\right]\boldsymbol{P}_{\boldsymbol{q}^\perp}.$$

Therefore

$$\Delta_4^H = \frac{3}{\theta m^2}\left\|\boldsymbol{P}_{\boldsymbol{q}^\perp}\left[\boldsymbol{A}\boldsymbol{X}_0\operatorname{diag}\left(\bar{\boldsymbol{\eta}}^{\circ 2}\right)\boldsymbol{X}_0^T\boldsymbol{A}^T\right]\boldsymbol{P}_{\boldsymbol{q}^\perp}\right.$$

$$- \boldsymbol{P}_{\boldsymbol{q}^{\perp}} \left[ 3 \left( 1 - \theta \right) \boldsymbol{A} \operatorname{diag} \left( \boldsymbol{\zeta}^{\circ 2} \right) \boldsymbol{A}^T + \theta \boldsymbol{I} \right] \boldsymbol{P}_{\boldsymbol{q}^{\perp}} \Big\|_2 \tag{8.49}$$

$$\leq \frac{3}{\theta^2 m^2} \left\| \frac{1}{m} \boldsymbol{X}_0 \operatorname{diag} \left( \boldsymbol{X}_0^T \boldsymbol{\zeta} \right)^{\circ 2} \boldsymbol{X}_0^T - \mathbb{E} \left[ \cdot \right] \right\|_2 \tag{8.50}$$

Under the assumption for sample size that $m \geq C \left( 1 - \theta \right)^{-2} \kappa^4 \min \left\{ \left( 2 C_\star \mu \right)^{-2/3}, k \right\} k^3 \log^5 \left( \kappa k \right)$, applying Lemma 8.2, we have

$$\left\| \frac{1}{m} \boldsymbol{X}_0 \operatorname{diag} \left( \boldsymbol{X}_0^T \boldsymbol{\zeta} \right)^{\circ 2} \boldsymbol{X}_0^T - \mathbb{E} \left[ \cdot \right] \right\|_2 \leq \frac{c}{9} \theta \left( 1 - \theta \right) \| \boldsymbol{\zeta} \|_4^4 . \tag{8.51}$$

simultaneously at every $\boldsymbol{q} \in \hat{\mathcal{R}}_{2 C_\star}$ with probability no smaller than $1 - c_1 \exp \left( -k \right) - c_2 k^{-4}$.

**Upper Bound for $\Delta_5^H$.** Note that this term is essentially the difference between

$$\Delta_5^H = \left\| \left[ \langle \boldsymbol{q}, \nabla \psi \left( \boldsymbol{q} \right) \rangle - \frac{3 \left( 1 - \theta \right)}{\theta m^2} \| \boldsymbol{\zeta} \|_4^4 - \frac{3}{m^2} \right] \boldsymbol{P}_{\boldsymbol{q}^{\perp}} \right\|_2 \tag{8.52}$$

$$\leq \left| \langle \boldsymbol{q}, \nabla \psi \left( \boldsymbol{q} \right) \rangle - \frac{3 \left( 1 - \theta \right)}{\theta m^2} \| \boldsymbol{\zeta} \|_4^4 - \frac{3}{m^2} \right| \tag{8.53}$$

$$\leq \frac{1}{\theta^2 m^2} \left| \frac{1}{m} \left\| \boldsymbol{X}_0^T \boldsymbol{\zeta} \right\|_4^4 - 3 \theta \left( 1 - \theta \right) \| \boldsymbol{\zeta} \|_4^4 - 3 \theta^2 \right|$$

$$+ \left| \langle \boldsymbol{q}, \nabla \psi \left( \boldsymbol{q} \right) \rangle - \frac{1}{\theta^2 m^2} \left\| \boldsymbol{X}_0^T \boldsymbol{\zeta} \right\|_4^4 \right| \tag{8.54}$$

$$\leq \frac{1}{\theta^2 m^2} \left\| \frac{\boldsymbol{A} \boldsymbol{X}_0 \left( \boldsymbol{X}_0^T \boldsymbol{\zeta} \right)^{\circ 3}}{m} - 3 \theta \left( 1 - \theta \right) \boldsymbol{A}^T \boldsymbol{\zeta}^{\circ 3} - 3 \theta^2 \boldsymbol{q} \right\|_2$$

$$+ \frac{1}{m} \left| \| \boldsymbol{\eta} \|_4^4 - \| \bar{\boldsymbol{\eta}} \|_4^4 \right| \tag{8.55}$$

Recall that

$$\mathbb{E} \left[ \frac{1}{m} \boldsymbol{A} \boldsymbol{X}_0 \left( \boldsymbol{X}_0^T \boldsymbol{A}^T \boldsymbol{q} \right)^{\circ 3} \right]$$

$$= \mathbb{E} \left[ \boldsymbol{A} \boldsymbol{x}_i \left( \boldsymbol{x}_i^T \boldsymbol{A}^T \boldsymbol{q} \right)^3 \right] \tag{8.56}$$

$$= 3 \theta \left( 1 - \theta \right) \boldsymbol{A} \boldsymbol{\zeta}^{\circ 3} + 3 \theta^2 \boldsymbol{q}, \tag{8.57}$$

With similar argument as in Lemma 7.1, we can show that this term can be bounded by $\frac{c}{6} \frac{1-\theta}{\theta m^2} \| \boldsymbol{\eta} \|_4^4$ whenever

$$m \geq C' \frac{\min \left\{ \left( \mu \kappa^2 \right)^{-4/3}, k^2 \right\}}{\left( 1 - \theta \right)^2 \sigma_{\min}^2} \kappa^6 k^4 \log^3 \left( \frac{\kappa k}{\left( 1 - \theta \right) \sigma_{\min}} \right) . \tag{8.58}$$

Moreover, with probability $1 - \varepsilon_0 - \varepsilon_B$

$$\frac{1}{m} \left| \| \boldsymbol{\eta} \|_4^4 - \| \bar{\boldsymbol{\eta}} \|_4^4 \right|$$

$$\leq \frac{1}{m} \left| \langle \boldsymbol{\eta} - \bar{\boldsymbol{\eta}}, 4 \boldsymbol{\eta}^{\circ 3} \rangle \right| \tag{8.59}$$

$$\leq \frac{4}{m} \| \boldsymbol{\eta} - \bar{\boldsymbol{\eta}} \|_2 \| \boldsymbol{\eta} \|_6^3 \tag{8.60}$$

$$\leq \frac{16 \kappa^3 \delta}{\sigma_{\min} m} \left( 1 + \delta / 2 \right) \left( 1 + \frac{4 \kappa^3 \delta}{\sigma_{\min}} \right)^2 \frac{4k}{\theta m} \log \left( 2 \theta m / \varepsilon_B \right) \tag{8.61}$$

$$\leq \frac{153k}{\theta m^2} \log\left(2\theta m/\varepsilon_B\right) \cdot \frac{\kappa^3 \delta}{\sigma_{\min}} \tag{8.62}$$

$$\leq \frac{3060}{C^{1/4}} \frac{(1-\theta)}{\theta m^2} \left\|\boldsymbol{\zeta}\right\|_4^4 \cdot \frac{\log\left(2\theta m/\varepsilon_B\right)}{k^{1/2} \log m}, \tag{8.63}$$

which is bounded by $\frac{c}{6} \frac{1-\theta}{\theta m^2} \left\|\boldsymbol{\zeta}\right\|_4^4$ whenever

$$C \geq \left(6 \times 3060 c^{-1} \frac{(1-\theta) \log\left(2\theta m/\varepsilon_B\right)}{k^{1/2} \log m}\right)^4. \tag{8.64}$$

The right hand side is bounded by an absolute constant for all $m$.

Adding up failure probabilities, we have that with probability larger than $1 - c_2 \exp\left(-k\right) - c_2 k^{-4} - \varepsilon_0$,

$$\left\|\text{Hess}\left[\psi\right]\left(\boldsymbol{q}\right) - \frac{3\left(1-\theta\right)}{\theta m^2} \text{Hess}\left[\varphi\right]\left(\boldsymbol{q}\right)\right\|_2 \leq c \frac{1-\theta}{\theta m^2} \left\|\boldsymbol{A}^T \boldsymbol{q}\right\|_4^4 \tag{8.65}$$

holds as desired for all $\boldsymbol{q} \in \hat{\mathcal{R}}_{2C_\star}$, where $\varepsilon_0 = 2\exp\left(-\theta k\right) + 24k\exp\left(-\frac{1}{144}\min\left\{k, 3\sqrt{\theta m}\right\}\right) + 48k^{-7} + 48m^{-5}$.
∎

## 8.1 Proof of Lemma 8.2

**Lemma 8.2** *Suppose $\boldsymbol{x}_0 \sim_{\text{i.i.d.}} \text{BG}\left(\theta\right)$. There exist constants $C > 0$ that whenever*

$$m \geq C \frac{\min\left\{\left(2C_\star \mu \kappa^2\right)^{-4/3}, k^2\right\}}{\left(1-\theta\right)^2} k^4 \log^3\left(\kappa k\right), \tag{8.66}$$

*and $\theta k > 1$, then with probability no smaller than $1 - c_1 \exp\left(-k\right) - c_2 k^{-4}$,*

$$\left\|\frac{1}{m} \boldsymbol{X}_0 \text{diag}\left(\boldsymbol{X}_0^T \boldsymbol{A}^T \boldsymbol{q}\right)^{\circ 2} \boldsymbol{X}_0^T - \mathbb{E}\left[\cdot\right]\right\|_2$$

$$\leq c\theta\left(1-\theta\right) \left\|\boldsymbol{A}^T \boldsymbol{q}\right\|_4^4, \tag{8.67}$$

*holds for all $\boldsymbol{q} \in \hat{\mathcal{R}}_{2C_\star}$ with positive constant $c \leq 0.005 \leq \left(1 - 6c_\star - 36c_\star^2 - 24c_\star^3\right)/3$.*

**Proof** The proof strategy for the finite sample concentration of the Hessian is similar to that of the gradient as presented in Lemma 7.2. For simplicity, we will only demonstrate some key steps here, please refer to Lemma 7.2 for detailed arguments.

Again, from Lemma 2.5, the coefficient satisfies $\left\|\boldsymbol{x}_0\right\|_\infty \leq B$ with probability no smaller than $1 - 2\theta m e^{-B^2/2}$. We write $\bar{\boldsymbol{x}}_0(i) = \boldsymbol{x}_0(i)\mathbb{1}_{|\boldsymbol{x}_0(i)| \leq B}$, and let $\bar{\boldsymbol{X}}_0$ denote the circulant matrix generated by the truncated vector $\bar{\boldsymbol{x}}_0$. Denote

$$\boldsymbol{H}_E = \mathbb{E}\left[\frac{1}{m} \boldsymbol{X}_0 \text{diag}\left(\boldsymbol{X}_0^T \boldsymbol{A}^T \boldsymbol{q}\right)^{\circ 2} \boldsymbol{X}_0^T\right], \tag{8.68}$$

$$\bar{\boldsymbol{H}}_E = \mathbb{E}\left[\frac{1}{m} \bar{\boldsymbol{X}}_0 \text{diag}\left(\bar{\boldsymbol{X}}_0^T \boldsymbol{A}^T \boldsymbol{q}\right)^{\circ 2} \bar{\boldsymbol{X}}_0^T\right], \tag{8.69}$$

then

$$\mathbb{P}\left[\left\|\frac{\boldsymbol{X}_0 \text{diag}\left(\boldsymbol{X}_0^T \boldsymbol{\zeta}\right)^{\circ 2} \boldsymbol{X}_0^T}{m} - \boldsymbol{H}_E\right\|_2 \geq c\theta(1-\theta)\|\boldsymbol{\zeta}\|_4^4\right]$$

$$\leq \mathbb{P}\left[\left\|\frac{\bar{\boldsymbol{X}}_0 \operatorname{diag}\left(\bar{\boldsymbol{X}}_0^T \boldsymbol{\zeta}\right)^{\circ 2} \bar{\boldsymbol{X}}_0^T}{m} - \boldsymbol{H}_E\right\|_2 \geq c\theta(1-\theta)\|\boldsymbol{\zeta}\|_4^4\right]$$

$$+ 2\theta m e^{-B^2/2} + 2m \exp\left(-\frac{3}{4}\theta k \log m\right) \tag{8.70}$$

while via triangle inequality,

$$\left\|\frac{1}{m}\bar{\boldsymbol{X}}_0 \operatorname{diag}\left(\bar{\boldsymbol{X}}_0^T \boldsymbol{A}^T \boldsymbol{q}\right)^{\circ 2} \bar{\boldsymbol{X}}_0^T - \boldsymbol{H}_E\right\|_2$$

$$\leq \left\|\frac{1}{m}\bar{\boldsymbol{X}}_0 \operatorname{diag}\left(\bar{\boldsymbol{X}}_0^T \boldsymbol{A}^T \boldsymbol{q}\right)^{\circ 2} \bar{\boldsymbol{X}}_0^T - \bar{\boldsymbol{H}}_E\right\|_2$$

$$+ \left\|\bar{\boldsymbol{H}}_E - \boldsymbol{H}_E\right\|_2. \tag{8.71}$$

**Truncation Level.** Next, we choose a large enough entry-wise truncation level $B$ such that the expectation of the Hessian $\mathbb{E}\left[\boldsymbol{X}_0 \operatorname{diag}\left(\boldsymbol{X}_0^T \boldsymbol{A}^T \boldsymbol{q}\right)^{\circ 2} \boldsymbol{X}_0^T\right]$ is close to that of its truncation $\mathbb{E}\left[\bar{\boldsymbol{X}}_0 \operatorname{diag}\left(\bar{\boldsymbol{X}}_0^T \boldsymbol{A}^T \boldsymbol{q}\right)^{\circ 2} \bar{\boldsymbol{X}}_0^T\right]$. Moreover, we introduce following events notation

$$\mathcal{E}_i \doteq \left\{\|\boldsymbol{x}_i\|_\infty > B \ \cup \ \|\boldsymbol{x}_i\|_0 > 4\theta k \log m\right\}, \tag{8.72}$$

then

$$\left\|\bar{\boldsymbol{H}}_E - \boldsymbol{H}_E\right\|_2$$

$$= \left\|\mathbb{E}\left[\frac{1}{m}\sum_i \langle \boldsymbol{x}_i, \boldsymbol{\zeta}\rangle^2 \boldsymbol{x}_i \boldsymbol{x}_i^T \cdot \mathbf{1}_{\boldsymbol{E}_i}\right]\right\|_F \tag{8.73}$$

$$\leq \frac{1}{m}\sum_i \left\|\mathbb{E}\left[\langle \boldsymbol{x}_i, \boldsymbol{\zeta}\rangle^2 \boldsymbol{x}_i \boldsymbol{x}_i^T \cdot \mathbf{1}_{\boldsymbol{E}_i}\right]\right\|_F \tag{8.74}$$

$$\leq \frac{1}{m}\sum_i \left(\mathbb{E}\left[\left\|\langle \boldsymbol{x}_i, \boldsymbol{\zeta}\rangle^2 \boldsymbol{x}_i \boldsymbol{x}_i^T\right\|_F^2\right] \cdot \mathbb{E}\left[\mathbf{1}_{\boldsymbol{E}_i}\right]\right)^{1/2} \tag{8.75}$$

$$\leq \left(\mathbb{E}\left[\|\boldsymbol{x}_i\|_2^8\right]\right)^{1/2} \times$$

$$\sqrt{\mathbb{E}\left[\mathbf{1}_{\|\boldsymbol{x}_i\|_\infty > B}\right] + \mathbb{E}\left[\mathbf{1}_{\|\boldsymbol{x}_i\|_0 > 4\theta k \log m}\right]} \tag{8.76}$$

$$\leq 50k^2\sqrt{4\theta k e^{-B^2/2} + \exp\left(-\frac{3}{4}\theta k \log m\right)} \tag{8.77}$$

By setting

$$B \geq C' \log^{1/2}\left(\frac{k^7}{\theta(1-\theta)^2}\right) \tag{8.78}$$

we have

$$\theta k e^{-B^2/2} \leq c'\theta^2 (1-\theta)^2 \frac{\|\boldsymbol{\zeta}\|_4^8}{k^4} \tag{8.79}$$

In addition, whenever

$$\theta k \geq \frac{4}{3 \log m} \log\left(\frac{400^2 k^4}{c^2 \theta^2 (1-\theta)^2 \|\boldsymbol{\zeta}\|_4^8}\right), \tag{8.80}$$

we have

$$\exp\left(-\tfrac{3}{4}\theta k \log m\right) \le \frac{1}{2}\left(\frac{c\theta(1-\theta)}{100}\right)^2 \frac{\|\boldsymbol{\zeta}\|_4^8}{k^4}. \tag{8.81}$$

Hence,

$$\sqrt{4\theta k e^{-B^2/2} + \exp\left(-\tfrac{3}{4}\theta k \log m\right)} \le \frac{c\theta(1-\theta)}{100k^2}\|\boldsymbol{\zeta}\|_4^4. \tag{8.82}$$

Therefore, we can obtain that

$$\left\|\bar{\boldsymbol{H}}_E - \boldsymbol{H}_E\right\|_2 \le \frac{c}{2}\theta(1-\theta)\|\boldsymbol{\zeta}\|_4^4 \tag{8.83}$$

always holds, hence

$$\mathbb{P}\left[\left\|\frac{\bar{\boldsymbol{X}}_0 \operatorname{diag}\left(\bar{\boldsymbol{X}}_0^T\boldsymbol{\zeta}\right)^{\circ 2}\bar{\boldsymbol{X}}_0^T}{m} - \boldsymbol{H}_E\right\|_2 \ge c\theta(1-\theta)\|\boldsymbol{\zeta}\|_4^4\right]$$

$$\le \mathbb{P}\left[\left\|\frac{\bar{\boldsymbol{X}}_0 \operatorname{diag}\left(\bar{\boldsymbol{X}}_0^T\boldsymbol{\zeta}\right)^{\circ 2}\bar{\boldsymbol{X}}_0^T}{m} - \bar{\boldsymbol{H}}_E\right\|_2 \ge \frac{c}{2}\theta(1-\theta)\|\boldsymbol{\zeta}\|_4^4\right]. \tag{8.84}$$

**Independent Sub-matrices.** As we did in Lemma 7.2, we remove the dependence in $\boldsymbol{X}_0$ by sampling every $2k-1$ column such that

$$\boldsymbol{X}_0\boldsymbol{\Pi} = [\boldsymbol{X}_1, \boldsymbol{X}_2, \cdots, \boldsymbol{X}_{2k-1}], \tag{8.85}$$

where

$$\boldsymbol{X}_i = \left[\boldsymbol{x}_i, \boldsymbol{x}_{i+(2k-1)}, \cdots, \boldsymbol{x}_{i+(m-2k-1)}\right], \tag{8.86}$$

and $\boldsymbol{\Pi}$ is a certain permutation of the columns of $\boldsymbol{X}_0$.

Applying Bernstein inequality for matrix variables as in Lemma 9.7, with $\boldsymbol{M}_i = \left\langle\bar{\boldsymbol{x}}_i, \boldsymbol{A}^T\boldsymbol{q}\right\rangle^2 \bar{\boldsymbol{x}}_i\bar{\boldsymbol{x}}_i^T \in \mathbb{R}^{(2k-1)\times(2k-1)}$. Since

$$\|\boldsymbol{M}_i\|_2 = \left\|\left\langle\bar{\boldsymbol{x}}_i, \boldsymbol{A}^T\boldsymbol{q}\right\rangle^2 \bar{\boldsymbol{x}}_i\bar{\boldsymbol{x}}_i^T\right\|_2 \tag{8.87}$$

$$\le \|\bar{\boldsymbol{x}}_i\|_2^4 \tag{8.88}$$

$$\le 4B^4k^2 \tag{8.89}$$

and

$$\|\mathbb{E}\left[\boldsymbol{M}_i\boldsymbol{M}_i^*\right]\| = \|\mathbb{E}\left[\boldsymbol{M}_i^*\boldsymbol{M}_i\right]\| \tag{8.90}$$

$$= \left\|\mathbb{E}\left[\left\langle\bar{\boldsymbol{x}}_i, \boldsymbol{A}^T\boldsymbol{q}\right\rangle^4 \bar{\boldsymbol{x}}_i\bar{\boldsymbol{x}}_i^T\bar{\boldsymbol{x}}_i\bar{\boldsymbol{x}}_i^T\right]\right\| \tag{8.91}$$

$$= \left\|\mathbb{E}\left[\left\langle\bar{\boldsymbol{x}}_i, \boldsymbol{\zeta}\right\rangle^4 \|\bar{\boldsymbol{x}}_i\|_2^2 \bar{\boldsymbol{x}}_i\bar{\boldsymbol{x}}_i^T\right]\right\| \tag{8.92}$$

$$\le \mathbb{E}\left[\left\langle\bar{\boldsymbol{x}}_i, \boldsymbol{\zeta}\right\rangle^4 \|\bar{\boldsymbol{x}}_i\|_2^4\right], \tag{8.93}$$

we obtain the following upper bound:

$$\mathbb{E}\left[\left\langle\bar{\boldsymbol{x}}_i, \boldsymbol{\zeta}\right\rangle^4 \|\bar{\boldsymbol{x}}_i\|_2^4\right]$$

$$= \mathbb{E}\left[\sum_{j_1,j_2}^{2k-1} \bar{\boldsymbol{x}}_i(j_1)^2\, \bar{\boldsymbol{x}}_i(j_2)^2 \sum_{j_3,\cdots,j_6} \prod_{\ell=3}^{6} \bar{\boldsymbol{x}}_i(j_\ell)\, \boldsymbol{\zeta}_{j_\ell}\right] \tag{8.94}$$

$$= 3\mathbb{E}\left[\sum_{j_1,j_2}^{2k-1} \bar{\boldsymbol{x}}_i(j_1)^2\, \bar{\boldsymbol{x}}_i(j_2)^2 \sum_{j_3 \neq j_4} \bar{\boldsymbol{x}}_i(j_3)^2\, \boldsymbol{\zeta}_{j_3}^2 \bar{\boldsymbol{x}}_i(j_4)^2\, \boldsymbol{\zeta}_{j_4}^2\right]$$

$$+ \mathbb{E}\left[\sum_{j_1,j_2}^{2k-1} \bar{\boldsymbol{x}}_i(j_1)^2\, \bar{\boldsymbol{x}}_i(j_2)^2 \cdot \sum_{j_3} \bar{\boldsymbol{x}}_i(j_3)^4\, \boldsymbol{\zeta}_{j_3}^4\right] \tag{8.95}$$

$$= 3\mathbb{E}\left[\sum_{\substack{j_1 \neq j_2 \\ \neq j_3 \neq j_4}} \bar{\boldsymbol{x}}_i(j_1)^2\, \bar{\boldsymbol{x}}_i(j_2)^2\, \bar{\boldsymbol{x}}_i(j_3)^2\, \boldsymbol{\zeta}_{j_3}^2 \bar{\boldsymbol{x}}_i(j_4)^2\, \boldsymbol{\zeta}_{j_4}^2\right]$$

$$+ 3\mathbb{E}\left[\sum_{j_1 \neq j_2 \neq j_3} \bar{\boldsymbol{x}}_i(j_1)^4\, \bar{\boldsymbol{x}}_i(j_2)^2\, \boldsymbol{\zeta}_{j_2}^2 \bar{\boldsymbol{x}}_i(j_3)^2\, \boldsymbol{\zeta}_{j_3}^2\right]$$

$$+ 6\mathbb{E}\left[\sum_{j_1 \neq j_2} \bar{\boldsymbol{x}}_i(j_1)^6\, \boldsymbol{\zeta}_{j_1}^2 \bar{\boldsymbol{x}}_i(j_2)^2\, \boldsymbol{\zeta}_{j_2}^2\right]$$

$$+ 6\mathbb{E}\left[\sum_{j_1 \neq j_2 \neq j_3} \bar{\boldsymbol{x}}_i(j_1)^2\, \bar{\boldsymbol{x}}_i(j_2)^4\, \boldsymbol{\zeta}_{j_2}^2 \bar{\boldsymbol{x}}_i(j_3)^2\, \boldsymbol{\zeta}_{j_3}^2\right]$$

$$+ 6\mathbb{E}\left[\sum_{j_1 \neq j_2} \bar{\boldsymbol{x}}_i(j_1)^4\, \boldsymbol{\zeta}_{j_1}^2 \bar{\boldsymbol{x}}_i(j_2)^4\, \boldsymbol{\zeta}_{j_2}^2\right]$$

$$+ 2\mathbb{E}\left[\sum_{j_1 \neq j_2} \bar{\boldsymbol{x}}_i(j_1)^2\, \bar{\boldsymbol{x}}_i(j_2)^6\, \boldsymbol{\zeta}_{j_2}^4\right]$$

$$+ \mathbb{E}\left[\sum_{j_1 \neq j_2 \neq j_3} \bar{\boldsymbol{x}}_i(j_1)^2\, \bar{\boldsymbol{x}}_i(j_2)^2\, \bar{\boldsymbol{x}}_i(j_3)^4\, \boldsymbol{\zeta}_{j_3}^4\right]$$

$$+ \mathbb{E}\left[\sum_{j_1 \neq j_2} \bar{\boldsymbol{x}}_i(j_1)^4\, \bar{\boldsymbol{x}}_i(j_2)^4\, \boldsymbol{\zeta}_{j_2}^4\right]$$

$$+ \mathbb{E}\left[\sum_{j} \bar{\boldsymbol{x}}_i(j)^8\, \boldsymbol{\zeta}_j^4\right] \tag{8.96}$$

$$\leq \left(105\theta + 18\theta^2 k + 60\theta^2 k + 12\theta^3 k^2\right) \|\boldsymbol{\zeta}\|_4^4$$

$$+ 3\left(21\theta^2 + 30\theta^2 + 4\theta^4 k^2 + 12\theta^2 k\right) \|\boldsymbol{\zeta}\|_2^4 \tag{8.97}$$

$$\leq C\theta^3 k^2 \tag{8.98}$$

Assuming $\theta m \geq 1$, hence

$$\sigma^2 = C\theta^3 km. \tag{8.99}$$

Setting $t = \frac{c}{2}\frac{\theta(1-\theta)m\|\boldsymbol{\zeta}\|_4^4}{2k-1}$ in Matrix Bernstein gives that

$$\mathbb{P}\left[\left\|\bar{\boldsymbol{X}}_i\left(\bar{\boldsymbol{X}}_i^T \boldsymbol{\zeta}\right)^{\circ 3} - \mathbb{E}\left[\cdot\right]\right\|_2 > t\right]$$

$$\leq 8k \exp\left(\frac{-t^2/2}{C\theta^3 km + C'B^4\theta^2 k^2 t}\right), \tag{8.100}$$

we obtain that

$$\mathbb{P}\left[\left\|\frac{\bar{\boldsymbol{X}}_i \operatorname{diag}\left(\bar{\boldsymbol{X}}_i^T \boldsymbol{\zeta}\right)^{\circ 2} \bar{\boldsymbol{X}}_i^T}{m} - \mathbb{E}\left[\cdot\right]\right\|_2 > c\frac{\theta\left(1-\theta\right)\|\boldsymbol{\zeta}\|_4^6}{2k-1}\right]$$

$$\leq 8k \exp\left(-\frac{cm\left(1-\theta\right)^2 \|\boldsymbol{\zeta}\|_4^8}{\theta k^3 + \theta\left(1-\theta\right)B^4 k^3 \|\boldsymbol{\zeta}\|_4^4}\right) \tag{8.101}$$

$\varepsilon$-**Net Covering** To obtain a probability bound for all $\boldsymbol{q} \in \mathbb{S}^{k-1}$, we choose a set of $\boldsymbol{\zeta}_n = \boldsymbol{A}^T \boldsymbol{q}_n$ with $n = 1, \cdots, N$. Since for any $\boldsymbol{q}, \boldsymbol{q}' \in \mathbb{S}^{k-1}$ and $\boldsymbol{\zeta}' = \boldsymbol{A}^T \boldsymbol{q}'$, we have

$$\left\|\frac{\bar{\boldsymbol{X}}_i \operatorname{diag}\left(\bar{\boldsymbol{X}}_i^T \boldsymbol{\zeta}\right)^{\circ 2} \bar{\boldsymbol{X}}_i^T}{m} - \frac{\bar{\boldsymbol{X}}_i \operatorname{diag}\left(\bar{\boldsymbol{X}}_i^T \boldsymbol{\zeta}'\right)^{\circ 2} \bar{\boldsymbol{X}}_i^T}{m}\right\|_2$$

$$= \frac{1}{m}\left\|\bar{\boldsymbol{X}}_i \operatorname{diag}\left[\left(\bar{\boldsymbol{X}}_i^T \boldsymbol{\zeta}\right)^{\circ 2} - \left(\bar{\boldsymbol{X}}_i^T \boldsymbol{\zeta}'\right)^{\circ 2}\right] \bar{\boldsymbol{X}}_i^T\right\|_2 \tag{8.102}$$

$$\leq \frac{\|\bar{\boldsymbol{X}}_i\|_2^2}{m}\left\|\operatorname{diag}\left[\left(\bar{\boldsymbol{X}}_i^T \boldsymbol{\zeta}\right)^{\circ 2} - \left(\bar{\boldsymbol{X}}_i^T \boldsymbol{\zeta}'\right)^{\circ 2}\right]\right\|_2 \tag{8.103}$$

$$\leq \frac{\|\bar{\boldsymbol{X}}_i\|_2^2}{m}\left\|\bar{\boldsymbol{X}}_i^T \boldsymbol{\zeta} + \bar{\boldsymbol{X}}_i^T \boldsymbol{\zeta}'\right\|_\infty \left\|\bar{\boldsymbol{X}}_i^T \boldsymbol{\zeta} - \bar{\boldsymbol{X}}_i^T \boldsymbol{\zeta}'\right\|_\infty \tag{8.104}$$

$$\leq L \|\boldsymbol{q} - \boldsymbol{q}'\|_2 \tag{8.105}$$

Then the Lipschitz constant $L$ is upper bounded by

$$L \leq \frac{\|\bar{\boldsymbol{X}}_i\|_2^2}{m}\left\|\bar{\boldsymbol{X}}_i^T \boldsymbol{A}^T\right\|_2 \left(\left\|\bar{\boldsymbol{X}}_i^T \boldsymbol{\zeta}\right\|_\infty + \left\|\bar{\boldsymbol{X}}_i^T \boldsymbol{\zeta}'\right\|_\infty\right) \tag{8.106}$$

$$\leq \frac{2}{m}\|\bar{\boldsymbol{X}}_i\|_2^4 \tag{8.107}$$

$$\leq 8\theta^2 mB^4. \tag{8.108}$$

With triangle inequality, we have

$$\left\|\frac{\bar{\boldsymbol{X}}_i \operatorname{diag}\left(\bar{\boldsymbol{X}}_i^T \boldsymbol{\zeta}\right)^{\circ 2} \bar{\boldsymbol{X}}_i^T}{m} - \mathbb{E}\left[\cdot\right]\right\|_2$$

$$\leq \left\|\frac{\bar{\boldsymbol{X}}_i \operatorname{diag}\left(\bar{\boldsymbol{X}}_i^T \boldsymbol{\zeta}\right)^{\circ 2}\bar{\boldsymbol{X}}_i^T}{m} - \frac{\bar{\boldsymbol{X}}_i \operatorname{diag}\left(\bar{\boldsymbol{X}}_i^T \boldsymbol{\zeta}_n\right)^{\circ 2}\bar{\boldsymbol{X}}_i^T}{m}\right\|_2$$

$$+ \left\|\mathbb{E}\left[\frac{\bar{\boldsymbol{X}}_i \operatorname{diag}\left(\bar{\boldsymbol{X}}_i^T \boldsymbol{\zeta}\right)^{\circ 2}\bar{\boldsymbol{X}}_i^T}{m}\right] - \mathbb{E}\left[\frac{\bar{\boldsymbol{X}}_i \operatorname{diag}\left(\bar{\boldsymbol{X}}_i^T \boldsymbol{\zeta}_n\right)^{\circ 2}\bar{\boldsymbol{X}}_i^T}{m}\right]\right\|_2$$

$$+ \left\|\frac{\bar{\boldsymbol{X}}_i \operatorname{diag}\left(\bar{\boldsymbol{X}}_i^T \boldsymbol{\zeta}_n\right)^{\circ 2}\bar{\boldsymbol{X}}_i^T}{m} - \mathbb{E}\left[\frac{\bar{\boldsymbol{X}}_i \operatorname{diag}\left(\bar{\boldsymbol{X}}_i^T \boldsymbol{\zeta}_n\right)^{\circ 2}\bar{\boldsymbol{X}}_i^T}{m}\right]\right\|_2 \tag{8.109}$$

$$\leq \left\|\frac{\bar{\boldsymbol{X}}_i \left(\bar{\boldsymbol{X}}_i^T \boldsymbol{\zeta}_n\right)^{\circ 2} \bar{\boldsymbol{X}}_i^T}{m} - \mathbb{E}\left[\frac{\bar{\boldsymbol{X}}_i \left(\bar{\boldsymbol{X}}_i^T \boldsymbol{\zeta}_n\right)^{\circ 2} \bar{\boldsymbol{X}}_i^T}{m}\right]\right\|_2$$

$$+ 2L\varepsilon \tag{8.110}$$

Next, we are going to choose the $\varepsilon$-net to cover the sphere of $\boldsymbol{q}$ with

$$\varepsilon = \frac{c}{4} \frac{\theta\left(1-\theta\right)}{\left(2k-1\right)L} \min_{\boldsymbol{q}\in\mathbb{S}^{k-1}} \|\boldsymbol{\zeta}\|_4^4, \tag{8.111}$$

hence the number of samples $N$ is bounded by

$$N = \left(\frac{3}{\varepsilon}\right)^k \tag{8.112}$$

$$\leq \exp\left(-k\ln\varepsilon\right) \tag{8.113}$$

$$\leq C\exp\left[k\log\left(\frac{\theta B^4 k^2 m}{1-\theta}\right)\right]. \tag{8.114}$$

For $n = 1,\cdots,N$, denote

$$P_i\left(\boldsymbol{q}_n\right) =$$
$$\mathbb{P}\left[\left\|\frac{\bar{\boldsymbol{X}}_i \operatorname{diag}\left(\bar{\boldsymbol{X}}_i^T\boldsymbol{\zeta}_n\right)^{\circ 2}\bar{\boldsymbol{X}}_i^T}{m} - \mathbb{E}[\cdot]\right\|_2 \geq \frac{c\theta(1-\theta)\|\boldsymbol{\zeta}_n\|_4^4}{4(2k-1)}\right] \tag{8.115}$$

together with union bound over all $\boldsymbol{q}_n$ , we obtain that,

$$\mathbb{P}\left[\sup_{\boldsymbol{q}\in\hat{\mathcal{R}}_{2C_\star}} \frac{\left\|\frac{\bar{\boldsymbol{X}}_i \operatorname{diag}\left(\bar{\boldsymbol{X}}_i^T\boldsymbol{\zeta}\right)^{\circ 2}\bar{\boldsymbol{X}}_i^T}{m} - \mathbb{E}\left[\cdot\right]\right\|_2}{\|\boldsymbol{\zeta}\|_4^4} \geq \frac{c\theta\left(1-\theta\right)}{2(2k-1)}\right]$$

$$\leq \sum_{\boldsymbol{q}_n\in\hat{\mathcal{R}}_{2C_\star}} P_i\left(\boldsymbol{q}_n\right) \tag{8.116}$$

$$\leq N \max_{\boldsymbol{q}_n\in\hat{\mathcal{R}}_{2C_\star}} P_i\left(\boldsymbol{q}_n\right) \tag{8.117}$$

$$\leq 8k \sup_{\boldsymbol{q}\in\hat{\mathcal{R}}_{2C_\star}} \exp\left(-\frac{cm\left(1-\theta\right)^2\|\boldsymbol{\zeta}\|_4^8}{\theta k^3 + \theta\left(1-\theta\right)B^4 k^3\|\boldsymbol{\zeta}\|_4^4}\right) \times$$
$$\exp\left(k\ln\left(\frac{3}{\varepsilon}\right)\right). \tag{8.118}$$

Hence

$$\mathbb{P}\left[\sup_{\boldsymbol{q}\in\hat{\mathcal{R}}_{2C_\star}} \frac{\left\|\frac{\bar{\boldsymbol{X}}_0 \operatorname{diag}\left(\bar{\boldsymbol{X}}_0^T\boldsymbol{\zeta}\right)^{\circ 2}\bar{\boldsymbol{X}}_0^T}{m} - \mathbb{E}\left[\cdot\right]\right\|_2}{\|\boldsymbol{\zeta}\|_4^4} \geq \frac{c}{2}\theta(1-\theta)\right]$$

$$\leq \sum_i \mathbb{P}\left[\sup_{\boldsymbol{q}\in\hat{\mathcal{R}}_{2C_\star}} \frac{\left\|\frac{\bar{\boldsymbol{X}}_i \operatorname{diag}\left(\bar{\boldsymbol{X}}_i^T\boldsymbol{\zeta}\right)^{\circ 2}\bar{\boldsymbol{X}}_i^T}{m} - \mathbb{E}\left[\cdot\right]\right\|_2}{\|\boldsymbol{\zeta}\|_4^4} \geq \frac{c\theta(1-\theta)}{2(2k-1)}\right] \tag{8.119}$$

$$\leq (2k-1)\max_i$$
$$\mathbb{P}\left[\sup_{\boldsymbol{q}\in\hat{\mathcal{R}}_{2C_\star}} \frac{\left\|\frac{\bar{\boldsymbol{X}}_i \operatorname{diag}\left(\bar{\boldsymbol{X}}_i^T\boldsymbol{\zeta}\right)^{\circ 2}\bar{\boldsymbol{X}}_i^T}{m} - \mathbb{E}\left[\cdot\right]\right\|_2}{\|\boldsymbol{\zeta}\|_4^4} \geq \frac{c\theta\left(1-\theta\right)}{2(2k-1)}\right] \tag{8.120}$$

$$\leq 16k^2 \sup_{\boldsymbol{q} \in \hat{\mathcal{R}}_{2C_\star}} \exp\left(-\frac{c'm(1-\theta)^2 \|\boldsymbol{\zeta}\|_4^8}{\theta k^3 + \theta(1-\theta)B^4 k^3 \|\boldsymbol{\zeta}\|_4^4}\right) \times$$
$$\exp\left(k \ln\left(\frac{3}{\varepsilon}\right)\right) \tag{8.121}$$

Therefore, by taking

$$m \geq \frac{C\theta}{(1-\theta)^2} \min\left\{(2C_\star \mu \kappa^2)^{-4/3}, k^2\right\} k^4 \log^3 k \tag{8.122}$$

$$\geq C'\theta k \log\left(\frac{\theta k m B^4}{(1-\theta)\|\boldsymbol{\zeta}\|_4^4}\right) \frac{k^3 + (1-\theta)B^4 k^3 \|\boldsymbol{\zeta}\|_4^4}{(1-\theta)^2 \|\boldsymbol{\zeta}\|_4^8} \tag{8.123}$$

and adding up failure probability, we obtain

$$\left\|\frac{1}{m}\boldsymbol{X}_0 \operatorname{diag}\left(\boldsymbol{X}_0^T \boldsymbol{A}^T \boldsymbol{q}\right)^{\circ 2} \boldsymbol{X}_0^T - \mathbb{E}[\cdot]\right\|_2$$
$$\leq c\theta(1-\theta)\left\|\boldsymbol{A}^T \boldsymbol{q}\right\|_4^4 \tag{8.124}$$

with probability no smaller than $1 - c_1 \exp(-k) - c_2 \theta(1-\theta)^2 k^{-4} - c_3 \exp(-\theta k)$.

∎

# 9 Tools

**Lemma 9.1 (Moments of the Gaussian Random Variables)** *If $X \sim \mathcal{N}(0, \sigma^2)$, then it holds for all integer $p \geq 1$ that*

$$\mathbb{E}[|X|^p] = \sigma^p (p-1)!! \left[\sqrt{\frac{2}{\pi}} \mathbb{1}_{p \text{ odd}} + \mathbb{1}_{p \text{ even}}\right] \tag{9.1}$$

$$\leq \sigma^p (p-1)!!. \tag{9.2}$$

**Lemma 9.2 (Moments of the $\chi^2$ Random Variables)** *If $X \sim \chi^2(n)$, then it holds for all integer $p \geq 1$,*

$$\mathbb{E}[X^p] = 2^p \frac{\Gamma(p+n/2)}{\Gamma(n/2)} \tag{9.3}$$

$$= \prod_{k=1}^{p}(n+2k-2) \leq p!(2n)^p/2 \tag{9.4}$$

**Lemma 9.3 (Moments of the $\chi$ Random Variables)** *If $X \sim \chi(n)$, then it holds for all integer $p \geq 1$,*

$$\mathbb{E}[X^p] = 2^{p/2} \frac{\Gamma(p/2+n/2)}{\Gamma(n/2)} \leq p!!n^{p/2}. \tag{9.5}$$

**Lemma 9.4 (Moment-Control Bernstein's Inequality for Scalar RVs, Theorem 2.10 of [FR13])** *Let $X_1, \ldots, X_p$ be i.i.d. real-valued random variables. Suppose that there exist some positive number $R$ and $\sigma^2$ such that*

$$\mathbb{E}[|X_k|^m] \leq \frac{m!}{2}\sigma^2 R^{m-2}, \text{ for all integers } m \geq 2.$$

*Let $S \doteq \frac{1}{p}\sum_{k=1}^{p} X_k$, then for all $t > 0$, it holds that*

$$\mathbb{P}[|S - \mathbb{E}[S]| \geq t] \leq 2\exp\left(-\frac{pt^2}{2\sigma^2 + 2Rt}\right). \tag{9.6}$$

**Corollary 9.5 (Moment-Control Bernstein's Inequality for Vector RVs, Corollary A.10 of [SQW15])** *Let $\boldsymbol{x}_1, \ldots, \boldsymbol{x}_p \in \mathbb{R}^d$ be i.i.d. random vectors. Suppose there exist some positive number $R$ and $\sigma^2$ such that*

$$\mathbb{E}\left[\|\boldsymbol{x}_k\|^m\right] \leq \frac{m!}{2}\sigma^2 R^{m-2}, \quad \text{for all integers } m \geq 2.$$

*Let $\boldsymbol{s} = \frac{1}{p}\sum_{k=1}^{p}\boldsymbol{x}_k$, then for any $t > 0$, it holds that*

$$\mathbb{P}\left[\|\boldsymbol{s} - \mathbb{E}\left[\boldsymbol{s}\right]\| \geq t\right] \leq 2(d+1)\exp\left(-\frac{pt^2}{2\sigma^2 + 2Rt}\right). \tag{9.7}$$

**Lemma 9.6 (Moment-Control Bernstein's Inequality for Matrix RVs, Theorem 6.2 of [Tro12])** *Let $\boldsymbol{X}_1, \ldots, \boldsymbol{X}_p \in \mathbb{R}^{d \times d}$ be i.i.d. random, symmetric matrices. Suppose there exist some positive number $R$ and $\sigma^2$ such that*

$$\mathbb{E}\left[\boldsymbol{X}_k^m\right] \preceq \frac{m!}{2}\sigma^2 R^{m-2}\boldsymbol{I}, \tag{9.8}$$

$$-\mathbb{E}\left[\boldsymbol{X}_k^m\right] \preceq \frac{m!}{2}\sigma^2 R^{m-2}\boldsymbol{I}. \tag{9.9}$$

*for all integers $m \geq 2$. Let $\boldsymbol{S} \doteq \frac{1}{p}\sum_{k=1}^{p}\boldsymbol{X}_k$, then for all $t > 0$, it holds that*

$$\mathbb{P}\left[\|\boldsymbol{S} - \mathbb{E}\left[\boldsymbol{S}\right]\| \geq t\right] \leq 2d\exp\left(-\frac{pt^2}{2\sigma^2 + 2Rt}\right). \tag{9.10}$$

**Lemma 9.7 (Bernstein's Inequality for Uncentered Matrix RVs)** *The matrix Bernstein inequality states that for independent random matrices $\boldsymbol{M}_1, \ldots, \boldsymbol{M}_n \in \mathbb{R}^{d_1 \times d_2}$, if*

$$\sigma^2 = \max\left\{\left\|\sum_{i=1}^{n}\mathbb{E}[\boldsymbol{M}_i\boldsymbol{M}_i^*]\right\|, \left\|\sum_{i=1}^{n}\mathbb{E}[\boldsymbol{M}_i^*\boldsymbol{M}_i]\right\|\right\}, \tag{9.11}$$

*and*

$$\|\boldsymbol{M}_i\|_2 \leq R \qquad a.s., \tag{9.12}$$

*then*

$$\mathbb{P}\left[\left\|\sum_i \boldsymbol{M}_i - \mathbb{E}\left[\cdot\right]\right\| > t\right] \leq (d_1 + d_2)\exp\left(\frac{-t^2/2}{\sigma^2 + 2Rt/3}\right). \tag{9.13}$$

**Proof** For zero mean random matrices

$$\boldsymbol{M}_1 - \mathbb{E}\boldsymbol{M}_1, \ldots, \boldsymbol{M}_n - \mathbb{E}\boldsymbol{M}_n \in \mathbb{R}^{d_1 \times d_2}, \tag{9.14}$$

we have that

$$\|\boldsymbol{M}_i - \mathbb{E}\boldsymbol{M}_i\|_2 \leq 2R, \tag{9.15}$$

and

$$\boldsymbol{0} \preceq \sum_{i=1}^{n}\mathbb{E}[(\boldsymbol{M}_i - \mathbb{E}\boldsymbol{M}_i)(\boldsymbol{M}_i - \mathbb{E}\boldsymbol{M}_i)^*] \tag{9.16}$$

$$\preceq \sum_{i=1}^{n}\mathbb{E}[\boldsymbol{M}_i\boldsymbol{M}_i^*], \tag{9.17}$$

$$\boldsymbol{0} \preceq \sum_{i=1}^{n}\mathbb{E}[(\boldsymbol{M}_i - \mathbb{E}\boldsymbol{M}_i)^*(\boldsymbol{M}_i - \mathbb{E}\boldsymbol{M}_i)] \tag{9.18}$$

$$\preceq \sum_{i=1}^{n} \mathbb{E}[\boldsymbol{M}_i^* \boldsymbol{M}_i]. \tag{9.19}$$

Plugging corresponding quantities back to Theorem 1.6 of [Tro12], we obtain that

$$\mathbb{P}\left[\left\|\sum_i \boldsymbol{M}_i - \mathbb{E}\left[\cdot\right]\right\| > t\right] \leq (d_1 + d_2) \exp\left(\frac{-t^2/2}{\sigma^2 + 2Rt/3}\right). \tag{9.20}$$

∎