[Reviews · NeurIPS 2018]

Reviewer 1



This paper considers sparse blind deconvolution. This is an inverse problem where we observe the (cyclic) convolution of two unknown signals: one sparse, and one short. The aim is to recover both signals, up to fundamental indeterminacies. This is done through non-convex optimization with a soft sparsity promoting cost function. The main contributions are an analysis of regimes where the non-convexity is benign. Importantly, there is no noise in the proposed model. This is an interesting problem. Both the problem and the proposed approach I would deem fairly original. Exposition is satisfactory, and results are interesting. There are two main points I would like the authors to clarify. A priori, I expect the authors will be able to resolve these two doubts I have, which is why my initial rating is high. But if I cannot be convinced, I will have to lower my grade. First, does the proposed theory cover more than a subclass of this problem which is trivial? Specifically, see the rate at L69. What does this mean exactly? From the model detailed after L69, about m\theta entries of x_0 are nonzero. If k is such that there is a good probability that just one nonzero entry is separated from any other nonzero entry by at least k zeros, then the kernel a_0 can be read off directly from the observation just by spotting the presence of two exact zeros around the signal, which is then only scaled by the corresponding g_i. Thus, to make sense of the sparsity rate (and to know if it is "relatively dense" as stated in the paper), it should be shown that under the proposed setting, this situation does not occur with large probability. Otherwise, the problem appears to be trivial. This issue could also be resolved by adding noise to the observation y, in which case the exact zeros would not appear, and it wouldn't be all that easy to locate the separated occurrences. More generally, it would be nice to treat the presence of noise, but this is not a requirement. Second, line 65, the word "benign" (and many other places where this sentiment is echoed): it is unclear to me that this effect is benign. Is it not the case that the shift truncation could eliminate most if not all of the signal? If that is the case, what is it really that we can hope to recover? In other words, does the measure proposed on line 253 really capture closeness to a meaningful version of a_0? Or are some of these shift truncated versions of a_0 somehow significantly shortened to the point that we didn't recover much? Other comments: Figure titles, legends, axis labels etc.: a lot of them are illegible. Please fix this. Eq. (3): at this point, the modulo operator has not been defined yet. On that note: notation would be a lot nicer with 0 indexing, as this would ease all the modulo business. Line 44: , -> . L62,63: this is nice intuition. Page 2, footnote 2: is it on purpose that y_1 is not moved? If so, please specify so readers are not left wondering if this is a typo. L78: what is entry-wise \ell^p norm? Also, this could conceivably interfere with the definition of _2 for operator norm. I understand that this is why you use the Frobenius norm for vectors. An alternative is to write \|.\|_{op}. In any case: there is some confusion in the paper itself about these norms. See for example eq. (9) where the 2 norm is presumably a Frobenius norm, since the input is a vector. Eq. (6): the matrix seems wrongly typeset L108: This justification (footnote 8) is a bit weak: one needs to argue first that YY'/m concentrates around its mean, then to argue that the same goes for the inverse matrix square root of it, and this this all happens nicely considering YY' appears in a product with other random variables. Footnote 5: you get m-1 such contributions -- then, how can I understand this as small? L143 (and likely elsewhere): optima is the plural of optimum (and minima/minimum, maxima/maximum) L148: small \kappa, but eq. (19) seems to say small kappa (condition number) is bad; what am I missing? L204: semidefinite -> should be definite. L205: convex -> should be locally geodesically convex L209-211: I do not understand this statement. Please clarify. L220, end: that -> such that L249: what does big mean here? very sparse or not very sparse? L249: move footnote mark outside parenthesis (and after period) L253: recover+y ####### Post rebuttal ####### I thank the reviewers for their thorough response to my two main concerns. Based on their convincing explanations, I maintain my assessment of 8. I strongly encourage the authors to include the points raised in their rebuttal in their paper in some form.

Reviewer 2



This paper focuses on short and sparse blind deconvolution problems. A nonconvex optimization approach is proposed to solve blind deconvolution, and its landscape is analyzed in details. Some experiments are provided to demonstrate the effectiveness of the approach, and to support the theoretical analysis. Strength: 1. The paper is well organized with detailed derivations and explanations on key ideas (e.g., derivation of nonconvex optimization problem and local geometry analysis); 2. Such a nonconvex optimization approach and its landscape analysis are novel and interesting. Moreover, the technical content of the paper appears to be correct albeit some small mistakes. Weakness: The paper appears to be technically sound, but the experiments are limited without much explanation. Since the paper is partially inspired by [16], a comparison may be included to further illustrate the effectiveness of the proposed method. Some related references are missing: Li et al., Optimal Rates of Convergence for Noisy Sparse Phase Retrieval via Thresholded Wirtinger Flow, 2016 Yang et al. Misspecified Nonconvex Statistical Optimization for Sparse Phase Retrieval, 2017 Li et al. Symmetry, Saddle Points and Global Optimization Landscape of Nonconvex Matrix Factorizaiton, 2016 Ge et al. No Spurious Local Minima in Nonconvex Low Rank Problems: A Unified Geometric Analysis, 2017 There are also some minor issues (ty¬¬pos): 1. The letter $m$ stands for two quantities simultaneously: the dimension of the observed signal y and the number of samples; 2. The words font does not follow the standard NIPS template; 3. Figures may be enlarged, since some content and axes are hardly readable.

Reviewer 3



This paper considered short and sparse blind deconvolution problem and proposed to maximize the ||x||_4^4 norm that kind of promotes sparsity and at the same time, is differentiable. The main result is to analyze the geometry of the resultant nonconvex optimization problem and show that every local optimum is close to some shift truncation of the ground truth. This paper has many overloading notations which make the paper not easy to follow, but overall it is well written. Compared to the main reference [16], this paper utilized a somewhat novel objective function, and the main results for geometrical analysis have less assumptions compared to that in [16].